



# The spatially distributed nature of subglacial sediment dynamics: using a numerical model to quantify sediment transport and bedrock erosion across a glacier bed in response to glacier behavior and hydrology

Ian Delaney[1], Leif S. Anderson[1,2], and Frédéric Herman[1]

[1]Institut des dynamiques de la surface terrestre (IDYST), Université de Lausanne, Bâtiment Géopolis, CH-1015 Lausanne
[2]Department of Geology and Geophysics, University of Utah, Frederick Albert Sutton Building, 115 S 1460 E,Salt Lake City, UT 84112-0102, USA

**Correspondence:** Ian Delaney (IanArburua.Delaney@unil.ch)

**Abstract.** In addition to ice and water, glaciers expel sediment. As a result, changing glacier dynamics and melt will result in changes to glacier erosion and sediment discharge, which can impact the landscape surrounding retreating glaciers, as well as communities and ecosystems downstream. To date, the available models of subglacial sediment transport on the sub-hourly to decadal-scale exist in one dimension, usually along a glacier's flow line. Such models have proven useful in describing

the formation of landforms, the impact of sediment transport on glacier dynamics, the interactions between climate, glacier dynamics, and erosion. However, because of the large role of sediment connectivity in determining sediment discharge, the geoscience community needs modeling frameworks that describe subglacial sediment discharge in two spatial dimensions over time. Here, we present SUGSET_2D, a numerical model that evolves a two-dimensional subglacial till layer in response to the erosion of bedrock and changing sediment transport conditions below the glacier. Experiments employed on test cases of

synthetic ice sheets and alpine glaciers demonstrate the heterogeneity in sediment transport across a glacier's bed. Furthermore, the experiments show the non-linear increase in sediment discharge following increased glacier melt. Lastly, we apply the model to Griesgletscher in the Swiss Alps where we use a parameter search to test model outputs against annual observations of sediment discharge measured from the glacier. The model captures the glacier's inter-annual variability and quantities of sediment discharge. Furthermore, the model's capacity to represent the data depends greatly on the grain size of sediment.

Smaller sediment sizes allow sediment transport to occur in regions of the bed with reduced water flow and channel size, effectively increases sediment connectivity into the main channels. Model outputs from the three test-cases together show the importance of considering heterogeneities in water discharge and sediment availability in two-dimensions.

## 1   Introduction

Increasing glacier ablation will change the ways that glaciers erode bedrock and supply sediment to downstream sources (e.g.

Church and Ryder, 1972; Lane et al., 2017; Milner et al., 2017). In the Alps already, increased fluxes of sediment in the Rhône Valley have been identified at the basin scale in response to increased glacier melt (Costa et al., 2018; Lane et al., 2019).



Changes to sediment discharge have also been identified in the Arctic, where large proglacial deltas have grown off the coast of Greenland in recent decades (Bendixen et al., 2017).

The impact of changes to sediment discharge from glaciers on mountains or high-latitude landscapes, as well as many downstream social and earth systems implies a need for predictive models to understand the response of these systems to glacier retreat. In alpine environments, increased sediment discharge results in the in-fill of proglacial hydropower reservoirs (Thapa et al., 2005) and abrasion of hydropower infrastructure (e.g. Felix et al., 2016). Additionally, the flux of sediment from glaciers dramatically alters alpine ecosystems (Milner et al., 2017). In the Arctic, increased sediment discharge can affect biogeochemical cycles given that sediments may carry phosphorus and iron (Bhatia et al., 2013; Hawkings et al., 2014). These elements limit bio-availability in the Arctic, so any change to sediment discharge from the ice sheet may substantially alter Arctic ecosystems (Wadham et al., 2019). Modeling studies and observations suggest that increases in sediment output from alpine glaciers could occur when high melt extends up-glacier allowing sediment transport where no sediment could be transported before (Lane et al., 2017; Delaney and Adhikari, 2020).

Generally, two processes determine the sediment discharge below a glacier by adding or removing sediment from a subglacial till layer (Figure 1; Brinkerhoff et al., 2017; Delaney et al., 2019). Bedrock erosion adds material to the subglacial till layer. Bedrock erosion occurs by quarrying, when pressure differentials on opposing sides of obstacles cause fractures to expand and rock to detach (Iverson, 1990; Alley et al., 1997; Hallet et al., 1996; Iverson, 2012), and abrasion, when debris embedded in the ice grinds bedrock as the glacier slides above (Hallet, 1979; Alley et al., 1997). Physically describing these processes requires evaluating a large number of parameters (c.f. Ugelvig et al., 2018), so many researchers use empirical relationships that relate glacier sliding to glacier erosion. In doing so they represent processes such as abrasion and quarrying in their models (Humphrey and Raymond, 1994; Koppes et al., 2015; Herman et al., 2015; Cook et al., 2020). This sliding relationship proves especially useful in applications over large temporal and spatial scales, such as evaluating the interaction between glacier erosion and tectonics (e.g. Egholm et al., 2009; Prasicek et al., 2018; Herman et al., 2018; Prasicek et al., 2020).

Conversely, fluvial mobilization can remove sediment from the till layer below the glacier (e.g. Walder and Fowler, 1994; Ng, 2000; Creyts et al., 2013). When pressurized subglacial water increases above a critical velocity, then substantial amounts of sediment of a given grain size begin to be transported (Shields, 1936). However, the amount of sediment that may be mobilized given sediment transport conditions depends on the presence of sediment (e.g. Mao et al., 2014). In this way, fluvial sediment transport depends both on the subglacial hydraulic characteristics (e.g. Walder and Fowler, 1994) as well as the availability of sediment at the glacier bed (e.g. Willis et al., 1996; Swift et al., 2005).

Bedrock erosion and fluvial sediment transport vary depending on the timescales and characteristics of individual glaciers. For instance, erosive processes likely dominate over sediment transport processes on glaciers with minimal sediment storage and steep gradients (Humphrey and Raymond, 1994; Herman et al., 2015, 2021). Furthermore, landscape evolution models, that use empirical relationships to quantify erosion with respect to sliding speed, demonstrate the dominant role of erosional processes, as opposed to sediment transport ones, over geologic timescales (Harbor et al., 1988; Herman et al., 2011; Egholm et al., 2012). Conversely, over shorter timescales of months to decades, and on shallower glaciers, fluvial sediment transport is a key driver (e.g. Delaney et al., 2018b; Perolo et al., 2018; Delaney et al., 2019).





Numerical implementation of subglacial sediment transport thus far has focused on one-dimensional models (e.g. Creyts et al., 2013; Beaud et al., 2018; Hewitt and Creyts, 2019; Delaney et al., 2019). These frameworks have yielded foundational insights into the creation of eskers (Beaud et al., 2018; Hewitt and Creyts, 2019), the formation of subglacial canals

through which water flows (Walder and Fowler, 1994; Ng, 2000; Kasmalkar et al., 2019) and the behavior of tidewater glaciers (Brinkerhoff et al., 2017). Additionally, a subglacial sediment transport model demonstrates processes by which an over-deepened glacier bed causes sediment to be deposited on its adverse slope (Creyts et al., 2013). It is also been possible to reproduce measurements of sediment discharge using a subglacial sediment transport model (Delaney et al., 2019).

At the same time, sediment connectivity, the transfer of sediment from source to sink, plays a fundamental role in sediment

discharge (Figure 1; e.g. Bracken et al., 2015; Micheletti and Lane, 2016; Wohl et al., 2019; Mancini and Lane, 2020). Here, the spatial heterogeneities in the distribution of sediment and sediment transport capacity (largely controlled by water velocity) often result in less sediment being carried by the water than could be theoretically transported (e.g. Lane et al., 2017; Delaney et al., 2018b). As a result, reducing the problem to one dimension omits key processes that control sediment dynamics because subglacial water flows through spatially distributed networks of cavities and channels across the glacier bed (e.g. Werder

et al., 2013). Therefore, describing subglacial sediment transport inherently lends itself to a discretization of bedrock erosion, sediment transport, water flow, and sediment availability in two spatial dimensions.

In this manuscript, we present SUGSET_2D, a two-dimensional subglacial sediment transport model. The model implements subglacial sediment transport and bedrock erosional processes presented in Delaney et al. (SUGSET; 2019). We apply a routing scheme to the model that transports sediment down-glacier based upon the hydraulic potential gradient. Synthetic test cases

show the model's ability to reproduce known processes and yields insight into the spatially distributed processes responsible for subglacial sediment dynamics. Implementation of the model to existing glacier hydrology, topography, and sediment discharge datasets from the Griesgletscher helps to understand some subglacial sediment transport processes at this site that could be generalizable to other situations. Through these experiments, we discuss the impact of two dimensional sediment connectivity on subglacial sediment transport.

## 80  2 Model Description

The model presented here implements a hydraulic model and sediment routing scheme that translates the one-dimensional subglacial sediment transport model presented in Delaney et al. (2019) to two dimensions. In this section, we review the underpinnings of the model presented in Delaney et al. (2019), describe the routing scheme, and outline its numerical implementation in two-dimensions.

## 85  2.1 Hydraulic Model

SUGSET_2D requires a hydraulics model as a means to route sediment and water through the subglacial environment and to evaluate the sediment transport capacity of this water, based upon the hydraulic gradient, channel size and water flux (Table 1,





Section 2.2; e.g. Walder and Fowler, 1994; Alley et al., 1997). The hydraulic model here is based on the premise that subglacial water flows along the hydraulic potential gradient and glacier ice pressurizes water at the glacier bed (Shreve, 1972).

The hydraulic gradient $\Psi$ can be determined with a known hydraulic radius $D_h$ and water discharge $Q_w$, given the Darcy-Weissbach equation for fluid flow through a pipe

$$\Psi = s\, f_r\, \rho_w\, \frac{Q_w^2}{D_h^5}. \tag{1}$$

where, $s$ is a factor accounting for channel geometry (Hooke et al., 1990), $f_r$ is the Darcy-Weissbach friction factor, $\rho_w$ represents the density of water. Evaluating $\Psi$ in Equation 1 requires prescribing $Q_w$ and $D_h$.

To approximate hydraulic diameter $D_h$, we assign a representative water discharge $Q_w^*$ to $Q_w$, by taking a quantile of water discharge over a certain time (c.f. Delaney et al., 2019). This time represents the time period, assumed to be days, over-which hydraulic radius $D_h$ evolves in response to $Q_w$ (e.g. Nanni et al., 2020). We then evaluate $D_h$, the hydraulic radius given

$$D_h = \left(s\, f_r\, \rho_w\, \frac{Q_w^{*\,2}}{\Psi^*}\right)^{\frac{1}{5}}. \tag{2}$$

$\Psi^*$ is a representative hydraulic gradient at overburden pressure, evaluated using the Shreve potential gradient

$$\Psi^* = \nabla\left(\rho_i\, g\, (z_s - z_b) + \rho_w\, g\, z_b\right), \tag{3}$$

where $z_s$ and $z_b$ are surface and bed elevations, respectively, $\rho_i$ is the density of ice and $g$ is the gravitational acceleration constant.

Now equipped with $D_h$, we insert the instantaneous value of $Q_w$ into Equation 1 to evaluate the instantaneous hydraulic gradient $\Psi$. In this formulation, we assume that the timescales over which the channel size responds (days) are different than the those of instantaneous water discharge (minutes or hours). In this manner, we simulate key characteristics of an R-channel without explicitly describing properties such as creep closure and pressure melt of channel walls.

We note that to prevent unreasonable water pressures when $Q_w^*$ rapidly increases and $D_h$ is small, the model limits the minimal cross-sectional area of $D_h$ to $0.3\,\mathrm{m}^2$.

## 2.2 Till-layer model: bedrock- erosion and sediment transport

The model simulates the evolution of a subglacial till, which we define as transportable sediment below the glacier due to glacier erosion and fluvial sediment transport. Erosive processes such as abrasion and quarrying add material to the layer. Conversely, fluvial sediment transport in supply- and transport-limited regimes mobilize and deposit sediment, adding or removing material from the till layer (Brinkerhoff et al., 2017; Delaney et al., 2019). To quantify these processes, we implement the Exner Equation (Exner, 1920a,b; Paola and Voller, 2005), a mass conservation relationship, to solve for the till layer height given the erosive and fluvial conditions.

$$\underbrace{\frac{\partial H}{\partial t}}_{\text{till evolution}} = \underbrace{\dot{m}_t}_{\text{bedrock erosion}} - \underbrace{\frac{1}{l}\nabla \cdot Q_s}_{\text{sediment transport}} \tag{4}$$



$H$ is till thickness and $t$ is time (Table 1). The first term captures bedrock erosion processes, where $\dot{m}_t$ is a bedrock erosion rate. The remaining terms on the right side of the equation represent fluvial sediment transport processes, where $Q_s$ represents

115 sediment transport in either supply- and transport- limited regimes. $l$ is an characteristic length-scale for sediment mobilization, over which sediment mobilization adjusts to sediment transport conditions.

Sediment discharge is calculated using the mobilization equation from Delaney et al. (2019).

$$
\nabla \cdot Q_s = 
\begin{cases}
\dfrac{Q_{sc} - Q_s}{l} & \text{if } \frac{Q_{sc} - Q_s}{l} \leq \dot{m}_t & \text{(5a)} \\[2ex]
0 & \text{if } H = H_{lim} \quad \& \quad \frac{Q_{sc} - Q_s}{l} \leq 0 & \text{(5b)} \\[2ex]
\dfrac{Q_{sc} - Q_s}{l} \, \sigma(H) + \dot{m}_t \cdot (1 - \sigma(H)) & \text{otherwise} & \text{(5c)}
\end{cases}
$$

$Q_{sc}$ is sediment transport capacity, or the amount of sediment that could be transported given hydraulic conditions. $\sigma$ is a sigmoidal function of $H$

$$
\sigma(H) = \left(1 + \exp\left(\frac{2 - \Delta\sigma H}{5}\right)\right)^{-1}, \tag{6}
$$

that enables smooth transition over the range: $H = 2\Delta\sigma^{-1} \pm \Delta\sigma^{-1}$ in Equation 5c. $\Delta\sigma$ is a value below which $\sigma$ substantially deviates from 1, and reduces sediment mobilization.

Condition 5a represents the case where bedrock erosion exceeds sediment mobilization, thus sediment transport exists in a transport -limited regime. Condition 5b impedes mobilization or deposition, transporting sediment to the next cell when a till thickness is equal to $H_{lim}$, the value of which is chosen to be on the order of maximal change in till height over the model run

($\sim 10\,\text{cm}$). This term prevents unbounded sediment accumulation in over-deepenings where excessive deposition can occur (Alley et al., 2003). Mechanisms to capture processes occurring in over-deepenings are not included in the model. Condition 5c allows sediment mobilization to smoothly transition between transport- and supply-limited regimes. In this case, when $H$ is small, sediment mobilization is limited to the sediment production term $\dot{m}_t$.

We calculate sediment transport capacity $Q_{sc}$ using the total sediment transport relationship by Engelund and Hansen (1967),


$$
Q_{sc} = \frac{0.4}{f_r} \, \frac{1}{D_{m_{50}} (\frac{\rho_s}{\rho_w} - 1)^2 g^2} \, \left(\frac{\tau}{\rho_w}\right)^{\frac{5}{2}} w_c, \tag{7}
$$

where $\rho_s$ ($\rho_w$) is the bulk density of the sediment (water), $D_{m_{50}}$ is the mean sediment grain-size, $w_c$ denotes width of the channel floor that integrates the sediment transport rate across the width of the channel (c.f. Delaney et al., 2019) and $\tau$ represents the shear stress between the water and the channel bed. We determine the shear stress through the Darcy-Weisbach

formulation:

$$
\tau = \frac{1}{8} f_r \rho_w v^2, \tag{8}
$$

where $v = \frac{Q_w}{S}$ is the water velocity and $S$ is evaluated in Equation 1. We note that because Engelund and Hansen (1967)'s formulation relies on shear stress, other sediment transport relationships could be exchanged by the model operator (e.g. Meyer-





Peter and Müller, 1948). We chose Engelund and Hansen (1967)'s formulation due to the representation of both suspended and
bedload transport. Furthermore, the continuous nature of the relationship improves the model stability.

Source term $\dot{m}_t$ is described as,

$$\dot{m}_t = \dot{e}\left(1 - \frac{H}{H_{max}}\right), \tag{9}$$

where $H_{max}$ is a till height beyond which no further erosion, $\dot{e}$ may occur.

We chose to use an empirical relationship with sliding velocity $u_b$ to describe bedrock erosion,

$$\dot{e} = k_g \, u_b{}^{l_{er}}, \tag{10}$$

$k_g$ is an erodability constant and $l_{er}$ is an exponent, which varies from between 0.66 and 3 (Herman et al., 2021). $u_b$ is
assumed to be a given fraction $f_{sl}$ of the ice deformation velocity (e.g. Weertman, 1957), and is calculated via the shallow ice
approximation (Hutter, 1983):

$$u_b = f_{sl} \frac{2A}{n+1} \left(\rho_i \, g \sin\alpha\right)^n \left(z_s - z_b\right)^{n+1}. \tag{11}$$

Here, $n$ is the exponent in Glen's flow law, $A$ is the ice flow rate factor, and $\alpha$ is the surface slope. Note that because $\dot{m}_t$ is a
source term, alternative parameterizations of erosion can easily be exchanged, as we do in some cases below.

## 2.3 Spatial and temporal discretization, and parameters

Here, we describe the numerical implementation of the equations presented above, and in particular the routing scheme that
enables a two dimensional representation of subglacial fluvial and till dynamics.

### 2.3.1 Routing algorithm and implementation

Sediment and water are routed down the hydraulic gradient using a multi-cell routing scheme (Quinn et al., 1991), implemented
in a similar way as Bovy et al. (2016) and Braun and Willett (2013) on a regular grid. Sediments and water move from one
cell to another using a steepest-descent algorithm. For each cell, a list of receivers (in other words cells that receive water or
sediment from another cell), $r_s$, are established by searching for the surrounding four neighboring cells that share an edge for a
positive hydraulic gradient. The receiver cells and the positive hydraulic potentials are then stored in an array. For each cell, the
positive hydraulic gradients are summed and divided by the number of receivers, $n_r$, to establish the weight or percentage of
flow, $w_r$, to each receiver, using the array. The algorithm then uses the information about the number of receivers to determine
the donor cells (in other words cells from which the given cell receives water of sediments), $d_n$. Note there can be several donor
cells, $n_d$, for each cell. Equipped with the information about the receivers and donors from each cell, the algorithm creates a
stack $s_t$, a vector of cells ordered to perform operations.

These algorithms are encoded in a function,

$$s_t, r_s, n_r, w_r, n_d = \text{make\_stack}(f_f). \tag{12}$$





The flotation fraction for a cell is calculated by $f_f = \frac{\phi}{\phi^*}$, where hydraulic potential $\phi$ comes from integrating Equation 1 up the stack to a cell and $\phi^*$ comes from determining the Shreve potential at overburden. We distribute the mean value of $f_f$ across the glacier, then implement the routing scheme for the hydraulic potential determined from the Shreve potential. Here, we also assume that water does not inherit previous channels or canals (Figure 2; Shreve, 1972; Werder et al., 2013; Zechmann et al.,
2020). The user can select whether the node ordering remains fixed for a given flotation fraction, $f_f$, over the model run or whether the node ordering evolves in response to hydraulic forcing. In the model implementation we chose, the node ordering algorithm is executed every time step to account for diurnal variations in water pressure (e.g. Iken and Bindschadler, 1986), which unavoidably increases simulation times. However, to improve stability during periods of rapidly changing sediment transport conditions, we reorder the stack, based upon the hydraulic conditions to the nearest $6\,\mathrm{mn}$. Smaller solving tolerances
increase the computational time (Figure 3 e, f) due to 1) increased accuracy of the solution and 2) the reassessment of flow fractions between the adjacent cells, which results in different routing configurations as the model converges. We fill closed basins or over-deepenings in the hydraulic potential to maintain continuous sediment transport through the domain. The model uses an external algorithm, slightly modified, from the package *WhereTheWaterFlows.jl*[1] that contains flow routing and basin filling algorithms based upon rasterised DEMs.

### 2.3.2  Numerical implementation and parameter constraints

We discretize Equation 4 over the model domain space using a finite volume scheme on a regular grid. Spatial discretization must be substantially smaller than characteristic length, $l$, in Equation 4.

For the discretization in time, the model implements the VCABM solver (Hairer et al., 1992; Radhakrishnan and Hindmarsh, 1993) from the package *DifferentialEquations.jl* (Rackauckas and Nie, 2017) to evolve till layer height $H$. This solver imple-
ments an adaptive time step and uses a linear multi-step method (Adams Moulton) that is well-suited for non-stiff problems. We impose a maximum time step of $6\,\mathrm{h}$ to ensure that the model captures the response to diurnal variations in melt input. In practice, the solver commonly uses a time step of roughly $20\,\mathrm{mn}$, which varies depending on sediment transport conditions and solver tolerance. Longer time steps occur over periods when glacier melt, and thus sediment transport, cease, such as winter months. Table 3 presents the numerical parameters used.

Benchmark experiments were conducted based upon the model setup discussed below in Section 3.2. Benchmark results show that the accuracy and performance of the model is optimized when solving tolerances (abstol, reltol, Table 3) are both on the order of $10^{-8}$ (Figure 3). Results also show that model behavior varies to a degree on grid-size due to the quantity of material in a grid cell and the flow of water though these cells (Figure 3, a,c).

We impose boundary conditions on the edge cells such that no sediment flux enters the domain. At outlet cells, a flux of
sediment leaves the domain, based upon sediment transport conditions. Boundary conditions could also be set to represent processes such as hill-slope erosion that route material to the subglacial environment (e.g. Andersen et al., 2015).

Evolving Equation 4 requires an initial till height, $H_0$, to be chosen by the model user. This initial till height represents material from bedrock erosion created prior to the model initialization. We apply a "spin-up" procedure to create a reasonable

---

[1]https://github.com/mauro3/WhereTheWaterFlows.jl





relationship between the amount of fluvial sediment transport and bedrock erosion. The amount of time needed to add material
to the till layer means that an equilibrium between fluvial sediment transport and bedrock erosion will likely take centuries or
longer to attain, if such a thing may even exist in light of variable climatic, and thus glacier, conditions. Should an equilibrium
exist, it is probably outside a feasible computational time (Delaney and Adhikari, 2020). We consider this equilibrium (or lack
thereof) and the implications of sediment storage in the subglacial environment an important research topic (e.g. Riihimaki
et al., 2005; Otto et al., 2009), yet also one that is only partially assessed in this manuscript.

In addition to the initial condition, SUGSET contains 20 parameters that influence simulated physical processes in the
model (Table 2 and 3). Subglacial hydrology models are well known to contain a large number of unconstrained parameters
(e.g. de Fleurian et al., 2018). About half of these parameters are approximated in previous studies. Work in recent years has
aimed to better evaluate these parameters through inverse methods (Brinkerhoff et al., 2016) as well as detailed modeling and
measurements (e.g. Chen et al., 2018; Covington et al., 2020). Nonetheless, the situation leaves much to be desired.

New versions of the code are tested against reference test cases to ensure that new versions remain consistent. Additionally
in each test, we ensure mass conservation by checking that the amount of sediment leaving the system through fluvial transport
is consistent with the till-height change and erosion occurring under the simulated glacier.

## 3    Model Application

We present the model in three different applications that show its viability under increasingly complex situations. Two of the
cases are synthetic scenarios based upon the Subglacial Hydrology Model Inter-comparison Project (SHMIP;  de Fleurian
et al., 2018), which provided a qualitative comparison of subglacial hydrology models using set of benchmark experiments.
First, we apply the model with a synthetic ice sheet geometry with point inputs of water to the bed (moulins). This allows us
to highlight the importance of evaluating water routing and sediment transport in two dimensions, even for the most simple
glacier and bed geometries. Second, we apply the model to a synthetic alpine glacier, also based on SHMIP, to illustrate the
model's performance in a more-complex, realistic topography forced by a seasonally varying hydrology. Lastly, we apply the
model to the topography, and sediment and water discharge at Griesgletscher in the Swiss Alps. We demonstrate the proficiency
of the model by comparing sediment transport model output and data (Delaney et al., 2018a). We also identify some drivers of
subglacial sediment discharge from these simulations.

### 3.1    Synthetic ice sheet test cases

### 3.1.1    Model parameterization and experiment design

We run synthetic ice sheet glacier experiment where the surface topography has a parabolic profile and bed topography remains
flat. The spatial domain is $100\,\mathrm{km}$ long and $20\,\mathrm{km}$ wide. The identical geometry has been applied to the subglacial hydrology
model presented in Werder et al. (2013) and SHMIP (de Fleurian et al., 2018). SHMIP also inspires the hydrology parameteri-
zation. Moulins route surface melt to the glacier bed as in Suite B4 of SHMIP (c.f. de Fleurian et al., 2018). Here, 50 moulins





are distributed across the glacier, with a greater concentration in the lower elevation sections of the glacier. The location of these moulins can be found in the code repository or at the SHMIP website[2].

To summarize the hydrology parameterization presented in Suite B4 of SHMIP, the meltwater source term, $\dot{m}_w$, is defined as,

$$\dot{m}_w(z_s) = \begin{cases} M_f\,T(z_s) & \text{if } T(z_s) > 0 \\ 0 & \text{if } T(z_s) \leq 0 \end{cases} \tag{13}$$

where $M_f = 0.01\,\mathrm{m\,K^{-1}\,d^{-1}}$ is a melt factor, and $T(z_s)$ is air temperature at elevation $z_s$.

$$T(z_s) = \left( A_d \cos\left(\frac{2\,\pi\,t}{s_{day}}\right) + \Delta T \right) \cdot \left( 1 + z_s \frac{dT}{dz} \right), \tag{14}$$

$A_d$ is the diurnal amplitude in temperature, $\Delta T$ is a temperature offset that is adjusted to control the meltwater input and $s_{day}$ is the number of seconds in one day. In these simulations, melt varies diurnally, but not seasonally.

We then apply Equation 13 to cells for which a moulin exists to localize the water source, however, $\Delta T$ is greatly increased so that the model produces realistic values of water discharge.

Here, we adjust the parameterization of erosion such that that the glacier uniformly creates $3\,\mathrm{mm\,a^{-1}}$ of till across the bed, replacing Equation 10 with

$$\dot{e} = \frac{0.003}{s_{year}}, \tag{15}$$

such that $\dot{m}_t$ still depends on till-height $H$ (Equation 9). $s_{year}$ represents the number of seconds in a year.

We run the model using only a diurnal hydrology forcing. For the first $10\,\mathrm{a}$ a steady forcing of $\Delta T$ is applied. After $10\,\mathrm{a}$, $\Delta T$ is increased $1.5°$ every year for $10\,\mathrm{a}$ then run for another 10, such that total model run time is $30\,\mathrm{a}$. The model run initiates with $5\,\mathrm{cm}$ of till at the bed, the limit of till grown ($H_{max}$ Equation 9). Quantities of sediment below glaciers remain generally unknown and may be quite substantial (e.g. Truffer et al., 2000). We chose $5\,\mathrm{cm}$ as an initial condition as we predict that it captures erosion processes on the millennium timescales, given erosion rates of $1$–$3\,\mathrm{mm\ a^{-1}}$ (Hallet et al., 1996). A spin-up is conducted over the initial hydrological conditions until the annual erosion rate reaches $10^{-4}\,\mathrm{mm\,a^{-1}}$. We believe this spin-up sets the conditions for quantifying hourly sediment transport in light of the millenia scale quantities of bedrock erosion, that represent our initial condition.

### 3.1.2 Model outputs and findings

Over the course of the model run, water discharge increases roughly $10\,\%$ between year 10 and year 20. Over the corresponding period, sediment discharge increases to roughly $2\,\%$ from the pre-perturbation values of sediment discharge. At the same time, the general trend of increasing till-height persists through the climate forcing, although at a slower rate (Figure 4 a). Sediment transport capacity, $Q_{sc}$ lies at over 10 times sediment discharge.

---

[2]https://shmip.bitbucket.io/instructions.html





In our simulations, sediment transport occurs in a very limited portion of the glacier bed, where the water flows. Thus till-height remains consistent through the model run across the vast majority of the domain (Figure 5). At the same time, even this localized water flow and sediment transport produces catchment erosion rates of roughly $1\,\mathrm{mm}\ \mathrm{a}^{-1}$, which is on the order of some measured erosion rates from the Greenland Ice Sheet (Cowton et al., 2012; Overeem et al., 2017), but also in mountainous regions (Hallet et al., 1996).

Sediment discharge in the model run decreases after the climate warms and reaches a stable regime. The decrease occurs due to sediment exhaustion from increased water discharge, and thus sediment transport, which removed till that was unable to be transported in a cooler climate with less available meltwater able to transport sediment.

We note that till-height along certain channels is discontinuous, and the till-height increases from where the water enters the glacier through a moulin to the terminus (Figure 5). At a certain distance from the moulin (roughly $2\,\mathrm{km}$ in this case), subglacial water attains its transport capacity and cannot mobilize additional sediment at the same rate as higher on the glacier. We note that due to sediment exhaustion these features are transient, and their persistence, location and size will respond to the sediment source term ($\dot{m}_t$) and the sediment e-folding length ($l$). These bare bedrock patches serve as a source of sediment that is in part responsible for the increase in sediment discharge after the climate forcing at the end of the model run (Figure 5).

Given our routing parameterization and the laterally consistent geometry of the glacier, this test case essentially represents many one-dimensional models set up in parallel, as each cell has a single receiver cell. Yet, even in this simple case, collapsing the problem to a single dimension means that sediment thickness will either be averaged over the entire glacier width, or that only a single moulin or water source can be considered. In light of these experiments, we suggest that in most applications where the researcher's interest lies in sediment discharge, two-dimensional consideration of sediment dynamics more adequately captures relevant processes than a one-dimensional framework.

## 3.2 Synthetic alpine test cases

### 3.2.1 Model parameterization and experiment design

We run simulations using an alpine glacier geometry and hydrological forcing following the SHMIP project experiments (de Fleurian et al., 2018)). The domain is $6000\,\mathrm{m}$ on one axis and $1080\,\mathrm{m}$ on the other and the resulting geometry approximates the Bench Glacier. The U-shaped bed and variable ice-sheet thickness mean that variable hydrologic gradients will occur laterally across the glacier and water can be routed across multiples cells, unlike the ice sheet case from Section 3.1.

To parameterize hydrology, we return to the melt model in Equation 13. Temperature $T$ at elevation $z_s$ for this test case is defined as

$$T(z_s) = \left(-A_a \cos\left(\frac{2\pi t}{s_{year}}\right) + A_d \cos\left(\frac{2\pi t}{s_{day}}\right) + \Delta T - 5\right) \cdot \left(1 + z_s \frac{dT}{dz}\right), \tag{16}$$

where $A_a$ and $A_d$ are the annual and diurnal amplitudes, respectively, $\Delta T$ is a temperature offset, which is adjusted to control the meltwater input and $s_{day}$ are the number of seconds in one day, $s_{year}$ is the number of seconds in a year and $\frac{dT}{dz} =$



$-0.0075\,\mathrm{K\,m^{-1}}$ is the air temperature lapse rate. In this test case, we route water directly to the glacier bed at the location where the melt occurs, ignoring moulins that concentrate meltwater delivery to the bed.

Similar to the ice sheet case (Section 3.1.1), we run the model for 12 years with a steady climate forcing, then we apply a gradual climate forcing for 8 years followed by 10 years of steady climate forcing at the maximal $\Delta T$. The annual temperature signal experiences randomly distributed noise to mimic the effects of inter-annual variability. A spin-up over one year of the initial hydrological forcing is applied until either $150\,\mathrm{a}$ is reached or an annual change in the till layer height is less than $10^{-4}\,\mathrm{mm\,a^{-1}}$, i.e., well below the annual erosion rate in most glacierized catchments (Hallet et al., 1996).

## 3.2.2   Model outputs and findings

Model outputs show that over seasonal timescales, sediment discharge increases at the onset of melt and decreases shortly thereafter, prior to the maximum amount of water discharge that occurs each melt season (Figure 6). Daily-averaged sediment discharge decreases until the very end of the melt season when sediment discharge increases again (Figure 8). This occurs when water stops flowing during the night and sediment may accumulate in the channels from bedrock erosion. Increased sediment

discharge at the beginning of the melt season results from greater sediment availability following the growth of the till layer over the winter months, when little water is available for sediment transport.

Increases in sediment discharge at the onset of melt and subsequent exhaustion of sediment have been documented in field observations (Willis et al., 1996; Swift et al., 2005; Riihimaki et al., 2005; Delaney et al., 2018b) and reproduced in the one-dimensional version of this model (Delaney et al., 2019). However, in SUGSET_2D, larger diurnal increases in sediment

discharge occur because water only accesses certain patches of the glacier bed for a limited time during the diurnal cycle. This transient flow maintains the till layer there, allowing it to even grow slightly from bedrock erosion. The temporal variability in meltwater access to portions of the glacier bed lies contrary to the one-dimensional model, where meltwater access to the glacier bed is only limited along the glacier's longitude.

Over the course of the model run, the mean till-height continues to decrease over annual scales, despite the spin-up threshold

of $10^{-4}\,\mathrm{mm\,a^{-1}}$, highlighting the difficulty in evaluating an equilibrium between bedrock erosion and sediment transport (Figure 6,c). However, till-height decrease accelerates following the onset of increased melt at year 12. Over the same periods, annual amounts of sediment discharge increased due to decreased subglacial sediment storage (Figure 7). When a new steady climate was reached at year 20, annual quantities of sediment discharge began to decrease, as the system approached a new equilibrium between sediment transport and bedrock erosion. This process also occurred in the ice sheet test case (Section 3.1).

Interestingly, the model recreates "first-flush" events of increased sediment discharge early in the melt season (Figure 8), followed by decreased sediment discharge (Swift et al., 2005; Delaney et al., 2018b). This seasonal evolution in sediment discharge has been attributed to increased access to subglacial sediment early in the season, followed by decreased access to sediment when the flow is channelized (e.g. Willis et al., 1996; Swift et al., 2005). In a warmer climate, seasonal sediment discharge increases, largely due to increased sediment transport high on the glacier (Figure 6). However, seasonal maximum

values of sediment discharge during "first-flush" event decrease because sediment transport also occurs over that time. Winter sediment transport prevents the till layer from growing over the winter months, creating a surplus of sediment to be transported





in the spring. While this model output could be considered interesting, we note that the model does not couple ice dynamics to the hydrology, so erosive potential, water discharge and subglacial area will likely decrease as well in response to the new climate. The increase could also occur with increased sliding, and thus erosion, following melt (Ugelvig et al., 2018).

For the test cases above, bedrock erosion essentially relied on driving stress and thus sliding and bedrock-erosion did not vary seasonally, except for limits on sediment production with till-layer height (Figure 8 a, b, Section 3.2). This causes a buildup of sediment during the winter months, which subsequently provided ample material for transport when melt increases in the spring. The model's bedrock erosion scheme is likely valid on land-terminating glaciers and over long-time scales when driving stress exerts the primary control on glacier sliding (e.g. Weertman, 1957). However, by coupling subglacial hydrology

to erosion Ugelvig et al. (2018) showed that erosion varies seasonally and abrasion largely occurs solely during the summer months. Additionally, in the test case presented above, sediment production occurred primarily near the glacier front, where ice deformation, and thus sliding, is greatest (Figure 2, a).

To test the effects of spatially variable erosion and the role of hydrology, we present two additional test cases to supplement the alpine glacier test case above, *ORIGINAL*.

The first test case, *SEASON*, simulates bedrock erosion by increasing sliding during the summer months (e.g. Iken and Bindschadler, 1986), the same erosion relationship is applied as the case as Section 3.2, however, erosion only occurs when the amount of water input substantially exceeds the basal melt-rate. In the second test case, *CONST*, bedrock erosion remains constant over the entirety of the glacier at a rate of $1 \, \mathrm{mm \, a^{-1}}$.

The *ORIGINAL* test case discharges over $4650 \, \mathrm{m^3}$ of sediment per year, while the *SEASON* case discharged only 60% of

that value due to the absence of bedrock erosion during the winter months. The *CONST* case discharged $2160 \, \mathrm{m^3}$ of sediment over the year. This value is substantially less than the $1 \, \mathrm{mm \, a^{-1}}$ erosion rate due to decreased erosion efficiency with till height and the limited portion of the bed over-which sediment transport occurs (Figure 7).

Over the three cases, sediment discharge increases at the onset of melt and substantially decreases by the end of the melt season due to sediment exhaustion. In *ORIGINAL* (Figure 8 a, b), more sediment discharge occurs compared to the alternate

test cases (*SEASON* and both due 1) to the prolonged period over which bedrock erosion occurs adding more sediment to the layer and 2) that bedrock erosion occurs low on the glacier where much sediment transport takes place, compared to the *CONST* case. The peak sediment discharge in *CONST* (Figure 8 e, f) occurs slightly earlier in the season, due to the increased amounts of sediment on the lower glacier margins.

### 3.3  Griesglestcher

#### 3.3.1  Model parameterization and experiment design

Lastly, we apply the model to the Griesgletscher in the Swiss Alps using topographic data from (Delaney et al., 2019). Hourly water discharge from the glacier was modeled in Delaney et al. (2018a). Here, we use the time series from 2009–2017. Sub-glacial sediment discharge from the glacier was determined for four different time periods since fall 2011 by differencing



repeated bathymetry maps (Delaney et al., 2018a). To distribute glacier melt across the glacier with respect to elevation, we
use,

$$\dot{m}_w(x,y) = \dot{b}^0 + \gamma(z_s(x,y) - z_s^0). \tag{17}$$

$\gamma$ is the mass balance gradient and $z_s^0$ represents the glacier's lowest elevation. $\dot{b}^0$ represents the melt rate at the glacier's lowest
extent. $\dot{b}^0$ was evaluated numerically at each water discharge value using the hypsometry of the glacier.

We apply a parameter search over a range of values of sediment grain-size ($D_m$; a primary control on fluvial transport of
subglacial sediment), sliding velocity ($f_{sl}$; a control on bedrock erosion), and the initial till height condition ($H_0$; to approxi-
mate the effects of existing quantities of sediment below the glacier). 100 model runs were executed with randomly selected
parameters a uniform distribution. No spin-up was applied in this test, because of computational expense and the wide range
of $H_0$ values explored.

The wall time for single model run averaged $8.9\,\mathrm{h}$ and each run for a parameter set was executed on a single CPU. Instead of
applying the mean flotation fraction across the glacier, as was done in the previous test cases, the maximum value was applied
with an upper limit of $1$.

Only model outputs resulting in a perfect rank correlation across the four data collection periods and an error less than
$85,000\,\mathrm{m}^{-3}$ were considered. For the test case presented below we only show the simulation with the lowest absolute error
between model output and the sediment transport data.

### 3.3.2   Model outputs and findings

The model successfully captured the inter-annual variability in sediment discharge from the Griesgletscher. The absolute error
between the model and the measurements is roughly $62,477\,\mathrm{m}^3$. The model captures the last three time-spans very well, but it
has trouble reconciling the increased sediment discharge in 2012 and 2013 (Figure 9). This suggests that processes not included
in the model are responsible for the increase in sediment transport, such as activation of new patches of the glacier bed or the
relocation of channels (e.g. Zechmann et al., 2020), potentially due to changes to glacier surface topography. Furthermore,
sliding parameter $f_{sl}$ remains constant over the model run in the absence of glacier velocity data, in turn, inter-annual variability
in bedrock erosion is not considered (Herman et al., 2015).

The error from this parameter search is slightly less than half of the $131,300\,\mathrm{m}^3$ total sediment discharged from the Gries-
gletscher over this time period (Delaney et al., 2018a), and the error is slightly more than the $58,300\,\mathrm{m}^3$ from the best model
run of the one-dimensional model (c.f. Delaney et al., 2019). However, in contrast to the ensemble model runs in Delaney
et al. (2019) this model's ability to reproduce the validation data largely depends on the grain-size parameter, $D_m$. Compared
to Delaney et al. (2019), the sliding fraction and initial condition parameters ($f_{sl}$ and $H_0$) have a minimal influence when tuned
to the data here Figure 9. This is due to the subglacial sediment connectivity parameterized in this two dimensional version
of the model. In SUGSET_2D, the channelized nature of flow means that sediment transport may only occur over a relatively
narrow patch of the glacier bed (Figure 11). As sediment grain-size decreases, sediment from locations of the glacier bed with




relative small water velocity and discharge can more easily be transported to the main glacier channel and be expelled from the glacier. In a one-dimensional model, sediment access occurs over the entire glacier bed for a given elevation band. Thus, the bedrock erosion or sediment production term (largely controlled by $f_{sl}$) represents this process, and increased sediment production results in greater connectivity.

The size and shape of the subglacial channels contributes to the discharge of sediment, as well. The sediment transport due to the velocity of subglacial water is limited by the channel width in smaller channels ($w_c$, Equation 7). For this reason, sediment exhaustion occurs mainly in localized channels, where channel widths are sufficiently large to allow substantial sediment transport (Figure 2). Conversely, sediment persists in patches of the glacier bed where velocity could be high, but insufficient channel-size effectively reduces sediment transport capacity. Experience with tuning the model to the available dataset shows

that increasing friction factor $f_i$ increases the area of the glacier bed over-which water with substantial velocity flows. Thus the model has trouble capturing inter-annual variability because sediment exhaustion does not occur over a substantial portion of the glacier bed.

Smaller, and probably more realistic, values of $f_{sl}$ represent the data in SUGSET compared to SUGSET_2D (0.36 and 4.09, respectively). Yet, because of the low dependence on $f_{sl}$, smaller values could perform well, but not be captured in the

relatively small number of model runs herein. However, because sediment production decreases with till-height (Equation 9), sediment production is limited to the narrow patches of the glacier bed where minimal till persists and bedrock erosion may occur. As a result, the model requires more sliding to produce the equivalent amount of sediment. At the same time, the limited spatial extent of glacier erosion and sediment transport points to a need, beyond this manuscript, to evaluate the precise location of bedrock erosion and the impact of subglacial till layers on bedrock erosion.

The best performing model run shows strong temporal variability in sediment discharge (Figure 10), with water discharges from the glacier above roughly $2~\mathrm{m^3\,s^{-1}}$ responsible for much of the sediment transport. Despite the strong dependence on grain-size and fluvial transport of sediment in the inversion, sediment transport capacity $Q_{sc}$ remains roughly an order of magnitude higher than sediment discharge ($Q_{sc}$). A steep section of the glacier experiences sediment depletion over the model run, as do several channels near the over-deepening and high on the glacier (Figure 11 c d). Several of these areas surrounding

the depleted areas show signs of deposition. On some parts of the upper glacier, bedrock erosion in the absence of substantial sediment transport is visible. With changing melt patterns or evolving glacier hydraulic gradients, this sediment could be mobilized and increase sediment discharge down glacier.

## 4   Implications

Results of both the one-dimensional model (SUGSET;  Delaney et al., 2019) and SUGSET_2D highlight the importance

parameterizing the spatial heterogeneities in bedrock erosion, sediment availability and sediment transport capacity. Yet, in the one-dimensional model, only the limits in the till-layer model (e.g. Equation 5c) and variations in sediment access along the glacier flow line impact sediment mobilization. In the two-dimensional model here, sediment access and transport is not averaged over the glacier width. Rather, by considering the spatial distribution in water discharge and sediment availability





laterally below a glacier, the model evaluates where heterogeneities may persist and their impact on subglacial sediment
dynamics (Figures 5, 7, and 11).

In the context of an ice-sheet-like glacier where moulin inputs localize water flow, this means that in a one-dimensional
model, the spatial area below the glacier over-which water from a moulin can transport sediment will be dramatically over-
estimated (Section 3.1). The model here fails to account for the seasonal evolution in water routing from a distributed to
channelized drainage system (e.g. Werder et al., 2013). Even so, the localization of water flow means that sediment transport
may often occur in a supply-limited regime and thus bedrock erosion or input of sediment into the channel figures strongly in
to sediment discharge (Figures 4 and 5). Should this model's characteristics hold true in the natural environment, then it may
well explain why a glacier's erosion potential might be a stronger control on sediment discharge from the Greenland Ice Sheet
compared to water discharge (Overeem et al., 2017).

Subglacial drainage below the Greenland Ice Sheet in response to climate warming remains highly uncertain, and the evo-
lution of subglacial channels largely depends on the locations where meltwater reaches the glacier bed (Poinar et al., 2015;
Gagliardini and Werder, 2018; Poinar et al., 2019). The experiments presented here suggest that rising temperatures and in-
creased glacier melt may result in greater sediment transport from ice sheets (e.g. Bendixen et al., 2017) by accelerating
sediment mobilization in regions of the bed where mobilization was reduced before. We note though, that the simulations
presented here did not account for changes in ice sheet thickness and velocity following glacier melt (e.g. Sundal et al., 2011;
Tedstone et al., 2015; Mouginot et al., 2019). Such changes will certainly impact bedrock erosion (e.g. Herman et al., 2021).
Additionally, changes to topography and relatively lower hydraulic gradients on the ice sheet may well reroute subglacial water
(Chu et al., 2016), and in the processes, new subglacial sediment sources might be accessed and reached. Given the prolonged
increase anticipated melt (e.g. Aschwanden et al., 2019), sediment transport and bedrock erosion are unlikely to trend toward
a new equilibrium, as they do after the model reaches a new climate (Figure 4).

The large diurnal and seasonal fluctuations in sediment transport at the synthetic alpine glacier result from diurnal and
seasonal variations in water routing and thus increased sediment availability because sediment transport only occurs over a
patch of bed for a short amount of time (Section 3.2). For instance here, diurnal fluctuations in sediment discharge in the
middle of the season can be over to 50%, which aligns more closely with some field observation of sediment discharge (e.g.
Swift et al., 2005; Delaney et al., 2018b) compared to the one-dimensional model (c.f. Delaney et al., 2019). Furthermore, the
results show that the location of bedrock erosion, processes in the till-layer and the timing of melt all play an important role in
the quantity of sediment discharge and the peak sediment discharge that is reached.

In the final test case, we compared model runs across a parameter space to sediment discharge data from Griesgletscher
in the Swiss Alps (Section 3.3). These results depended solely on sediment grain size compared to the initial till condition
or bedrock erosion (Figure 9). The model's strong dependence on grain-size in capturing data is caused by the transport of
sediment from patches of the glacier bed with slow water velocity or channel cross-sectional area to the larger, main channels.
This process could not be considered in a one-dimensional model, though it appears quite important at this relatively small
and shallow glacier. Yet, these results suggest that connectivity between the distal patches of the glacier bed and predominant
channels flowing below the glacier remain highly important for the quantity of sediment discharge. This process could be





through transport of small sediments as applied here or but could conceivably occur through other processes not considered
in the model, such as till deformation (e.g. Damsgaard et al., 2020). In these locations, slightly lower water velocities and
reduced channel width for sediment mobilization to occur limit the amount of sediment mobilization that occurs. Limited
bedload measurements suggest that suspended sediment, as opposed to bedload, comprises the majority of sediment leaving
glaciers (Riihimaki et al., 2005; Geilhausen et al., 2013; Delaney et al., 2018b). The model here does not differentiate between
bedload and suspended sediment transport, yet water access to sediment contributes substantially to the export of suspended
sediment, compared to bedload transport. Meltwater access to sediment is a process that SUGSET_2D represents, which is
also an evidently important process in sediment discharge from glaciers.

## 5    Conclusions

This manuscript presents a two-dimensional subglacial sediment transport model, SUGSET_2D, that evolves a till-layer in
response to subglacial sediment transport conditions. Model test cases utilize geometries and hydrological forcings from a
synthetic ice sheet, synthetic alpine glacier, and real glacier. The model captures sediment transport in supply- and transport-
limited regimes and conserves mass in the till-layer. Results from each test case point to the need to quantify the spatial distri-
bution of subglacial sediment and water when determining sediment discharge from the subglacial environment. Furthermore,
outputs align with some observed sediment dynamics.

Despite the model's capabilities, it contains a large number of poorly constrained parameters. Furthermore, to our knowledge,
the only one study quantified the thickness of till below a glacier, and this value was a single point measurement (Truffer et al.,
2000). As a result, in the absence of this knowledge, prescribing initial till-height conditions must be done in a thoughtful
manner that considers the hysteresis in the till layer stemming from bedrock erosion and fluvial sediment transport. Depending
on research questions, two-dimensional models may provide the most robust outcomes when operated across a parameter space
and in concert with available observations of erosion or sediment transport. Additionally, the method to route subglacial water
at glaciers with complex geometry must be carefully evaluated. Here, we route water and sediment based upon the Shreve
potential by using a spatially uniform flotation-fraction that evolves temporally (e.g. Section 3.3). Future work may consider
using a coupled model of channelized and distributed drainage networks (Hewitt et al., 2012; Werder et al., 2013). Such models
could even be run offline if the operator assumes, as we do, that rates of change in till height remain small compared to the
evolution in cross-section of the subglacial conduit.

Model outputs highlight that increased glacier melt do not necessarily result in proportional changes to sediment discharge if
new subglacial sediment sources are not accessed. Additionally, results demonstrate the role of spatially varying water routing
and lateral sediment connectivity in subglacial sediment discharge. In our opinion, among the most important topics regarding
glacier erosion lies in assessing the response time of glacial erosion and sediment transport to changes in glacier dynamics.
Hopefully modeling work, such as this, and observational studies will help to better understand the time-scales over-which
these respective processes occur and their response to climate and glacier dynamics.





*Code availability.* The code library along with illustrative examples are available at https://bitbucket.org/IanDelaney/sugset.jl/src/id-2d. The running and plotting scripts used in the cases herein are stored at https://bitbucket.org/IanDelaney/2d_runners/src/master/.

*Video supplement.* Videos of models application to Griesgletcher are available at https://bit.ly/3nPvVUI. The videos will be transfered to a permanent location pending acceptence.

*Author contributions.* ID designed the study, developed the model, ran the test cases and lead writing the manuscript. LA assisted with the writing the manuscript and provided key advice designing and troubleshooting the model. FH provided guidance with implementing and designing the model and preparing the manuscript.

*Competing interests.* The authors declare no competing interests.

*Acknowledgements.* We thank J. Braun, B. Bovy, F. De Doncker, S.N. Lane, G. Prasicek and M. Werder for fruitful discussions and insightful
comments. We are also grateful to Grégoire Mariéthoz and the Scientific Computing and Research Support Unit at Université de Lausanne for providing computing resources.





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



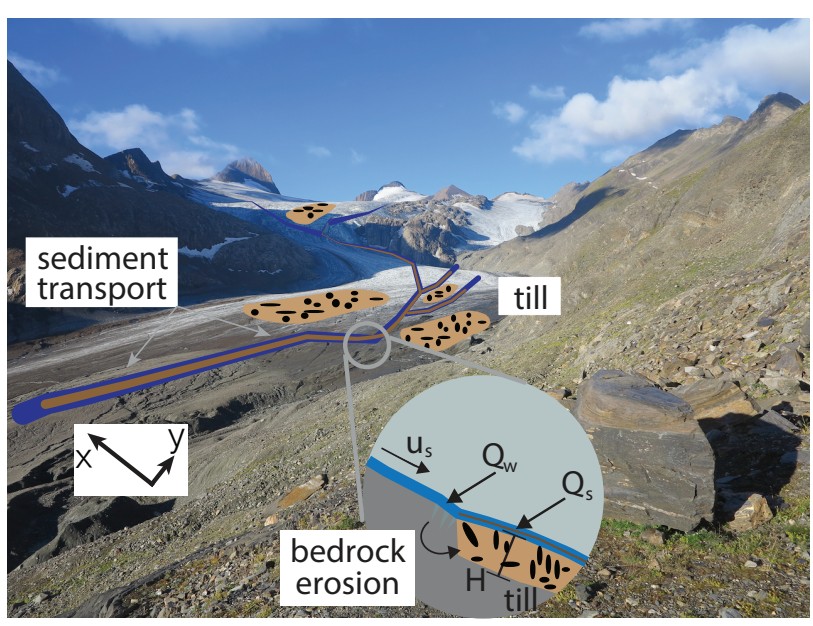

**Figure 1.** Cartoon of erosional and sediment transport processes considered in model below image of Griesgletscher in 2016. Bedrock erosion scales with sliding speed ($u_s$) and adds material to the till layer with thickness $H$, while water ($Q_w$) transports sediment ($Q_s$) fluvially, if sediment persists in that location of the glacier bed and fluvial transport conditions are sufficient.

Earth **Surface**
**Dynamics**
Discussions

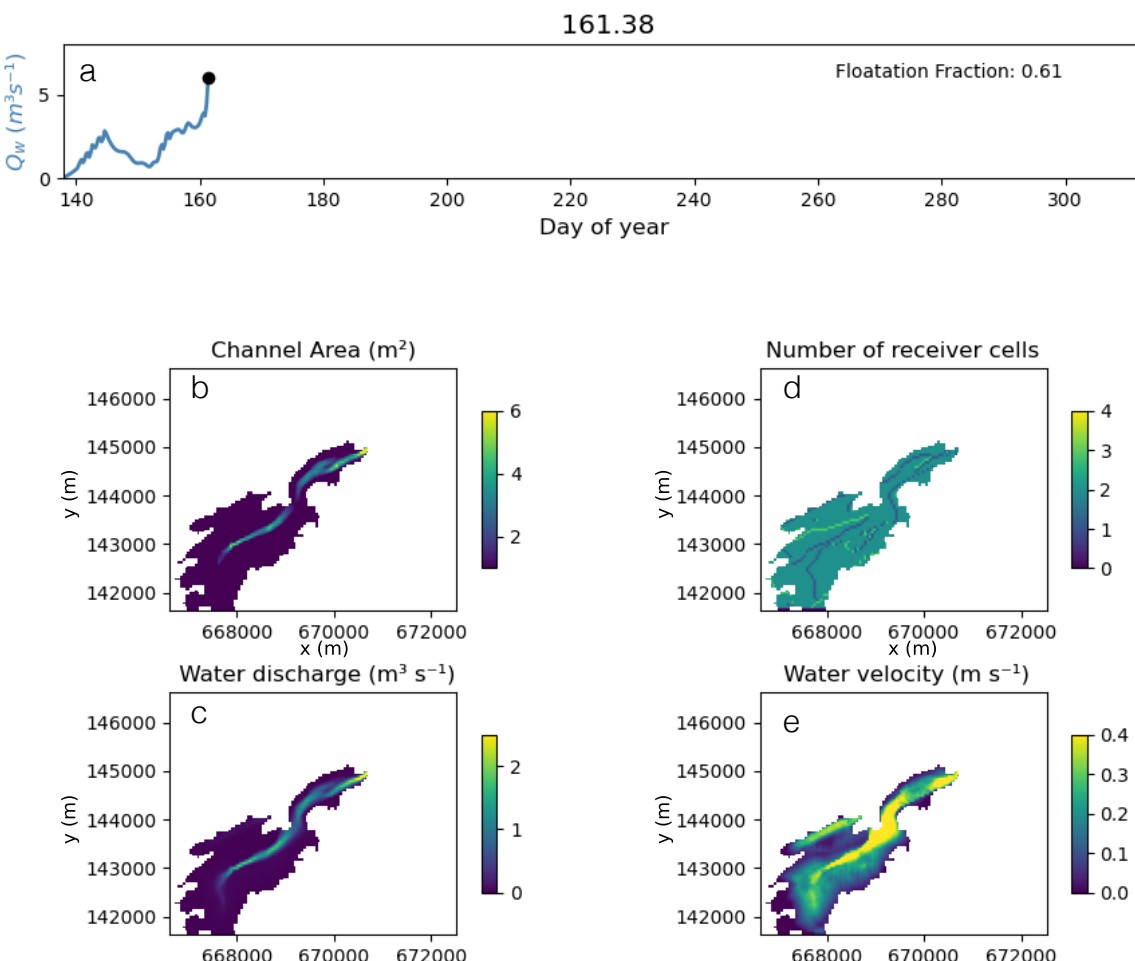

**Figure 2.** Example of model parameters and variables for the snap-shot of the Griesgletscher test-case Section 3.3. Water discharge from the catchment and glacier flotation fraction (a). Channel cross-sectional area $S$ (b) with distributed water discharge (c), the number of receivers cells, $r_t$ for a given cell (d), and the water velocity (e). Conditions b-d evolve with different hydrological conditions (e.g. a) over the glacier run. High water velocities persist at this time step due to rapid increase in water discharge (a).



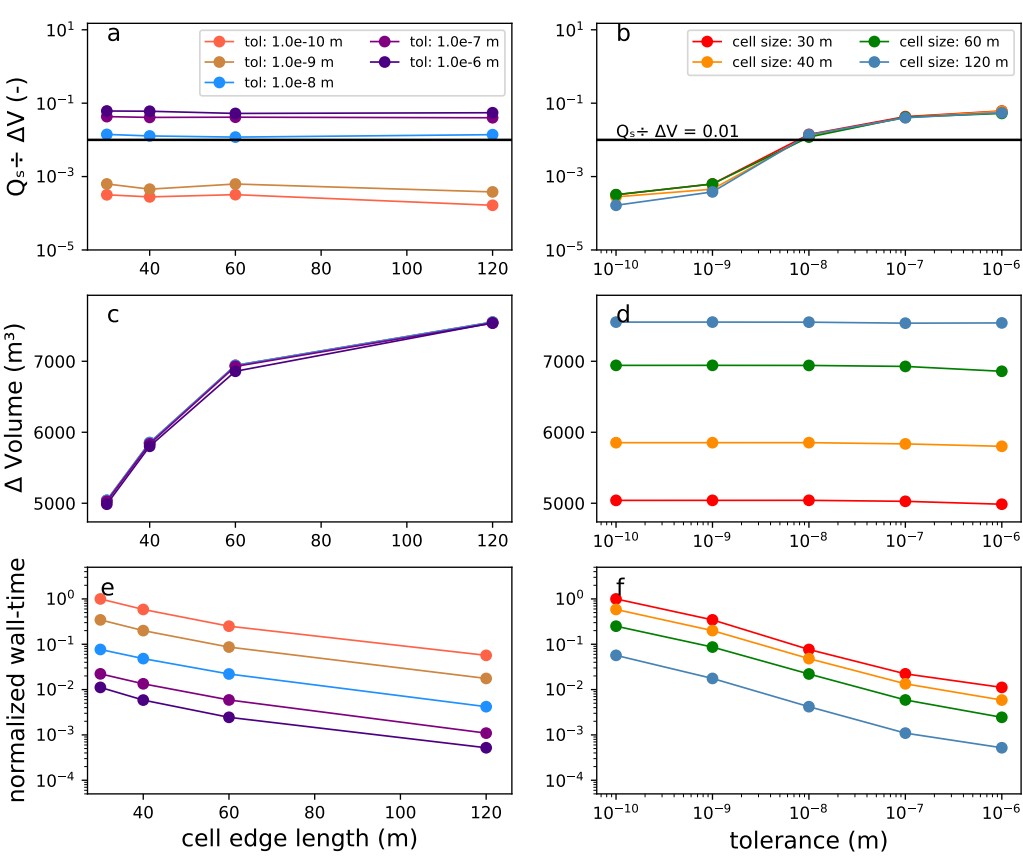

**Figure 3.** Mass conservation test (a, b), sediment export volumes (c, d) and wall-time on a single processor (e,f) for a variety of solving tolerances and cell sizes. Note that reltol and abstol are equal (Table 3). A diurnal forcing with no seasonal variation was applied to the synthetic alpine glacier case (Section 3.2).



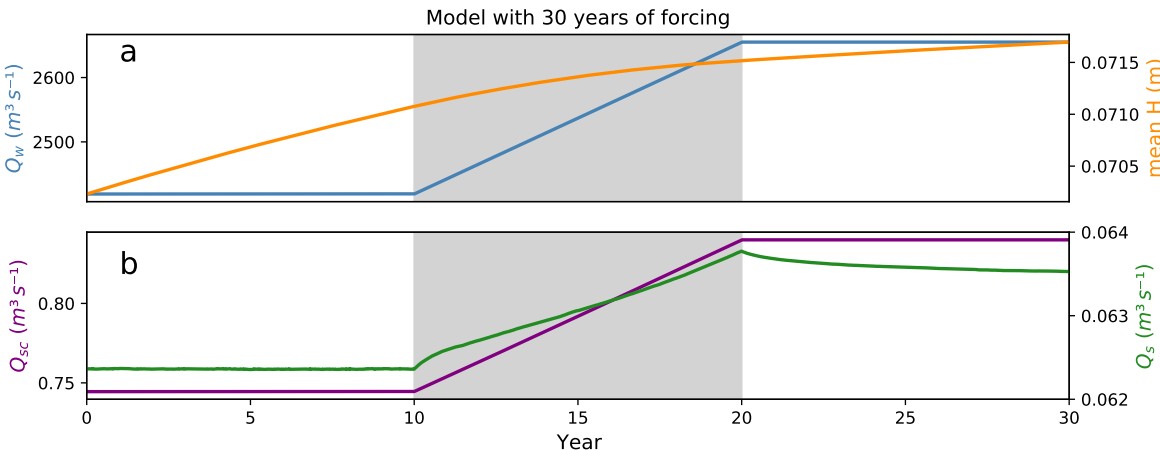

**Figure 4.** Model output from an ice sheet topography over a 20 year run with diurnal variations in melt to moulin inputs. Diurnal values averaged for clarity. Top panel a) represents water discharge ($Q_w$) and mean till-height ($H$). Lower panel b) shows sediment discharge capacity ($Q_{sc}$) and sediment discharge ($Q_s$).





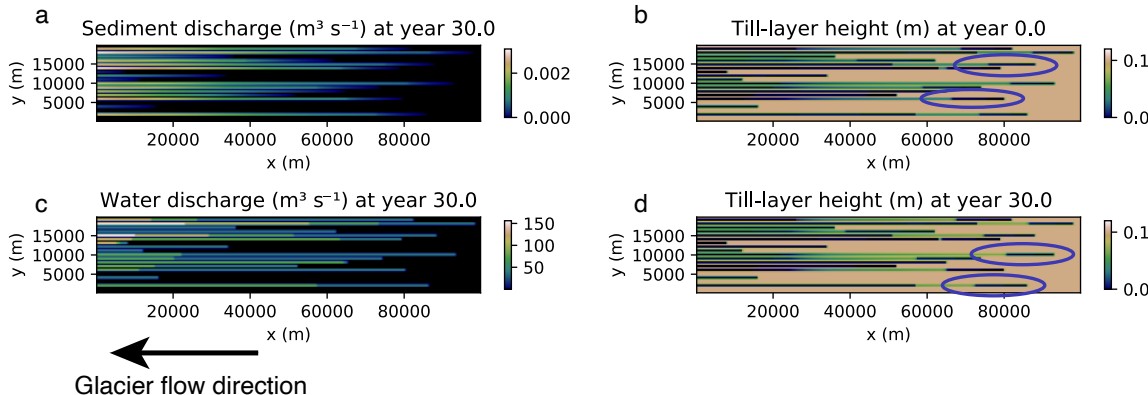

**Figure 5.** Spatial view of subglacial sediment transport (a), water discharge (c), till layer height prior to increased melt (b) and after increased melt (d). Blue ovals denote places of decreased uptake due to subglacial water from the moulin reaching sediment transport capacity.



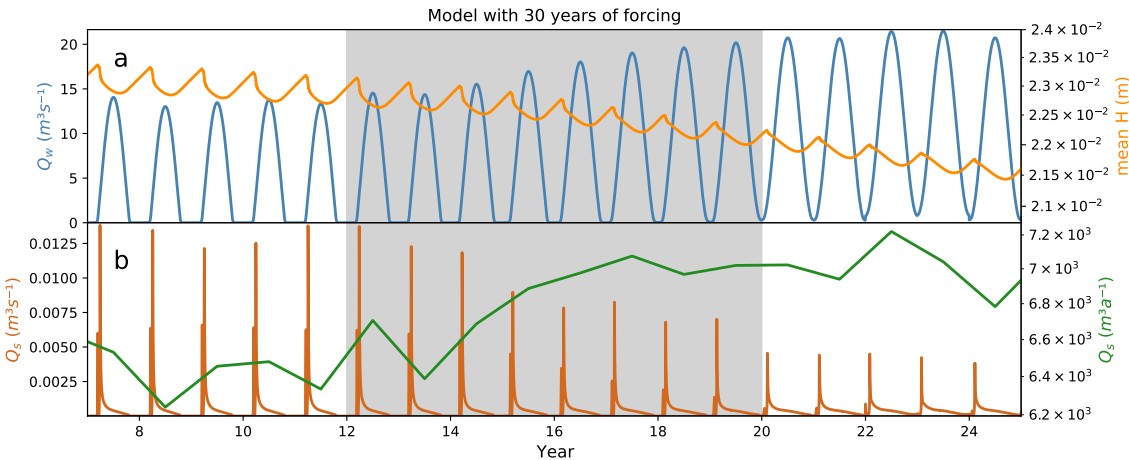

**Figure 6.** Model output from alpine topography and forcing over a 30 year run with diurnal and seasonal variations in melt input. Grey box represents time period of increasing glacier melt. a) Seasonally varying water discharge ($Q_w$) increases from year 12 to 20, while till height ($H$) decreases. b) Sediment discharge increases over this time period, with highest sediment discharge occurring in years 14–17 when increasing glacier melt can access new sediment sources high on the glacier. Following the increase in temperature, melt persists year around, so sediment accumulated during the winter months is no longer available, and thus the glacier does not experience periods of high sediment discharge, although annual sums might be higher.





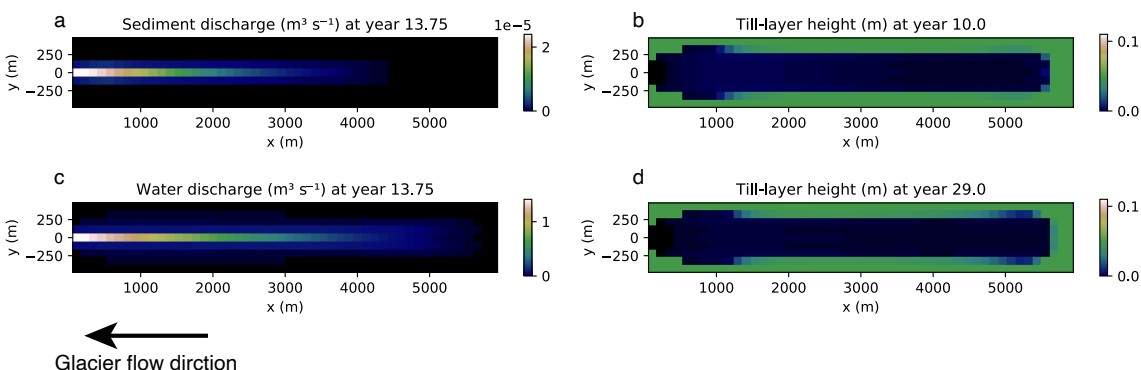

**Figure 7.** Spatial view of subglacial sediment transport (a), water discharge (c), till layer height prior to increased melt (b) and after increased melt (d). Spatial discontinuities in the distribution of water and sediment discharge in plots a) and c) result from the depletion of subglacial till beneath the glacier. Following the increase in melt, sediment transport increased so that it exceeded bedrock erosion. We have included an animation of this figure in the video supplement.



Earth **Surface**
**Dynamics**
Discussions

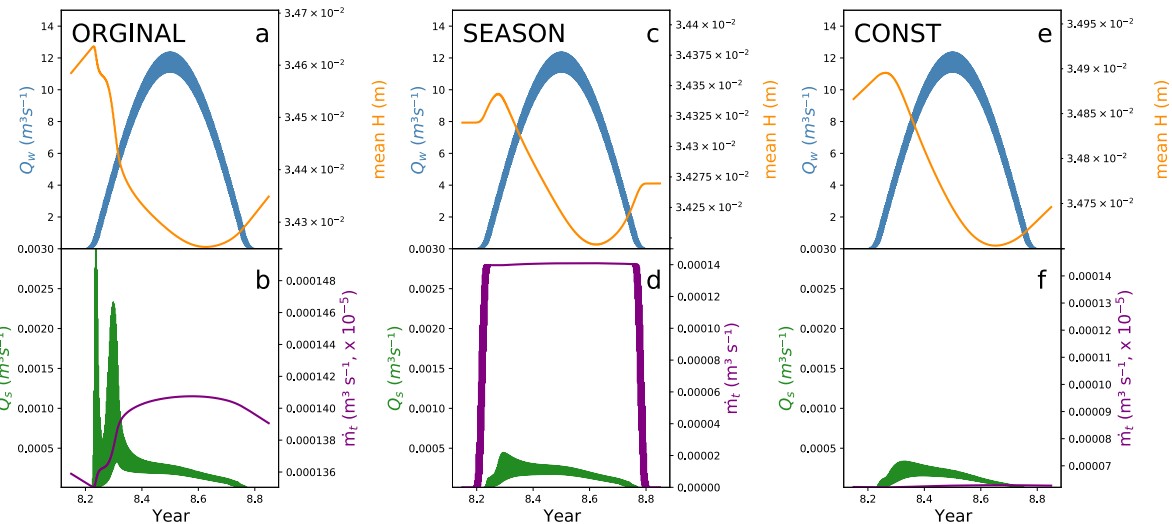

**Figure 8.** Annual response to different erosion patterns across the glacier, although diminishing bedrock erosion with respect to till-height is still in place given Equation 9. (a,b) Conventional model setup, where sediment is produced year around. This results in the peak amounts of sediment discharge occur in scenario 1 (a,b) where large amounts of sediment accumulated at the glacier terminus during the winter months. (c,d) Equivalent setup to previous, except sediment is only produced in summer months, when water is present at the glacier bed. Thus, till-height remains constant over the winter months. (e,f) Steady erosion of $1\,\mathrm{mm\,a^{-1}}$ across the entire glacier, with no spatial or temporal variability in sediment production. Yet, the different bedrock erosion scenarios each demonstrate increased sediment discharge at the onset of melt and subsequent exhaustion over the course of the season.



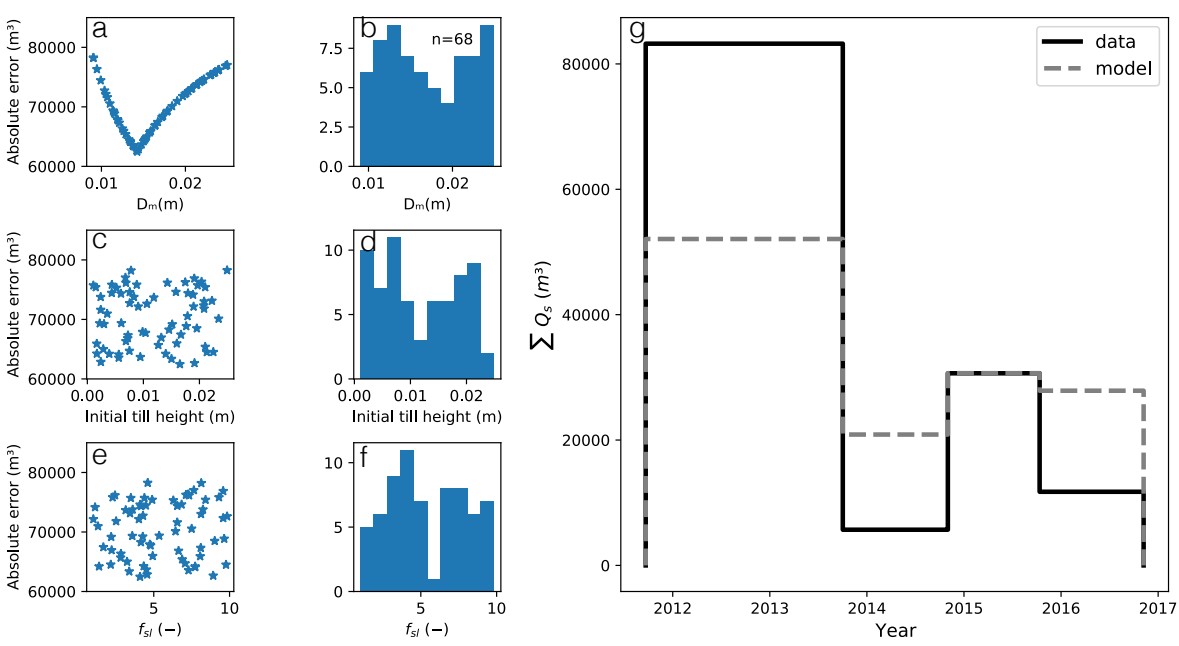

**Figure 9.** Results of the parameter search (column 1), the frequency of parameter values that produced a rank correlation of 1 (column 2) and the best fit model run amongst the parameter combinations (column 3). The model likely fails to adequately capture the 2012–2013 period probably due to processes not considered in the framework.



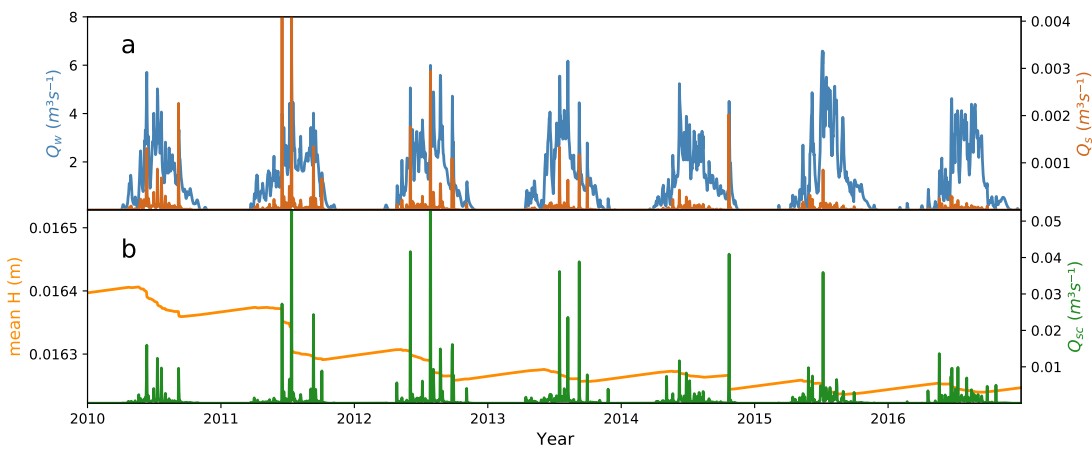

**Figure 10.** Sediment and water discharge from Griesgletscher (a) and sediment transport capacity and average till height (b).



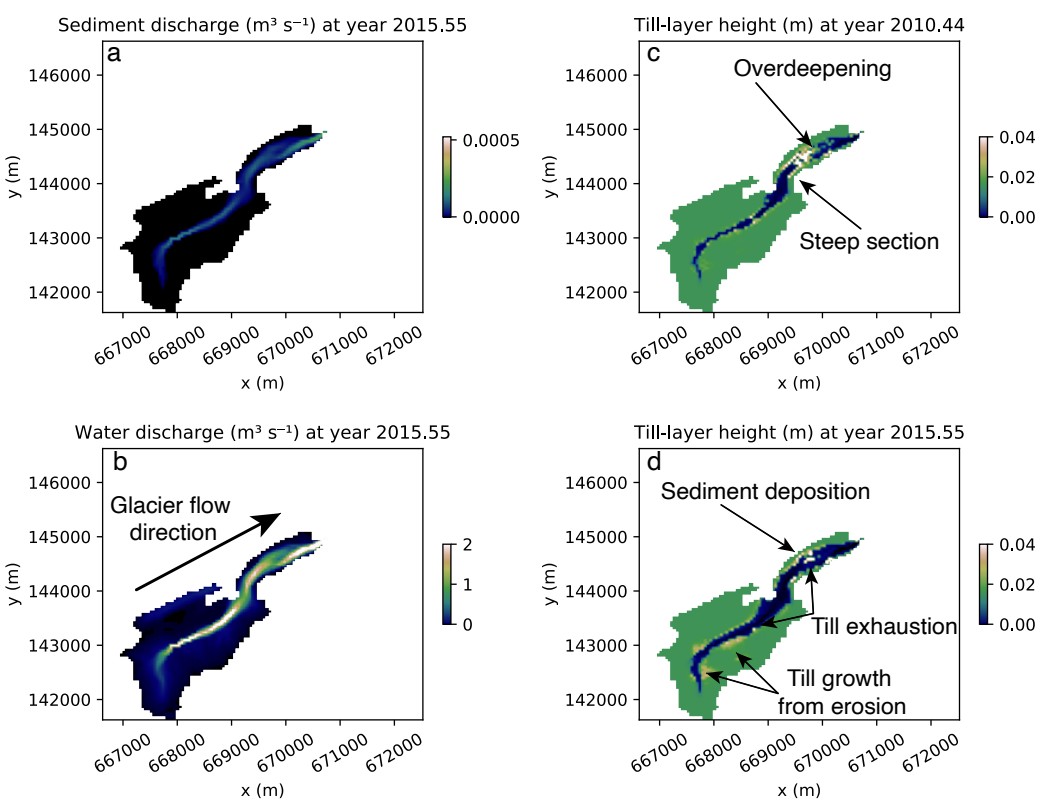

**Figure 11.** Spatial view of characteristics from the select Griesgletscher model run. Figure 1 shows images of this glacier. Subglacial sediment transport (a) and water discharge (c) are highly variable across the bed. Till layer height change substantially from the beginning of the model run (c) to after the model run (d). We point out the over-deepening near the glacier terminus as well as as a steep section connected the upper and lower glacier. Over this time till exhaustion in regions of high water flow are visible, while regions of sediment deposition and till growth from glacier erosion can be identified. We have included an animation of this figure in the video supplement.

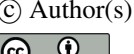



**Table 1.** Model variables

| Name | Symbol | Units |
|------|--------|-------|
| Horizontal (x,y) , vertical and time coordinates | $x, y, z, t$ | m, m, m, s |
| Surface and bed elevation | $z_s, z_b$ | m, m, m |
| Glacier surface slope | $\alpha$ | - |
| Channel hydraulic diameter | $D_h$ | m |
| Width of channel floor | $w_c$ | m |
| Channel cross-sectional area | $S$ | $\mathrm{m}^2$ |
| Water discharge (instantaneous) | $Q_w$ | $\mathrm{m}^3\,\mathrm{s}^{-1}$ |
| Water source term | $\dot{m}_w$ | $\mathrm{m}\ \mathrm{s}^{-1}$ |
| Representative water discharge | $Q_w^*$ | $\mathrm{m}^3\,\mathrm{s}^{-1}$ |
| Hydraulic potential | $\phi$ | Pa |
| Gradient of $\phi$ | $\Psi$ | $\mathrm{Pa}\ \mathrm{m}^{-1}$ |
| Flotation fraction | $f_f$ | - |
| Representative gradient of $\phi$ | $\Psi^*$ | $\mathrm{Pa}\ \mathrm{m}^{-1}$ |
| Water velocity | $v$ | $\mathrm{m}\,\mathrm{s}^{-1}$ |
| Water shear-stress | $\tau$ | Pa |
| Till source term | $\dot{m}_t$ | $\mathrm{m}\,\mathrm{s}^{-1}$ |
| Sediment discharge | $Q_s$ | $\mathrm{m}^3\,\mathrm{s}^{-1}$ |
| Sediment discharge capacity | $Q_{sc}$ | $\mathrm{m}^3\,\mathrm{s}^{-1}$ |
| Glacier sliding velocity | $u_b$ | $\mathrm{m}\,\mathrm{s}^{-1}$ |
| Erosion rate | $\dot{e}$ | $\mathrm{m}\,\mathrm{s}^{-1}$ |
| Till layer height | $H$ | m |
| Mass-balance rate at terminus | $\dot{b}^0$ | $\mathrm{m}\,\mathrm{s}^{-1}$ |



**Table 2.** Physical model parameters and constants

| Name | Symbol | Value | Units |
|---|---|---|---|
| Darcy-Weisbach friction factor | $f_r$ | 15 | - |
| Hooke angle of channel | $\beta$ | 22.5 | ° |
| Source quantile | $s_q$ | 0.75 (Gries: .2) | - |
| Source average time | | 2.5 (Alpine: 1.5; Gries: 0.5) | d |
| Sediment-uptake $e$-folding length | $l$ | 100 (Ice sheet: 500 ) | m |
| Sediment grain mean diameter | $D_{m_{50}}$ | $5 \times 10^{-4}$ (Gries: 0.014) | m |
| Till height limit | $H_{lim}$ | 0.10 | m |
| Till height erosion limit | $H_g$ | 0.05 | m |
| Gravitational constant | $g$ | 9.81 | $\mathrm{m\,s^{-2}}$ |
| Density of water | $\rho_w$ | 1000 | $\mathrm{kg\,m^{-3}}$ |
| Density of ice | $\rho_i$ | 900 | $\mathrm{kg\,m^{-3}}$ |
| Density of bedrock | $\rho_b$ | 2650 | $\mathrm{kg\,m^{-3}}$ |
| Bulk density of sediment | $\rho_s$ | 1500 | $\mathrm{kg\,m^{-3}}$ |
| Erosional exponent | $l_{er}$ | $2.02^{\,a}$ | - |
| Erosional constant | $k_g$ | $2.7^{-7\,a}$ | $\mathrm{m^{1-l_{er}}\ s^{l_{er}-1}}$ |
| Seconds per year | $s_{year}$ | $3.1536^7$ | s |
| Seconds per day | $s_{day}$ | $86,400$ | s |
| Glen's $n$ | $n$ | 3 | - |
| Ice flow rate factor | $A$ | $2.4 \times 10^{-24}$ | $\mathrm{s\ Pa^{-3}}$ |
| Mass-balance gradient | $\gamma$ | 0.00625 | $\mathrm{a^{-1}}$ |
| Basal melt rate | $\dot{m}_b$ | $7.3 \times 10^{-11}$ | $\mathrm{m\,s^{-1}}$ |
| Glacier sliding fraction | $f_{sl}$ | 1 (Gries: 4.09) | - |





**Table 3.** Numerical model parameters

| Name | Symbol | Value | Units |
|---|---|---:|---|
| Solver tolerance (relative) | reltol | $10 \times 10^{-8}$ | - |
| Solver tolerance (absolute) | abstol | $10 \times 10^{-8}$ | m |
| Maximum timestep | dtmax | 21600 (6) | s (hr) |
| Minimum timestep | dtmin | 1 | s |
| Edge length $(x)$ | $ds$ | | m |
| Edge length $(y)$ | $dh$ | | m |
| Cell area | $\delta$ | | $m^2$ |
| Sediment connectivity factor | $\Delta\sigma$ | $10^{-3}$ | m |
| Minimum hydraulic diameter | $D_{h\,min}$ | 0.3 | m |
| Number of cells | $n_n$ | - | - |
| Stack | $s_t$ | $\overrightarrow{n_n}$ | - |
| Receivers | $r_s$ | $4 \times n_n$ | - |
| Number of receivers per cell | $n_r$ | $\overrightarrow{n_n}$ | - |
| Donors | $d_n$ | $4 \times n_n$ | - |
| Number of donors per cell | $n_d$ | $\overrightarrow{n_n}$ | - |
| Weight of each receiver | $w_r$ | $4 \times n_n$ | - |