# Peer review of "The spatially distributed nature of subglacial sediment dynamics: using a numerical model to quantify sediment transport and bedrock erosion across a glacier bed in response to glacier behavior and hydrology"

_Earth Surface Dynamics, 2021_

## Referee Comment (RC3)

**The spatially distributed nature of subglacial sediment dynamics: using a numerical model to quantify sediment transport and bedrock erosion across a glacier bed in response to glacier behavior and hydrology**

I. Delaney, L. S. Anderson and F. Herman

esurf-2021-88

*Referee's report*

Well, the title is a mouthful. How about 'A numerical model of subglacial sediment transport'? It would shorten the paper considerably.

The paper describes a model for subglacial sediment transport, implemented as a numerical code. For reasons which will emerge below, I think this paper should be rejected. However, I suspect that this is a rather unfashionable view, and certainly there are many examples of cellular computational models which have been published, notably in the field of hillslope evolution (Willgoose, Tucker, etc., etc.). Whether such an approach is justified in the present instance may be a matter of opinion. From my perspective, however, there is nothing I can usefully learn from this paper.

Already in describing water flow, the basic physics is shelved. A Röthlisberger-type theory for channel flow involves an evolution equation for cross-sectional area, and thus the hydraulic radius; this is avoided here by parameterising the hydraulic radius. But actually, I think it is worse than this. Eventually, the model is applied to sediment transport beneath both glaciers and ice sheets. We know that there are R channels under glaciers, but then most of the erosion is elsewhere; how is it thought the sediment gets into the channel? Eventually we get bedload or suspended load, but these concepts are for rivers. Although this is a model, it does not seem to be one which engages with physical principle.

I think the Exner equation (4) is muddled. The relaxation length $l$ should not be there. The Exner equation is just $H_t + \boldsymbol{\nabla}.\mathbf{Q} = m$, and then $\mathbf{Q}$ has to be prescribed. Commonly one just takes $\mathbf{Q} = \mathbf{Q}_b(\tau)$, but if one wants to include the relaxation length, then one can take (in one dimension) $lQ_x = Q_b - Q$, as is commonly done in modelling dune formation (e. g., Kroy *et al.* 2002, equation (6)). In two dimensions, you would need a bit of differential geometry. The basal stress is a vector $\boldsymbol{\tau}$, and if $\mathbf{T} = \dfrac{\boldsymbol{\tau}}{|\boldsymbol{\tau}|}$ is the tangent unit vector along a (water) flow line, then you would have $\mathbf{Q} = Q\mathbf{T}$, and $\mathbf{T}.\boldsymbol{\nabla}Q = Q_b - Q$, I suppose. Equations (5) just look silly.

I'm very surprised to see the exponent 5/2 in equation (7). Most of these bedload transport laws have 3/2. I don't have the Engelund-Hansen report, but in his 1970 JFM paper, he uses Meyer-Peter/Müller (and doesn't reference this report). Ah, I see reading on (140) that there is a reason for this, as it supposedly includes suspended sediment. Of course, a proper treatment of suspended sediment then requires an evolution equation for the suspended sediment concentration. It seems to me that if you go to the trouble to include bed erosion to the bedload layer, then it is logically commensurate to include suspended load concentration, at least in some fashion. But

again, we must be thinking of streams, and then such streams do not cover the glacier bed; how should sediment transport to the streams be modelled? The statement "the continuous nature of the relationship improves the model stability" is poor. First you pose the model, then you deal with trying to solve it. You don't decide what is in the model on the basis of what you can solve (or you shouldn't).

At equation (11), I begin to wonder what is the point of this exercise. Yes, what happens at the bed is complicated. But to choose the sliding velocity to be a fixed fraction of the shearing velocity is simply making things up. Particularly, sliding depends on the subglacial hydrology through its dependence on the effective pressure. One might argue that the emphasis here is on sediment transport, but that fundamentally depends on water flow (and also actually till deformation), and I see little point in trying to deal with one at the expense of the others, at least if the results of the model aim to be realistic.

As we come to the numerical implementation, I belatedly realise that the point of all the simplifications to the physics is that it allows a cellular model to be constructed. It reminds me a bit of the paper by Barchyn *et al.* (2016). I am not a big fan of this approach, which seems to me to be motivated by the wish to produce pretty pictures at the expense of doing science. A model is only as good as its formulation, and I find the modelling here to be weak in a number of points.

This point is perhaps illustrated by the comment at 371, the 'model successfully captured the inter-annual variability in sediment discharge from the Griesgletscher'. But a parameter search was used to find parameter values which worked. So can we conclude that the model is a good one? No. Does it then have predictive value? No. And, most importantly, should we expect it to be a good representation of physical process? In view of my comments above, I would have to say no.

Smaller points:

At equation 1. This looks a bit odd to me. In my way of thinking, for force balance in a channel, you would have $\tau l = \rho g S A$, where $\tau$ is the stress, $l$ the wetted perimeter, $S$ a suitable slope and $A$ the cross-sectional area. So $\Psi = \rho g S = \dfrac{\tau l}{A}$, and if $\tau = f \rho u^2$ and $Q = A u$, then $\Psi = \dfrac{f \rho Q^2 l}{A^3}$. You can get equation (1) if $l \propto A^{1/2}$, as for a circular or triangular cross section; but it seems to me that equation (1) is not the basic law. For example, a wide stream has $l \propto A$, approximately.

Line 105. This is unclear. $D_h$ is a length, not an area.

115. and $\rightarrow$ or, presumably.

181. The wording here suggests that a partial differential equation is being solved, but if I understand this correctly, this is not the case. (4) with (5) form a set of ordinary differential equations.

**Reference**

Barchyn, T. E., T. P. F. Dowling, C. R. Stokes and C. H. Hugenholtz 2016 Subglacial bed form morphology controlled by ice speed and sediment thickness, Geophys. Res. Lett. **43** (14), 7,572-7,580.

Engelund, F. 1970 Instability of erodible beds. J. Fluid Mech. **42** (2), 225-244.

Kroy, K., G. Sauermann and H. J. Herrmann 2002 Minimal model of sand dunes. Phys. Rev. Letts. **88** (5), 054301.

---

## Author Comment (AC1)

We greatly appreciate the constructive and thoughtful review, and largely agree with the recommendations made. Following each of the comments, we provide our amendment to the manuscript or the justification in deviating from the comment, in normal font.

Thank you for your time and efforts to comment on our work.

Best regards,

Ian Delaney, on behalf of all authors.

- **For the synthetic ice sheet case, this study finds that sediment discharge in the model run decreases after the climate warms and reaches a stable regime. The decrease occurs due to sediment exhaustion from increased water discharge, and associated sediment transport, which removed till that was unable to be transported in a cooler climate with less available meltwater able to transport sediment.**

  **I find this model behavior problematic to generalize: it would be an interesting finding, depletion of sediment flux, yet the outcome is entirely controlled by the model assumptions on initial till height and bedrock sediment production. So, it is hard to draw any conclusions on this process being of relevance?**

  **I think the paper would be strengthened by omitting this case study, it oversimplifies the ice sheet system by too much.**

  **Results on the effect of spatially-distributed fluxes seem of more of importance, and this effect is more pronounced in the alpine glacier case. The alpine glacier case recreates high sediment concentrations in early season, and his is explained by spatial variability in the till distribution and sediment transport access to these sediment patches. This effect has been observed in proglacial streams.**

  We appreciate these comments and can understand why this section could be cumbersome. After consideration, we agree with Dr. Overeem in the conclusion that the paper will be enhanced by omitting this section.

- **It seems that there is a large process parametrization discrepancy between the general bedrock sliding law applied (with no grainsize dependence and validated over glacial-deglacial scales) and the heavily grainsize dependent sediment transport law (Engelund-Hansen). A note of caution should probably already be added into the results section, and refer to the later discussion section on this topic.**

  This is a very good point and one of the assumptions of the sliding-erosion relationship is the presence of basal debris to provide a means of abrading the bedrock (Hallet, 1979). In the introduction, we note that this contributes to sediment transport. Additionally, comments regarding the role of debris concentration have been added to the new **Model Limitations** Section.

**On the structure of the paper:**

- **Title: I recommend simplifying the title.**

  **Suggestions: "Bedrock erosion and sediment transport variations across a glacier bed controlled by glacier behavior and hydrology" or "Modeling of the spatially distributed nature of subglacial sediment erosion and transport dynamics"?**

  We agree with this comment and find the current title a bit long and convoluted. We have changed it to *Modeling of the spatially distributed nature of subglacial sediment erosion and transport dynamics.*

- **Introduction: can be tightened. Please read carefully through and omit some of the sections that are repetitive or wander.**

The introduction has been shortened and some content has been omitted from the current draft.

- **Model description: I do appreciate the review of the hydraulic model, even if it previously been well described in Delaney, 2019, but is needed here to have this paper be a stand-alone contribution.**

We have updated this section to include a complete description of the hydraulic model. The primary addition contains the explanation of the shape factors, such $s$, that impact the shape of the subglacial channel and channel width, $w_c$, which is required for sediment transport relationships.

The section now reads:

SUGSET_2D requires a hydraulics model as a means to route sediment and water through the subglacial environment and to evaluate the sediment transport capacity of this water, based upon the hydraulic gradient, channel size and water flux (Table 1, Section 2; e.g. Walder and Fowler, 1994; Alley et al., 1997). The hydraulic model here is based on the premise that subglacial water flows along the hydraulic potential gradient and glacier ice pressurizes water at its bed (Shreve, 1972). We aim simulate key characteristics of an Röthlisberger-channel without explicitly describing properties such as creep closure and pressure melt of channel walls.

The hydraulic gradient for a section of channel at a location below the glacier at a certain time $\Psi$ can be determined with a known hydraulic radius $D_h$ and water discharge $Q_w$ of that channel, given the Darcy-Weissbach equation for fluid flow through a pipe

$$\Psi = s\, f_r\, \rho_w\, \frac{Q_w^2}{D_h^5}. \tag{1}$$

the density of water is given by $\rho_w$, $f_r$ represents the Darcy-Weisbach friction factor, and $s$ accounts for channel geometry Hooke et al. (1990)

$$s = \frac{2\,(\beta - \sin\beta)^2}{(\frac{\beta}{2} + \sin\frac{\beta}{2})^4}, \tag{2}$$

where $\beta$ is the central angle of the circular segment representing the channel edge. Smaller values of $\beta$ result in broad channels and $\beta = \pi$ corresponds to a semicircular channel. The channel width $w_c$ is given by

$$w_c = 2\sin\frac{\beta}{2}\sqrt{\frac{2\,S}{\beta - \sin\beta}}, \tag{3}$$

where $S$ is the cross-sectional area of the channel given by

$$S = \frac{D_h^2}{2}\,\frac{\left(\frac{\beta}{2} + \sin\frac{\beta}{2}\right)^2}{\beta - \sin\beta}. \tag{4}$$

To approximate hydraulic diameter $D_h$, we assign a representative water discharge $Q_w^*$ to $Q_w$, by taking a quantile of water discharge over a certain prior time period (days), over which the subglacial conduit responds to hydraulic conditions (source quantile; c.f. de Fleurian et al., 2018; Delaney et al., 2019; Nanni et al., 2020). The source quantile is assumed to be the water discharge over which the hydraulic radius responds to changes in water discharge. For instance, we would not expect a short-lived increase in water discharge due to a short precipitation event to greatly impact the hydraulic radius, where as the hydraulic radius would respond to a prolonged increase in water discharge from the increased melt.

This time represents the time period, assumed to be days, over-which hydraulic radius $D_h$ evolves in response to $Q_w$ (e.g. Nanni et al., 2020). $Q_w$ and $Q_w$ comprise the amount of melt produce upglacier, such that no storage exists in the model.

We then evaluate $D_h$, the hydraulic radius given

$$D_h = \left(s\, f_r\, \rho_w\, \frac{Q_w^{*\,2}}{\Psi^*}\right)^{\frac{1}{5}}. \tag{5}$$

$\Psi^*$ is a representative hydraulic gradient at overburden pressure, evaluated using the Shreve potential gradient

$$\Psi^* = \nabla(\rho_i \, g \, (z_s - z_b) + \rho_w \, g \, z_b), \tag{6}$$

where $z_s$ and $z_b$ are surface and bed elevations, respectively, $\rho_i$ is the density of ice and $g$ is the gravitational acceleration constant.

With knowledge of $D_h$, we insert the instantaneous value of $Q_w$ into Equation 1 to evaluate the instantaneous hydraulic gradient $\Psi$. We note that to prevent unreasonable water pressures when $Q_w^*$ rapidly increases and $D_h$ is small, the model limits the minimal cross-sectional area $S$ to $0.5\,\mathrm{m}^2$.

- **I found the order in Section 2.2 non-intuitive. Indeed, it is important to detail the Exner equation approach first, but then begin with bedrock erosion parametrization, as that is the term that produces sediment, and then describe sediment transport.**

We appreciate the issue with the ordering of the equation. However, we choose to present sediment transport first, as this depends largely on the hydraulics, which are presented directly above. However, we have modified the Equation so that it's ordering now aligns with the text.

*The model simulates the evolution of a subglacial till height, which we define as transportable sediment below the glacier due to glacier erosion and fluvial sediment transport. Fluvial sediment transport in supply- and transport-limited regimes mobilize and deposit sediment, adding or removing material from the till layer (Brinkerhoff et al., 2017; Delaney et al., 2019). Conversely, erosive processes such as abrasion and quarrying add material to the layer. To quantify these processes, we implement the Exner Equation (Figure 2; Exner, 1920a,b; Paola and Voller, 2005), a mass conservation relationship, to solve for the till layer height given the erosive and fluvial conditions.*

$$\underbrace{\frac{\partial H}{\partial t}}_{\text{till evolution}} = - \underbrace{\nabla \cdot Q_s}_{\text{sediment transport}} + \underbrace{\dot{m}_t}_{\text{bedrock erosion}} \tag{7}$$

*$H$ is till thickness and $t$ is time (Table 1). The first term represents fluvial sediment transport processes, where $\nabla \cdot Q_s$ represents sediment mobilization in either supply- or transport-limited regimes. The second term captures bedrock erosion processes, where $\dot{m}_t$ is a bedrock erosion rate.*

- **One suggestion to make each of the source terms and in-out fluxes clear is to add a diagram of a mass fluxes in the Exner equation, and label the processes. Like Fig 1 in Paola and Voller, but then made specific for this special implementation.**

We appreciate this comment and agree that such a figure would make interpretation of the relevant processes easier. In turn, we have made the following figure.

[Figure]

Figure 1: Illustration of terms in Equation 7, detailing the layers of bedrock, till, water and ice.

Line-by-line minor comments:

- **line 6: the concept of 'sediment connectivity' needs explanation or definition before being invoked as an important control. Either explain within the abstract, or omit.**
  The text now reads:

  *However, sediment discharge depends heavily on sediment connectivity, the movement of sediment between its detachment in source areas and its deposition in sinks. In turn, the geoscience community needs modeling frameworks that describe subglacial sediment discharge in two spatial dimensions over time.*

- **Line 14: this sentence is unclear."We find that sediment grainsize plays an important role. Smaller sediment sizes....**

  The text now reads:

  *Additionally, the model's capacity to represent the data depends greatly on the sediment grain size parameter in the model, impacting the sediment transport capacity of the subglacial water.*

- **Line 24: possibly add this paper by Dongfeng Li 2021, although the increasing sediment loads are not just attributed to glacier melt but also due to permafrost thaw and rainfall change. Li, D., Lu, X, Overeem, I., Walling, D., Syvitski, J., Kettner, A.J., Bookhagen B., Zhou, Y., Zhang, T., 2021). Exceptional increases in fluvial sediment fluxes in a warmer and wetter High Mountain Asia. Science, 10.1126/science.abi9649**

  Many thanks for the recommendation. Citation fits well here and is added.

- **Line 30: replace with: 'are limiting nutrients in the oceanic ecosystem'**
  Done.

- **Line 45: add flow after ... water**
  Done.

- **Line 51: although if there is little sediment embedded there is little abrasion!**

  This is an important point and the text now reads:

  *Bedrock erosion and fluvial sediment transport vary depending on the timescales and characteristics of individual glaciers. For instance, erosive processes likely dominate over sediment transport processes on glaciers with minimal sediment storage, large concentrations of subglacial debris and steep gradients (Hallet, 1979; Humphrey and Raymond, 1994; Herman et al., 2015; Ugelvig and Egholm, 2018; Herman et al., 2021).*

- **Line 90: please clarify this approach: is there a single hydraulic diameter and single associated water discharge for the entire distributed drainage system? Or dothese properties Q\* and Dh get assigned/calculated for individual drainage channels?**

  We appreciate this comment and understand the need to clarify this matter.

  They text at line 90 now reads: *The hydraulic gradient for a section of channel at a location below the glacier at a certain time $\Psi$ can be determined with a known hydraulic radius $D_h$ and water discharge $Q_w$ of that channel, given the Darcy-Weissbach equation for fluid flow through a pipe. . .*

- **Line 94: this selection of representative discharge seems really difficult and perhaps arbitrary? how do you decide ahead of time what quantile (or did you mean quartile?).**

  Selection of a representative discharge does require the assumption of time periods of the response of channel's size. Both numerical modeling (e.g. de Fleurian et al., 2018) and observations of subglacial hydrology (Nanni et al., 2020) suggest that this time period is on the order of days. With regard to selecting a representative quantile, we select this value with the assumption that certain water discharges influence the size of the conduit.

  The text now reads:

  *To approximate hydraulic diameter $D_h$, we assign a representative water discharge $Q_w^*$ to $Q_w$, by taking a quantile of water discharge over a certain prior time period (days), over which the subglacial conduit responds to hydraulic conditions (source quantile; c.f. de Fleurian et al., 2018; Delaney et al., 2019; Nanni et al., 2020). The source quantile is assumed to be the water discharge over which the hydraulic radius responds to changes in water discharge. For instance, we would not expect a short-lived increase in water discharge due to a short precipitation event to greatly impact the hydraulic radius, where as the hydraulic radius would respond to a prolonged increase in water discharge from the increased melt.*

  We have also discussed the limitations and implications of our subglacial hydrology parameterization in the Section **Model limitations**.

- **Line 103: please spell out R-channel at its first occurrence.**

  Done. This sentence is also moved to the top of this Section.

- **Line 124: I am not sure about the limit, H-lim, at 10 cm, this seems really arbitrary. Sediment transport in a pressurized pipe flow can probably easily scour bedforms to 4-5 times that depth almost instantaneously? I understand that perhaps there is little data to constrain this parameter, but it may be prudent to explore whether the model is sensitive to this setting.**

  We have discussed this parameter to the **Model Limitations** section. While bedforms can be scoured to 4-5 times that depth, we suggest that this may only happen at a very specific event, where water is routed to a new patch of the glacier bed, or if sediment is thick during the model initialization. Over the time periods that we examine in this model, we believe

that it is not as imperative to capture these peak events. Rather, we choose this small value in order to reduce the spin-up period and response time of the model, as larger values of $H_{lim}$ would mean that longer time periods are needed for the till to accumulate and then be transported from the glacier in regions such as over deepenings.

- **line 165: this switch in describing code is a bit out of style, perhaps better to describe this in the code documentation as opposed to command lines in the paper.**

  This line has been omitted.

- **line 177 Reference is oddly formatted, please make conform journal requirements**

  This line has been omitted as well.

- **line 184 First explain why this would be a non-stiff problem, and then state that this solver is apprporiate for non-stiff problems.**

  The text now reads:

  *In turn, to discretize the problem in time, the model implements the VCABM solver (Hairer et al., 1992; Radhakrishnan and Hindmarsh, 1993) from the package DifferentialEquations.jl (Rackauckas and Nie, 2017) to evolve till layer height $H$. This solver implements an adaptive time step and uses a linear multistep method (Adams-Moulton) that is well-suited for non-stiff problems, which is optimal here due to the rapid fluctuations in sediment transport that can occur in response to variable hydraulic conditions.*

- **Line 202-208 this section has several 'disclaimers' on your assumptions that would be better suited for the discussion of model limitations later in the paper.**

  We have removed the topics from the here and moved them to a new section titled **Model limitations**.

- **Line 245: I am not intimately familiar with SHMIP, but does the rate of temperature offset originate from those setups? Do I understand it right that the total warming scenario is an added 15 degrees C to diurnal amplitude? Over 30 years? That warming rate seems really abrupt, and unprecedented even under the most catastrophic warming scenarios?**

  This warming is abrupt and rapid. The intent is not to mimic real warming scenarios, but rather to demonstrate the behavior of the model as climate warms. The results suggest that additional meltwater accessing sediment below a greater portion of the glacier bed will result in great amounts of sediment transport. We also note that while the glacier morphology and hydrology parameterizations are from SHMIP, SHIMP does not implement any warming in its hydrological forcing.

- **Line 255: line 255 replace till height by thickness?**

  We appreciate this comment. However, we have used till height throughout the paper and think it more appropriate.

- **Line 372. Repeat for careless readers, what are the last three time spans?**

  The text now reads:

  *The model captures the last three measured yearly sums of sediment discharge well. . .*

- **Line 374. Isn't one of the important changes of this approach that you do have a way to access different patches of the glacier bed? Could it be model underestimation of till thickness and erosion instead?**

  While the mismatch between the data and the model could result from underestimated till thickness, we find this unlikely given the weak dependence of the outputs on the model's

initial condition or the production of sediment. Other runs in the ensemble experienced greater bedrock erosion rates and till heights as initial conditions. However, these model runs overestimated the sediment discharge for the last three years of the study. As a result, we believe that the mismatch is due to processes not considered in the model, rather than a limited exploration of the parameter space.

**Figures**

- **Figure 2: This figure is really helpful in providing a feel for the dimensions of the subglacial drainage network , flow velocity magnitude (small), and spatial distribution of discharge.**

  Thank you.

- **Figure 3. Recommend this figure to go to an appendix. It is not too well explained in the text and seems important for a manual of the model, not for the scientific findings.**

  This figure has been removed from the text.

- **Figure 5: Perhaps improve this figure by changing the aspect ratio of these figures, at present they are hard to read. May help to add 2 initial panels with icesheet topography map and ice flow velocity? And then have the 'maps' of the basal conditions.**

  Inline with the recommendations by the reviewer to remove the ice sheet case, this figure has also been removed.

- **Figure 7**

  - **May help to add 2 initial panels with icesheet topography map and ice flow velocity? And then have the 'maps' of the basal conditions.**

    We value this comment and have added two panels with ice thickness and bedrock erosion rate. We included the bedrock erosion rate given that it is a function of basal sliding velocity, so the figure also yields information about glacier sliding.

  - **Perhaps improve this figure by changing the aspect ratio of these figures, at present they are hard to read.**

    We have only adjusted the aspect ratio with respect to the additional two panels for the comment above. However, the individual panels are constrained in order to keep the x and y axes scaled equally.

  - **Great to see this as animations.** Thank you.

- **Figure 9. Remove 'likely ' from the caption.**

  Done.

- **Figure 10. Add to the figure caption what parameters are modeled vs observed.**

  The text now reads:

[revised manuscript text omitted]

---

## Author Comment (AC2)

This review focuses on the implementation description and presentation of the model. The reviewer identifies points of confusion in the description of the model. We take these comments seriously and have addressed them below.

The reviewer's comments are in bold, while our response is in normal font, and excerpts from the manuscript are in italics.

Best regards,

Ian Delaney, on behalf of all authors.

- **(1) Sect. 2.2**

  **The main part where I am not immediately convinced that is is correct is Sect. 2.2 about sediment transport, starting from the balance (Exner) equation (Eq. 4). In its genuine form, such an equation would be written in 2D in terms of a sediment flux per unit width (m$^2$/s) instead of the sediment flux (m$^3$/s). In the 1D version (Delaney et al., JGR, 2019, Eq. 9), the sediment flux is used in combination with a channel width w. However, the version introduced here uses a length scale l that describes a "characteristic length-scale for sediment mobilization, over which sediment mobilization adjusts to sediment transport conditions." This length scale is a longitudinal length scale (in flow direction), while the formulation of the balance equation in terms of the flux requires a length scale perpendicular to the flow direction (such as the channel width). So a am quite sure that Eq. 4 is not correct in the form it is written, but I cannot assess whether it is correct in the implementation and whether it affects the results in case it is not correct.**

  Many thanks for this comment and given the fundamental nature of this equation, any misunderstandings must be clarified. We have altered the equation so that now it reads:

  *The model simulates the evolution of a subglacial till height, which we define as transportable sediment below the glacier due to glacier erosion and fluvial sediment transport. Fluvial sediment transport in supply- and transport-limited regimes mobilize and deposit sediment, adding or removing material from the till layer (Brinkerhoff et al., 2017; Delaney et al., 2019). Conversely, erosive processes such as abrasion and quarrying add material to the layer. To quantify these processes, we implement the Exner Equation (Figure 2; Exner, 1920a,b; Paola and Voller, 2005), a mass conservation relationship, to solve for the till layer height given the erosive and fluvial conditions.*

  $$\frac{\partial H}{\partial t}_{\text{till evolution}} = - \underbrace{\nabla \cdot Q_s}_{\text{sediment transport}} + \underbrace{\dot{m}_t}_{\text{bedrock erosion}} \tag{1}$$

  *$H$ is till thickness and $t$ is time (Table 1). The first term represents fluvial sediment transport processes, where $\nabla \cdot Q_s$ represents sediment mobilization in either supply- or transport-limited regimes. The second term captures bedrock erosion processes, where $\dot{m}_t$ is a bedrock erosion rate.*

  **There also seems to be a problem with the physical dimensions in Eq. 5 (beyond that the divergence of the sediment flux at the left-hand size should not be called "sediment discharge"). If $\dot{m}_t$ is indeed an erosion rate (m/s) as defined earlier, it is not consistent with the other properties, which are fluxes per length (m$^2$ s$^{-1}$).**

  We thank the reviewer for the careful examination of this text and the accompanying equation as several inconsistencies and mistakes were identified. The relevant equation and text now read:

*Divergence of the sediment flux is calculated by approximating $\nabla \cdot Q_s$ with $\frac{\widetilde{\nabla \cdot Q_s}}{w}$ and using the mobilization scheme from Delaney et al. (2019)*

$$\frac{\widetilde{\nabla \cdot Q_s}}{w} = \begin{cases} \dfrac{Q_{sc} - Q_s}{l} & \text{if } \frac{Q_{sc}-Q_s}{l} \leq \dot{m}_t w & \text{(2a)} \\[2ex] 0 & \text{if } H = H_{lim} \quad \& \quad \frac{Q_{sc}-Q_s}{l} \leq 0 & \text{(2b)} \\[2ex] \dfrac{Q_{sc} - Q_s}{l}\,\sigma(H) + \dot{m}_t w\,(1 - \sigma(H)) & \text{otherwise} & \text{(2c)} \end{cases}$$

*$w$ is the width of a patch of glacier bed, prependicular to the direction of water flow. $Q_{sc}$ is sediment transport capacity, or the amount of sediment that could be transported given hydraulic conditions. $l$ is a characteristic length-scale for sediment mobilization, over which sediment mobilization adjusts to sediment transport conditions. $\sigma$ is a sigmoidal function of $H$*

$$\sigma(H) = \left(1 + \exp\left(\frac{2 - \Delta\sigma H}{5}\right)\right)^{-1}, \tag{3}$$

*that enables smooth transition over the range: $H = 2\Delta\sigma^{-1} \pm \Delta\sigma^{-1}$ in Equation 2c. $\Delta\sigma$ is a value below which $\sigma$ substantially deviates from $1$, and reduces sediment mobilization.*

- **(2) Sect. 2.3**

  **While I am confident that the numerical implementation is sound, I am not happy about the way it is described in this section. To be honest, I found it even more confusing than enlightening. The cited work by Bovy et al. (2016) used a standard finite-volume discretization of the fluxes, while the algorithm proposed by Braun and Willett (2013) used a single-flow-direction (D8) scheme (if I am not wrong). I read that you use a multi-flow-direction scheme, but it is not clear what the difference toward typical continuum schemes (finite volume) is or whether it is the same. Instead of (or in addition to) mentioning functions and software packages, I would ask you to state the equations that are finally solved.**

  - **(a) How exactly is the flux (water/sediment) distributed among the available directions? It looks as if this changes trough time, but I did not get it completely.**
    The text in **Section 2.3.1 Routing algorithm and implementation** now reads as:
    *Sediment and water are routed down the hydraulic gradient using a multi-cell routing scheme (Quinn et al., 1991), implemented in a similar way as Bovy et al. (2016), but on a regular grid. Sediments and water move from one cell to another using a steepest-descent algorithm, based upon the hydraulic potential. This routing scheme returns a stack, from which contains information about the order of cells to perform the calculations. The model evaluates the hydraulic potential at every time step, first, the flotation fraction for a cell at a given time is calculated by $f_f = \frac{\phi_o}{\phi^*}$, where hydraulic potential $\phi_o$ comes from*

    $$\phi_o = \sum_{j=1}^{n_r} \phi_{0_j} \cdot w_{r_j} + \Psi_j \cdot \delta^{\frac{1}{2}} \cdot w_{r_j}. \tag{4}$$

    *Here, where $\Psi_n$ comes from Equation 1, $\delta$ is the area of a cell on a regular grid yielding cell length, $n_r$ is the number of receivers that the cell has, and $w_r$ is the proportion of water flowing from one cell to the other. The operation executed on a cell by cell basis using the routing scheme above, beginning at the glacier terminus and moving up glacier.*

*We distribute the mean value of $f_f$ across the glacier, then implement the routing scheme for the hydraulic potential determined from the Shreve potential as*

$$\phi = f_f \, \rho_i \, g \, (z_s - z_b) + \rho_w \, g \, z_b. \tag{5}$$

*The node ordering algorithm is executed every time step in response to diurnal variations in water pressure and thus variable routing of subglacial water in response to changing hydraulic conditions (e.g. Iken and Bindschadler, 1986; Chu et al., 2016). However, to improve stability during periods of rapidly changing sediment transport conditions, we reorder the stack, based upon the hydraulic conditions to the nearest 6 mn. Smaller solving tolerances increase the computational time due to 1) increased accuracy of the solution and 2) the reassessment of flow fractions between the adjacent cells, which results in different routing configurations as the model converges. We fill closed basins or over-deepenings in the hydraulic potential to maintain continuous sediment transport through the domain. The model uses an external algorithm, that contains routes flow and fills basins based upon rasterised values of the hydraulic potential.*

*Using this routing scheme, we are able to evaluate the water discharge in a cell from melt upstream as*

$$Q_w = \dot{m}_w \cdot \delta \, + \, \sum_{j=1}^{n_d} Q_{w_j} \cdot w_{r_j}, \tag{6}$$

*where $\dot{m}_w$ is a melt water source term and $n_d$ is the number of donors of that cell. The sediment discharge $Q_s$ into a cell is like-wise computed as*

$$Q_s = \sum_{j=1}^{n_d} Q_s \cdot w_{r_j}. \tag{7}$$

*$Q_s$ is then used to evaluated the $\nabla \cdot Q_s$ given Equation 2 and subsequently, the change in till height using Equation 1. In both Equation 6 and 6, the operations are conducted given the node ordering information in the stack such that the flux in to a cell depends on the flux through the catchment above it.*

– **(b) What does "integrating Equation 1" (line 166) exactly mean here? Typically, "integrating" a differential equation is somehow one-dimensional or upstream in a tree, like the algorithm made popular by Braun and Willett (2013). However, if there are multiple flow directions, upstream paths may meet again, so that you would end up at different values of phi by integrating over different paths. I see that you are solving a system of equations, but I cannot see how.** The text now reads:

*We implement a regular grid for discretization. Spatial discretization must be substantially smaller than characteristic length-scale, $l$, in Equation 2. We then solve Equation 1 for till height $H$ for given initial and boundary conditions in response to till production $\dot{m}_t$ and divergence of the sediment discharge $Q_s$ using an explicit time integration scheme.*

We have omitted the citation Braun and Willett (2013) from the text, as this could be taken as misleading given the method we use.

– **(c) It would be good to state the balance equation for the water and for the sediment for each grid cell explicitly and in such a way that it becomes clear which system of equations finally has to be solved.** Dr Overeem, the other reviewer, brought up a similar point. A figure of the mass balance of sediment and water for a cell has been added to the text. Additionally, as shown above, we have included equations for the mass balance of sediment in and out of the cells.

- **(d) Also about the discretization: I guess you are assuming one single conduit of a given hydraulic radius for each grid cell. This may be questionable, but of course not necessarily bad. A similar assumption is typically made in karst evolution models. It should be discussed at least briefly. And is this conduit directed, or will it change its direction if phi changes?**

  Indeed, the model operates by assuming a single conduit with a given hydraulic radius for each cell. Because the hydraulic potential in each cell and time step may be evaluated, given the methods described in **Section 2.3.1 Routing algorithm and implementation**. By evaluating the hydraulic gradients in $x$ and $y$ directions, the water leaving the cell will be directed accordingly.

- **(3) Lines 104-105: $Q\cdot_w$ or $Q_w$? And I did not understand how the water pressures can increase to unreasonable values if the flux rapidly increases. I can imagine that this happens if D$_h$ is small for a single$-$flow$-$direction model, but why do the alternative flow directions not help here?**

  This happens in the model if D$_h$, and thus cross-section $S$, is small and water flux rapidly increases. While the 2D routing scheme does accommodate high water pressures, alternative flow directions does not necessarily accommodate the changes if D$_h$ remains small in the surrounding locations as well.

- **(4) Line 137: Here you use "capital S" (the cross section area), which is definitely different flow "lowercase s" used in Eq. 1 (probably a nondimensional factor).**

  A more thorough explanation of the hydraulics model has been added, including a complete explanation of the relationship between hydraulic diameter $D_h$ and channel cross-section $S$. The text is included in the response to Dr. Overeem's review.

- **(5) Line 144: As far as I know, the lower bound l$_{er}$ = 2/3 mentioned here was obtained from a worldwide comparison of glaciers under different conditions. So I am not sure whether this low value is relevant for Eq. 10.**

  We understand the reviewer's concern with the low values of l$_{er}$, however, we choose to test the model against the range of values available in the literature.

- **(6) Lines 144-149: Some other studies use the relation from Eq. 11 for the deformation velocity and a similar relation with an exponent n-1 instead of n+1 for the sliding velocity $u_b$. This relation predicts that deformation becomes more and more relevant if the thickness of the ice layer $z_s - z_b$ increases. Using this relation in my own work, I was even told by reviewers that the sliding velocity cannot be predicted from the deformation velocity at all. I do not share this point of view, but I think it should be discussed why even a weaker assumption than the relation typically used is employed here.**

  We thank the reviewer for this comment. Given the uncertainties in glacier sliding behavior, the relationship between glacier dynamics, sliding and erosion is yet more complex. The sliding relationship has been modified to a relationship between driving stress and sliding velocity, given the Weertman sliding relationship. The text and model have been changed in the following relationship with glacier sliding is used.

  *$u_b$ is assumed to be relates to basal shear stress ($\tau_b$ Weertman, 1957) given the following relationship*

  $$u_b = B_s \tau_b^m, \tag{8}$$

  *$B_s$ is a constant and we assume the exponent $m$ is equal to 1. We assume that $\tau_b$ is equal to driving stress (Cuffey and Paterson, 2010)*

  $$\tau_b = \rho_i \, g \, h \, (\sin \alpha), \tag{9}$$

*where $\rho_i$ is the density of ice, $h$ is the glacier thickness and $\alpha$ is the surface slope of the glacier.*

- **(7) Line 239: Why is there a need to increase DeltaT, and what is the value of DeltaT used here?**

  This test case has been removed following comments in Dr. Overeem's review, and the comment no longer applies. However, the large value of $\Delta$ T owed to the fact that a large value of $\Delta$ T was needed so that an adequate amount of water would flow through the moulins, given the location of the moulin within a single grid cell, as opposed to distributed across the glacier surface.

- **(8) Line 246: I did not get the point why $H_{max}$ grows.**

  As in the previous comment, this test case has been removed and the comment no longer applies.

- **(9) Line 285: What is the meaning of the $-5$ in Eq. 16, given that DeltaT is an (adjustable) temperature offset?**

  The $-5$ is included to be consistent with Equation 16 in de Fleurian et al. (2018), where the original glacier set up and forcing is presented. While this value can be subtracted from $\Delta$ T, we believe it is best to leave the factor in place so that readers may compare model characteristics and values with those in de Fleurian et al. (2018).

- **(10) Lines 339-342: Nice numbers, but you somehow let the chance to analyze your model more thoroughly pass by. From Eqs. 7 and 8, we see that the transport capacity $Q_{sc}$ is proportional to $v^5$ and thus also to $Q_w^5$. this is what you put into your model. So you could integrate $Q_w^5$ over, say 1 year periods, and look how well your sediment output correlates to this integral. If it correlated perfectly, then your sediment output was just what you put into your model equations, and everything else would be unimportant. I think it will not correlate perfectly, so that you can discuss which of the components of your model is important. However, this is just an idea how you could sharpen the discussion.**

  We thank the reviewer for this recommendation. The following text is included:

  *Lastly, we note that the model demonstrates the limits of using hydrological relationships to evaluate sediment discharge from glaciers. Equivalent values of water discharge inputs result in model outputs of sediment discharge that varies over orders of magnitude (Figure 10, a). Sediment transport capacity could be tuned through hydraulic parameters and sediment size to improve its performance. Yet, sediment discharge is substantially lower than the sediment transport capacity substantially (Figures 4 and 8). Additionally, model outputs show that sediment discharge consistently varies over an order of magnitude for a given sediment transport capacity in the test cases (Figure 10, b). In turn, using solely the water discharge or sediment transport capacity (e.g. Equation 10) fails to consider the changes to sediment availability caused by sediment transport, especially when changes to sediment storage can take place over seasons to decades.*

  Along with the following figure:

**References**

Bovy, B., Braun, J., and Demoulin, A.: A new numerical framework for simulating the control of weather and climate on the evolution of soil-mantled hillslopes, Geomorphology, 263, 99 − 112, doi:https://doi.org/10.1016/j.geomorph.2016.03.016, 2016.

[Figure]

Figure 1: Model outputs of sediment discharge from the glacier compared to water discharge (a) and sediment transport capacity (b).

Braun, J. and Willett, S.: A very efficient O (n), implicit and parallel method to solve the stream power equation governing fluvial incision and landscape evolution, Geomorphology, 180, 170–179, doi:10.1016/j.geomorph.2012.10.008, 2013.

Brinkerhoff, D., Truffer, M., and Aschwanden, A.: Sediment transport drives tidewater glacier periodicity, Nature Communications, 8, 90, doi:10.1038/s41467-017-00095-5, 2017.

Chu, W., Creyts, T. T., and Bell, R. E.: Rerouting of subglacial water flow between neighboring glaciers in West Greenland, Journal of Geophysical Research: Earth Surface, 121, 925–938, doi:10.1002/2015JF003705, 2016.

Cuffey, K. M. and Paterson, W. S. B.: The Physics of Glaciers, Butterworth-Heinemann, Burlington, MA, USA, Fourth edn., 2010.

de Fleurian, B., Werder, M. A., Beyer, S., Brinkerhoff, D., Delaney, I., Dow, C., Downs, J., Hoffman, M., Hooke, R., Seguinot, J., and Sommers, A.: SHMIP The Subglacial Hydrology Model Intercomparison Project, Journal of Glaciology, 64, 897–916, doi:10.1017/jog.2018.78, 2018.

Delaney, I., Werder, M., and Farinotti, D.: A Numerical Model for Fluvial Transport of Subglacial Sediment, Journal of Geophysical Research: Earth Surface, 124, 2197–2223, doi:10.1029/2019JF005004, 2019.

Exner, F. M.: Über die Wechselwirkung zwischen Wasser und Geschiebe in flüssen, Abhandlungen der Akadamie der Wissenschaften, Wien, 134, 165–204, 1920a.

Exner, F. M.: Zur Physik der Dünen, Abhandlungen der Akadamie der Wissenschaften, Wien, 129, 929–952, 1920b.

Iken, A. and Bindschadler, R. A.: Combined measurements of subglacial water pressure and surface velocity of Findelengletscher, Switzerland: conclusions about drainage system and sliding mechanism, Journal of Glaciology, 32, 101–119, 1986.

Paola, C. and Voller, V. R.: A generalized Exner equation for sediment mass balance, Journal of Geophysical Research: Earth Surface, 110, doi:10.1029/2004JF000274, URL https://agupubs.onlinelibrary.wiley.com/doi/abs/10.1029/2004JF000274, 2005.

Quinn, P., Beven, K., Chevallier, P., and Planchon, O.: The prediction of hillslope flow paths for distributed hydrological modelling using digital terrain models, Hydrological processes, 5, 59–79, 1991.

Weertman, J.: On the sliding of glaciers, Journal of Glaciology, 3, 33–38, 1957.

---

## Author Comment (AC3)

We thank the reviewer for the critical assessment of this work. While we disagree with many of the perspectives presented below, we also have considered them thoroughly. Broadly, it is our opinion that the reviewer's comments point to a need for additional explanation of our work. We address these items below. We have placed the reviewer's comments in bold, our commentary is in normal font, and the new text is in italics.

Best regards,

Ian Delaney on behalf of all authors

- **Well, the title is a mouthful. How about 'A numerical model of subglacial sediment transport'? It would shorten the paper considerably.**

  This is a perspective shared by another reviewer, using other words. We have adjusted the title to read: *Modeling of the spatially distributed nature of subglacial sediment transport and erosion*.

- **The paper describes a model for subglacial sediment transport, implemented as a numerical code. For reasons which will emerge below, I think this paper should be rejected. However, I suspect that this is a rather unfashionable view, and certainly there are many examples of cellular computational models which have been published, notably in the field of hillslope evolution (Willgoose, Tucker, etc., etc.). Whether such an approach is justified in the present instance may be a matter of opinion. From my perspective, however, there is nothing I can usefully learn from this paper. Already in describing water flow, the basic physics is shelved. A Röthlisberger- type theory for channel flow involves an evolution equation for cross-sectional area, and thus the hydraulic radius; this is avoided here by parameterising the hydraulic radius.**

  This comment points to shortcomings in the model description that Dr. Overeem and Dr. Hergarten addressed as well. For the sake of manuscript length, we chose to leave the complete description of the hydraulics model to the previous publication (Delaney et al., 2019), where it was originally published. Upon receiving the feedback here along with other reviewers we have decided to include the complete description of the hydraulics model in the manuscript. This has been added to the response to Dr. Overeem's comments. In this material, we describe how the hydraulic radius responds to hydraulic conditions below the glacier and is converted in to cross sectional area.

- **But actually, I think it is worse than this. Eventually, the model is applied to sediment transport beneath both glaciers and ice sheets. We know that there are R channels under glaciers, but then most of the erosion is elsewhere; how is it thought the sediment gets into the channel?**

  Given the spatial resolution of this model, we assume that the amount of sediment available in a cell is available for transport, with the uptake of sediment limited by the till height $H$. This is described in Equation 6.

- **Eventually we get bedload or suspended load, but these concepts are for rivers. Although this is a model, it does not seem to be one which engages with physical principle. I think the Exner equation (4) is muddled. The relaxation length l should not be there. The Exner equation is just Ht + $\triangle \cdot$Q = m, and then Q has to be prescribed. Commonly one just takes Q = Qb($\tau$), but if one wants to include the relaxation length, then one can take (in one dimension) lQx = Qb Q, as is commonly done in modelling dune formation (e. g., Kroy et al. 2002, equation (6)). In two dimensions, you would need a bit of differential geometry. The basal stress is a vector $\tau$, and if T = $\tau$ is the tangent unit vector along a (water) flow line, then you would have $\tau$ Q = QT, and T.$\triangle$Q = Qb Q, I suppose. Equations (5) just look silly.**

We appreciate the comments regarding the Exner equation and Dr. Hergarten also brought these issues to light. We have now adjusted the Exner equation so that it is consistent, and the exact text is in the response to Dr. Hergarten's review. The adjusted equation and accompanying are also included in that response.

The reviewer suggests that Q = Qb($\tau$) results in sediment discharge. While this generally true in transport-limited situations, in supply-limited cases, sediment transport depends also on till height $H$. When till height is small, sediment is not available for transport, and sediment transport exists in a supply-limited regime. As our equation 5 (now 8) describes, sediment transport is reduced when $H$ is small, to account for supply-limited transport. The dependence on till height $H$ means that we must evolve the Exner equation (equation 4, now 7) in response to bedrock erosion and sediment transport.

- **I'm very surprised to see the exponent 5/2 in equation (7). Most of these bedload transport laws have 3/2. I don't have the Engelund-Hansen report, but in his 1970 JFM paper, he uses Meyer-Peter/Müller (and doesn't reference this report). Ah, I see reading on (140) that there is a reason for this, as it supposedly includes suspended sediment. Of course, a proper treatment of suspended sediment then requires an evolution equation for the suspended sediment concentration. It seems to me that if you go to the trouble to include bed erosion to the bedload layer, then it is logically commensurate to include suspended load concentration, at least in some fashion. But again, we must be thinking of streams, and then such streams do not cover the glacier bed; how should sediment transport to the streams be modelled? The statement "the continuous nature of the relationship improves the model stability" is poor. First you pose the model, then you deal with trying to solve it. You don't decide what is in the model on the basis of what you can solve (or you shouldn't).**

We have chosen in this version of the model to include suspended sediment and bedload transport together for simplicity and because the point of the paper is to demonstrate the importance of sediment transport in two spatial dimensions. We do not aim to parse the relative roles of suspended sediment and bedload transport, especially because the fraction of sediment created by bedrock erosion that is bedload or suspended adds more assumptions to the model. The statement regarding the continuous nature of the relationship, upon the reviewer's recommendation, it has been omitted from the manuscript.

- **At equation (11), I begin to wonder what is the point of this exercise. Yes, what happens at the bed is complicated. But to choose the sliding velocity to be a fixed fraction of the shearing velocity is simply making things up. Particularly, sliding depends on the subglacial hydrology through its dependence on the effective pressure. One might argue that the emphasis here is on sediment transport, but that fundamentally depends on water flow (and also actually till deformation), and I see little point in trying to deal with one at the expense of the others, at least if the results of the model aim to be realistic.**

As discussed in the response to Dr. Hergarten, we have exchanged the fixed sliding fraction with a Weertman sliding parameterization based upon the driving stress, which has very similar erosional patterns and results compared to the fixed fraction of deformation. While sliding does depend on effective pressure, over the time periods needed to create substantial sediment at the bed for transport and on land terminating glaciers, Weertman sliding is largely accepted as being able to represent sliding velocities (e.g. Gimbert et al., 2021).

- **As we come to the numerical implementation, I belatedly realise that the point of all the simplifications to the physics is that it allows a cellular model to be constructed. It reminds me a bit of the paper by Barchyn et al. (2016). I am not a big fan of this**

**approach, which seems to me to be motivated by the wish to produce pretty pictures at the expense of doing science. A model is only as good as its formulation, and I find the modelling here to be weak in a number of points.**

We appreciate the critical review of our work. At the same time, the model accomplishes describing both bedrock erosion and sediment transport below glaciers and implements a routing scheme to transport this sediment across the glacier's bed in two spatial dimensions.

- **This point is perhaps illustrated by the comment at 371, the 'model successfully captured the inter-annual variability in sediment discharge from the Griesgletscher'. But a parameter search was used to find parameter values which worked.**

Our intent in applying the model to Griesgletscher is to 1) demonstrate its applicability to real scenarios, and 2) to understand potential processes that drive subglacial sediment transport. In Section 3, we describe how the model fulfills these objectives.

Indeed most models need some kind of tuning in order to be compared to data. Furthermore, by systematically calibrating the model to available data, we find that model performance heavily depends on the grain-size parameter, which is a strong influence on the sediment connectivity below the glacier. Thus we are able to use the model to suggest that sediment availability plays a larger role in sediment discharge from the glacier compared to other factors, such as the background bedrock production rate.

- **So can we conclude that the model is a good one? No. Does it then have predictive value? No. And, most importantly, should we expect it to be a good representation of physical process? In view of my comments above, I would have to say no.**

Nowhere in the manuscript do we say that the model is "good," that it has "predictive value" beyond the dataset it is calibrated to, or that it is necessarily a "good representation of physical processes".

In the introduction, we outline our objectives as:

*In this manuscript, we present SUGSET_2D, a two-dimensional subglacial sediment transport model. The model implements subglacial sediment transport and bedrock erosional processes presented in Delaney et al. (SUGSET; 2019). We apply a routing scheme to the model that transports sediment down-glacier based upon the hydraulic potential gradient. Synthetic test cases show the model's ability to reproduce known processes and yields insight into the spatially distributed processes responsible for subglacial sediment dynamics. Implementation of the model to existing glacier hydrology, topography, and sediment discharge datasets from the Griesgletscher helps to understand some subglacial sediment transport processes at this site that could be generalizable to other situations. Through these experiments, we discuss the impact of two dimensional sediment connectivity on subglacial sediment transport.*

In our opinion, the paper fulfills these objectives.

Smaller points: **At equation 1. This looks a bit odd to me. In my way of thinking, for force balance in a channel, you would have l = gSA, where is the stress, l the wetted perimeter, S a suitable slope and A the cross-sectional area. So = gS = l, A and if =fu2 and Q=Au, then = fQ2l. You can get equation(1) if l $\propto$ A1/2, as for a circular or triangular cross section; but it seems to me that equation (1) is not the basic law. For example, a wide stream has l $\propto$ A, approximately.**

This comment points to our decision in the previous manuscript to omit the complete description of the hydraulics model. This information has been added and explains the conversion of wetted perimeter to a cross-sectional area. In this way, equation 1 is the generalized law, with factor $s$ depending on the relationship between the hydraulic radius $l$ and the cross-sectional area $S$. The text is available in the response to Dr. Overeem's review.

**Line 105. This is unclear. Dh is a length, not an area.**

The text has been changed to read:

*We note that to prevent unreasonable water pressures when $Q_w^*$ rapidly increases and $D_h$ is small, the model limits the minimal cross-sectional area $S$ to $0.5$ $m^2$.*

**115. and to or, presumably.**

Done.

**181. The wording here suggests that a partial differential equation is being solved, but if I understand this correctly, this is not the case. (4) with (5) form a set of ordinary differential equations.**

The text now reads:

*We implement a regular grid for discretization. Spatial discretization must be substantially smaller than characteristic length-scale, $l$, in Equation 8. We then solve Equation 7 for till height $H$ for given initial and boundary conditions in response to till production $\dot{m}_t$ and divergence of the sediment discharge $Q_s$ using an explicit time integration scheme.*

**References**

Delaney, I., Werder, M., and Farinotti, D.: A Numerical Model for Fluvial Transport of Subglacial Sediment, Journal of Geophysical Research: Earth Surface, 124, 2197–2223, doi:10.1029/2019JF005004, 2019.

Gimbert, F., Gilbert, A., Gagliardini, O., Vincent, C., and Moreau, L.: Do Existing Theories Explain Seasonal to Multi-Decadal Changes in Glacier Basal Sliding Speed?, Geophysical Research Letters, 48, e2021GL092 858, doi:10.1029/2021GL092858, 2021.

---

## Author Response (AR1)

Dear Dr. Butcher,

I thank you for your work on this manuscript and for consideration following the reviewer's response.

As discussed in my response, I have addressed these issues. Additionally, I modified the cartoon figure with respect to your recommendations.

Please let me know if additional work is needed and how best to proceed.

Best regards,

Ian Delaney, on behalf of all authors.

---

## Referee Report (RR1)

This paper presents a two-dimensional model to capture the spatial heterogeneity of subglacial sediment transport and erosion. Overall, I believe the model is useful for the understanding of subglacial sediment transport. The results and analyses are also insightful to understand how spatial heterogeneity affects subglacial transport.

However, the paper may need to improve the way of describing the modeling framework and explaining the results. For modeling descriptions, first, this paper shows a 2D model, however, the governing equations do not show clearly show how the "2D" is represented in the model descriptions. Figures 3 and 9 show the 2D geometry of the studied area are complex geometries. However, it is not clear how these complex geometries are represented by mesh, and how such a mesh is incorporated into the governing equations. These details will be necessary for us to understand how the 2D is represented and variables defined on these 2D geometries are modeled. For the river routing model, Equation 16 shows the algorithm for calculating phi_o, however, the paper does not describe the governing equation for the river routing. It is not clear what governing equation and process are solved here. In addition, the model has a lot of parameters and variables that need to be solved. From the model description, it is not clear what variables are solved. Finally, the model used many equations that do not give sufficient descriptions of why you choose these equations, e.g., equations 2-4, 8, 9, 12, 16, 17, 21. I suggest the authors improve Section 2 to better describe these model equations.

For the results interpretations, I find it hard to link the conclusions or claims to the figures. In the current version, the conclusions usually come out first and then the claim is referred to a Figure. The linkage between the results and the message the figures describe are missing and is left for the potential audience to have the best guess. For example, on line 228-229, the paper states "Simulations show that over seasonal timescales, sediment discharge increases at the onset of melt and decreases shortly thereafter, prior to the maximum amount of water discharge that occurs each melt season (Figure 4)." However, Figure 4 has two subfigures and 5 lines. It is not straightforward to align the message of Figure 4 to the claim made here. This issue occurs in most of the explanations of the results. For example, at the lines 230 (linkage to Figure 6), 243 (linkage to Figure 4c); line 247 (link to Figure 6); line 265 (link to Figure 3a); line 393 (link to Figure 10a). The paper also has a few issues with missing figure titles such as Figure 4,5 and insufficient descriptions for each data, line, and color of a lot of subfigures (Figure 4a: sloping blue line). The figures use a lot of double y-axes. It is better to clearly define what each axis mean in the caption or inside the figure, not just by using different color of lines and leaving the potential readers to identify which one is which one. I suggest the authors pay more attention to the details of the figure legends, captions, and subtitles, trying to make sure each figure tells the message on its own.

More details of the comments are listed as follows for reference:

Line 4: add "are" after sediment

Line 41: move "to" to "explore"

Line 69: change "subglacial the" to "the subglacial"

Line 111: is there a reference for selecting 0.5 m^2 as the limit?

Line 122: is "w" the same as "wc" shown in equation 3?

Equation 16: where does this equation come from? Do you have a governing equation for this equation? What is the difference between the two phi_o?

Line 184: how to determine the w_{rj}?

Equation 17: What is the difference between phi_o, phi^*, and \Phi_*? How does this equation relate to Equation 6?

Line 191: what is "mn"?

Equation 18: what is the governing equation of this equation?

Equation 19: is Qs in the right-hand side the same as Qw defined in equation 18?

Equation 21: why do you choose Equation 21 to represent temperature?

Line 228/Figure 4 caption: From Figure 4, I can see that the sediment discharge is highest at years 19-26, why are you saying the highest discharge is in years 14-17?

Figure 4b: the y-labels are the same for the two lines.

Figure 4a: it seems the water discharge starts to increase at year 10, but the captions say from year 12. Could you explain?

Figure 4: subfigure title a, b are missing.

Line 230: Figure 6 includes 6 subfigures. Which subfigure should I see to support the claim you made here?

Line 234-239: The descriptions in this paragraph are not well supported by the model results. These descriptions are more like conclusions but are not results. It is hard to link these claims to the results.

Line 242: Figure 5 shows the results are distributed values, which means these variables vary with coordinates x and y. However, I didn't see an equation in the method section describing x and y. What equations are solved to obtain these distributed values?

Line 243: Figure 4c is missing.

Line 247: Which subfigure in Figure 6 am I supposed to observe to understand the claim here? What is the meaning of early melt season? Do you mean the time at year 8.3? This needs accurate descriptions.

Line 250: It will be useful to draw a line in Figure 4 to show which time is spring.

Line 251-252: This claim cannot be observed in Figure 4b. I can observe that the highest sediment discharge occurs at years around 11.5 and 16.5. The sediment discharges in this time period do not show a decreasing trend. Where does this claim come from?

Line 253: Need to show where is winter time in Figure 4b.

Line 266-271: In the paragraph at line 260, Figure 6a, b has been referenced, however, the difference between different subfigures is defined in the paragraph at line 270. This makes it hard to understand the paragraph above. The order of these two paragraphs needs to switch.

Line 291: As you randomly select parameters, does this mean you are performing a sensitivity study? Why do you need to randomly select these parameters?

Line 300: As the parameters affect the model results, it is necessary to show the final parameters that give the lowest absolute error.

Line 304: From Figure 7, It seems the relative error between the model and observation is very large for most years except for the time between 2015-2016. Why do you only say it has trouble during 2012-2013?

Line 313: How is this claim supported by the results?

Line 314: The absolute error corresponding to B and Ht0 vary a lot. How does this support the claim of "minimal influence" of these parameters?

Line 329: In Figure 7e, the units for B is Mpa m/s. How does variable B in Figure B related to the "B" here? It seems the B has a velocity unit here, different from that in Figure 7e.

Line 330: How does this slide speed are calculated? From the model?

Line 339: How do you calculate this value 2 m^3/s?

Line 341: I guess sediment discharge should be Qs?

Figure captions: most of the current figure captions include certain explanations of the results. These explanations make the captions very long and not easy to understand. I suggest only including the descriptions for the line title, legends, and meaning in the captions, but leave the explanations of the results in the main text. Please try to make the captions short but can sufficiently tell the meaning of each line.

---

## Referee Report (RR2)

I think the authors' huge efforts to address my previous comments. The current version of the paper has improved a lot in terms of the model descriptions and the analyses of the model results. However, the paper still needs certain improvements in (1) model descriptions (2) results analyses in Section 3.1; (3) and the Conclusion.

For the model description, specific comments include:

(1a): For Section 2.1, the main goal of this section is to provide a way to "estimate" the Hydraulic diameter Dh. The main assumption is that Dh, a characteristic size of subglacial conduits, does not change too much over a certain period. With such an assumption, the Qw in Equation 1 can be estimated using a time-averaged value of discharge during the period. And the pressure gradient can be estimated from the Shreve potential gradient. With these two pieces of information, the Dh in Equation 1 can be approximately estimated and kept unchanged in other conditions. Though the paper finally achieves this goal in lines 109-111, I suggest the authors add the key motivations and assumptions of this Section at the very beginning of the Section, which will improve the readability of section 2.1.

(1b): Section 2.3.1 need improvements in two aspects: better descriptions for the grid and its sizes/area etc, donor, receiver, and the final linear combinations used for equation 18, 19, and 21; better descriptions for the routing schemes for equations 18-22. First, Section 2.3.1 is trying to obtain distributed values for hydraulic potential, water discharge, sediment discharge, and till height in a 2D domain. In a 2D domain, any grid (with grid id i) has multiple neighboring grids that can exchange potential/discharge/height to its donor grid. And how much exchanged is further affected by the grid size and the approach to calculating gradients between neighboring grids. In short, such a process is similar to the numerical discretization process and usually very complex. In this paper, lines 174-178, 184-185, 190-191, 192-193, 196, and 197 aim to describe these processes, however, are not successful, in my perspective, to explain how the grid size, weights, and equations (18.19,21) are organized together. To help clarify the equations in this section, I suggest the authors sketch to carefully describe mesh, donor, receiver, and variables defined at each donor grid in a 2D grid domain and then use this sketch to improve the description for equations 16-22. To draw such a sketch, the authors can check figure 3.1.1.1 or 3.1.1.2 at this link: http://www.thevisualroom.com/finite_volume_method_3.html. In this sketch, it is necessary to clarify the following definitions: lambda, nr, wr,j, phioi, phioj in Equation 18; delta, nd, wd,j, Qwi, Qwj in equation 19, lambda and nd, Qsi, Qsj in equation 21, and Hi, mti in equation 22. With the clarification of these terms in the sketch, section 2.3.1 will become clear.

For the result analyses: in Section 3.1.2, the texts in the paper are not consistent with the Figures shown in Figure 4. In Figure 4 caption, it says "a) Seasonally varying water discharge (Qw) increases from year 10 to 20, while till height (H) decreases. b) Annual sediment discharge (green) increases over with increasing melt". However, Figure 4b sediment discharge does not show an increasing trend, but shows a "decrease, increase, and then decrease" trend. A similar problem occurs for figures 5b,d,f at lines 257-258. In the author's response, the author mentioned this may be caused by using an old version of the manuscript, the authors are suggested to carefully re-exam all results in Section 3.1.2 to make sure the texts in the current version agree with the figures.

For the Conclusion section, the main problem is the Conclusion is not supported by quantitative evidence. The Conclusion in the current version has 4 paragraphs: the first one introduces the 2D model; the second one discusses the limitation, the third one discusses the future work, and the last one has two sentences commenting on the results and two sentences discussing further work. From my understanding, a

Conclusion should include what you did and what you have discovered, and provide quantitative evidence to support your discovery. The first paragraph did describe the 2D model, which is good. But no quantitative evidence to support your discovery makes the Conclusion very weak. Also, limitation and future work should NOT be the main texts in the Conclusion because they are not the 'Novel contribution' of the paper. Therefore, I believe the Conclusion section requires significant improvement, which needs to provide concrete QUANTITATIVE evidence to support the main Conclusions of the paper.

Other specific comments are listed below for reference:

Line 88: change hydraulic gradient to hydraulic pressure gradient

Equation 1: maybe change $\Psi$ to $\Psi(x, y, t)$, which helps to clarify the pressure gradient is a time-dependent and 2D distribution variable.

Equation 2: can you a sketch the angle $\beta$ in Figure 1 or 2?

Line 96: change "establish" to "estimate" or "approximate"?

Line 98: change "we call the source percentile" to "which is called the source percentile".

Line 97-98: How the surface melt and resulting discharge control the subglacial conduit diameter is a complicated problem. Could you add a few citations to support this assumption, if any?

Line 103: I think Qw is the instantaneous discharge, while Qw* is a representative scale of the instantaneous discharge. What is the "total instantaneous amount"?

Line 96-104: some of the texts here are confusing. Qw is the water inside the subglacial conduits. Without considering ice melting inside the conduit, such water should be discharged from surface melting, therefore, I would say water in the conduit equals the surface melting into the conduit, based on mass conservation, therefore, Qw = mw. In general, the surface melting varies with time, so here you assume mw is estimated by Qw* which could be a time-averaged value over a certain period (hours to days as mentioned). In thinking in this way, it is logical converting equation 1 to equation 3. However, lines 96, 99-100, and 103-104 cause confusion. I would suggest revising the texts between 96-104 to better reflect the logic as I suggested above.

Line 99-102: Two variables, sa and sp are introduced. And further explanations are added. However, these two variables are not used in Equations 3 and 4. They may be not useful to help us understand why equation 3 could be derived from equation 1, but may confuse readers. I suggest removing this information from lines 96-104.

Line 109: is Dh a function of x and y?

Line 109: Add one sentence at the beginning of this paragraph: With the data of representative surface melting rate Qw* and the static hydraulic pressure gradient, a representative hydraulic diameter Dh can be estimated. For a given short period, such a Dh is assumed time-independent and used in Equation 1.

Line 124: How does this conversion happen? Do you mean Qs = Qs~/w?

Equation 6: What is the necessity to use Qs~ rather than Qs? Trying to use variables as less as possible, if existing variables are sufficient to tell the story. Revise equation 6a to a formula of Qs but not Qs~.

Equation 15: what is the difference between $\tau$ in Equation 11 and $\tau_b$ in equation 15? Do they equal to each other?

Equation 15: does $\sin\alpha = \nabla z_s$?

Line 172: What is the purpose of flow routing? Is it used to distribute water potential at different cells?

Line 172: change "multicell routing scheme" to "2-D distributed routing scheme"?

Line 174: what is a "regular grid"? square grid or rectangular grid? Also, the result in Figure 3 shows an irregular distribution, please clarify this.

Line 174: What do "fluxes" mean? Is water flux driven by pressure gradients?

Line 175: Not sure what is a "stack" in Table 3. It is better to draw a 2D sketch to visualize st, nd, and nr for a single cell. See major comments 1b.

Line 175-178: These sentences describe the process of numerical discretization. The value of the cell, i.e., a stack defined here, is a linear combination of the values of certain variables of neighboring cells. There are different types of numerical discretizations and thus result in different linear combinations. As different combination means different discretization schemes have different accuracy, so it is generally required to describe what discretization schemes are used. The current sentences are very general descriptions of the discretization principle. Please refer to the major comment 1b to better describe this part.

Equation 18: In lines 176-177, you defined donors and receivers, why did you only use the information of potential from receives, i.e., summation from 1 to nr in Equation 18. Here the wr,j is determined by discretization schemes. How do you calculate this?

Line 186-187: It is very hard to understand this sentence by reading just words. You have to provide a figure to describe what is donor, receiver, edge length, etc. This is a typical practice for papers that involve numerical discretization algorithms. Also, for 1D grid, it only has one edge length. But for 2D grid, it has two edges and thus two edge lengths. For the regular grid, if the grid is square, then we can say it has only one edge, but if it is rectangular, it has two edges. As your paper is a 2D model, please make sure if you use a square grid or not.

Line 188: you mentioned "beginning at the based and moving up to the glacier", does this sentence mean the calculation is performed along a line from the base to somewhere? What does "flow paths evaluated in the routing scheme mean? You need to provide a sketch to carefully describe these details.

Equation 19: So you are applying the routing scheme to calculate a distributed discharge for water? Again, what is $\delta$, $w_{d,j}$,? And why you only calculate summation over donor cells? These really require a detailed figure to describe this. Combining equations 19 and 18, there have too many variables, it is impossible to accurately understand the meaning and how you calculate without a detailed sketch. By the way, as you give $\dot{m}_{w,l}$ different value for different cell, so do you mean you prescribed a distributed meltwater source term along a certain line?

Line 192: What is $\overline{Q_{s,i}}$? Where does this term come from and why you need to calculate this term? Most importantly, how are equations 20a-c derived? The term "like-wise" does not explain how this term is derived.

Line 195: Need a figure to describe what is $\lambda$? What is a response length scale? And where is this come from?

Line 196: how is equation 21 derived? Why Qsi = $\lambda Q_{s,i}$ + ...?

Line 198: Need to define cell area?

Line 215: Need a figure to show where the edge cells are.

Section 3.1 title: You only have one synthetic case. Use a singular form

Line 235: It is better to reproduce and visualize the synthetic glacier geometry in your paper.

Line 238: what is "laterally"? You haven't defined any coordinates here, so no way to understand "laterally".

Equation 23: What is the unit of T? For 0, is the unit K or C?

Line 249: o is not the same as Celsius (oC).

Line 255: Figure 4 appears earlier than Figure 3.

Line 256-257: In Figures 5b,d,f, each figure has two lines. One is purple and another one is yellow. So which line is the "Daily-averaged sediment discharge"? However,both curves in 5b,d,f shows complex behaviors. For purple lines b and f, they show decreasing, increasing, and decreasing trends. For d, it shows an increasing, constant, and decreasing trend. The yellow lines show more complex behaviors. In short, figures 5b,d,f show different behaviors compared to what you say at line 257 (decreases until....). Please carefully analyze the figures and make your text descriptions consistent with your figure.

Line 264: How do you define mean till height?

Line 295: Can you elaborate on the 4 time periods?

Line 314: add (see red stars in Figure 7a-c) after "… till height H0 of 2.5 cm".

Line 319: Why is the data a constant within each year? For example, the data is constant between late 2011 - late 2013?

Line 319: From Figure 7a,b,c, What is the absolute error for the optimal run (red star case)? Is the value 62,600? I can not see it clearly from the y-axis in Figure 6a.

Line 325: change "short-lived' to "short-lived period".

Line 325: This claim needs supporting evidence. From Figure 8a, you can identify the peak values for the sediment discharge. Meanwhile, you can obtain the corresponding water discharge. You can calculate a correlation between the two values. If the correlation coefficient is high, then this claim is supported.

Line 328: what is "high on the glacier"?

Line 340: check "and thus, glacier, conditions".

Line 390: check "This is especially so"

Line 398: Can you explain this sentence more? Also, Figure 10 appears only once in the whole paper, do you need to explain the meaning of Figure 10 more?

Conclusion section: The Conclusions are all qualitative descriptions or perspectives or future plans. No quantitative metrics or summaries are included. In my view, a Conclusion section written in this way is not a solid Conclusion. The Conclusion needs quantitative evidence to support it. Please try to summarize the paper and provide quantitative metrics to support your Conclusion.

Line 405-407: From my understanding, the main contributions are two: (1) proposed a 2D subglacial sediment transport model; (2) reproduced certain phenomena reported in previous work. The first paragraph (403-407) describes the first contribution, but the contribution for the second key point is vague. As mentioned in the first paragraph, there is a "need" and the model reproduced "many observed processes". However, in this conclusion section, it is not clear what the "need" is and what processes are reproduced. The authors should explicitly summarize the key findings in the paper and use the summary to support your "Conclusion".

Line 408-412: this is the limitation of the model, which is not a "Conclusion".

Line 413-419: These sentences belong to the future work. However, future work is not the work done in the current paper, it is not necessary to add these in the "Conclusion". They can be merged into the "Limitation" section.

Table 2: As mentioned in the paper, some of the parameters are adopted from other papers. For all the parameters not measured by the present paper, a reference is required. Potential audiences will need accurate references to verify how each of these parameters is derived.

Figure 4: the captions are not consistent with the Figure. For example, the green line in Figure 4b shows a decrease, increase, and then decrease behavior. But the caption here says "increases". Please have a careful check for all the figures and captions. The main text need further check if the Figure and captions are changed.

---

## Referee Report (RR3)

I thank the authors for their substantial revisions made to the paper. The details of the equations and numerical implementation have been significantly improved. Most of my previous comments have been addressed. I only have a few comments for the authors to final consideration. Please pay attention to comments 4, 5, 8, and 9. The equations mentioned for these comments have inconsistent units.

1. Line 10: change "Experiments" to "Numerical experiment".
2. Line 12: add "also" between "we" and "apply".
3. Line 91: change "Weissbach" to "Weisbach"
4. Equation 1: It seems wc was not explained. Also, in the previous version of the paper, Equation 1 does not have a wc term, but you have one in this version, could you explain the difference? Based on my ow calculation, the ratio of Psi*/rho_i has a unit of m/s^2 based on Equation 4. For equation 1, the ratio of Phi to rho_w should be the same unit, which means Psi/rho_w ~ Q_w^2/Dh^5*wc. However, the last term (Q_w^2/Dh^5*wc) has a unit of (m^3/s)^2/m^5*m =m^2/s^2. This means that the units of Psi in Equation 1 and Equation 4 are inconsistent. I suspect that wc should be removed from Equation 1. Please check this problem.
5. Equation 3: Equation 1 has the variable wc, while Equation 3 does not. Please check this inconsistency.
6. Line 124: Change "the first term" to "the first term on the right-hand side".
7. Line 128: what is the difference between "a width of the glacier bed w" at line 128 and "channel width wc" defined at Equation 9? Are they the same or not? If they are the same, please explicitly describe this. If not, then please explain the difference and clarify how to calculate glacier bed width w.
8. Line 135: In Equation 7, is the term, (2 - Delta sigma H)/5, a dimensionless value? I suspect the format Delta sigma*H should be Delta sigma/H? From the texts at Line 135, Delta sigma has the same unit of H. The unit will be the squared unit of H if you multiply Delta sigma with H. Please add clarification for this.
   If I am correct that (2 - Delta sigma H)/5 should be (2 - Delta sigma/H)/5, then H = Delta sigma, means sigma(H) = (1+exp(1/5))^{-1} = 0.45, which is not 0 as you mentioned at line 135. Please check on this.
9. Equation 8: In the previous version, the right-hand term is multiplied by wc, but in this new version, the wc is omitted. Is there a reason to do this? Based on my calculation, the units for Dm, g, and (tau/rho_w)^(5/2) is m, m/s^2, and (m/s)^5, this means that the Qsc has a unit of m^2/s. Based on Equations 5 and 6, erosion rate mt. has a unit of m/s (identical to partial H/partial t), which means Qsc and Qs should have a unit of m^3/s because their units are proportional to mt*w*l (with unit m^3/s) based on Equation 6a. These calculations mean that the new version of Equation 8 is not correct, while the old version is correct, in terms of their units. Please check if there is a mistyping error.
10. Line 254: add parenthesis to separate T and C?
11. Line 339: Here you mentioned that the grain size is the most influential factor controlling the model's predictive capability. In my understanding, the grain size also has impacts on the Darcy-Weisbach friction factor. In this paper, the friction factor is assumed as a constant. Could you add a few comments on how the combined impacts of grain size on friction factor (Equations 1 and 11) and transport capacity (Equation 8) can likely affect the mode performance?
12. Line 359: add a space between "13" and "are".

---

## Author Response (AR2)

We appreciate this review and the need to improve the description of the model's implementation. As a result, we have addressed the reviewer's comments to the best of our ability and interpretation of the comments.

Below, Dr. Hergarten's comments are in bold and our response is in normal font. Changes to the manuscript text are in italics.

Best regards,
Ian Delaney, on behalf of all authors

1. **Let me start from the sediment balance (Eqs. 7 and 8). If I am not totally wrong, (capital) Qs is still the sediment flux (volume per time). Then the divergence of Qs has a unit of square meters per second, while the other terms in Eq. 7 are meters per second. So the Exner equation is still not written correctly. The term inside the divergence must be flux per unit width, not flux! Then you may approximate it by flux per width. From its physical dimension, your property Qs (line 122) would be m4 per second, which would make no sense at all. So please write the sediment balance correctly**

   We appreciate this comment and have corrected the sediment transport equation. The text now reads:

   *The model simulates the evolution of a subglacial till layer, which we define as transportable sediment below the glacier due to glacier erosion and fluvial sediment transport. Fluvial sediment transport, in supply- and transport-limited regimes, mobilizes and deposits sediment, adding or removing material from the till layer (Brinkerhoff et al., 2017; Delaney et al., 2019). Conversely, erosive processes such as abrasion and quarrying add material to the layer, while we do not consider processes such as fluvial abrasion that appear to produce minimal sediment (Beaud et al., 2018). To represent these processes, we implement the Exner Equation (Figure ??; Exner, 1920a,b; Paola and Voller, 2005), a mass conservation relationship, to solve for the till layer height given the erosive and fluvial conditions.*

   $$\underbrace{\frac{\partial H}{\partial t}}_{\text{till evolution}} = - \underbrace{\nabla \cdot Q_s}_{\text{sediment transport}} + \underbrace{\dot{m}_t}_{\text{bedrock erosion}} , \qquad (1)$$

   *$H$ is till thickness and $t$ is time (Table 1). The first term represents fluvial sediment transport processes, where $\nabla \cdot Q_s$ represents sediment mobilization in either supply- or transport-limited regimes. The second term captures bedrock erosion processes, where $\dot{m}_t$ is a bedrock erosion rate.*

   *We evaluate the mobilization of sediment in both supply- and transport- limited conditions. Divergence of the sediment flux is evaluated by approximating $\nabla \cdot Q_s$ with $\frac{\widetilde{\nabla \cdot Q_s}}{w}$ and using the mobilization scheme from Delaney et al. (2019)*

   $$\nabla \cdot \widetilde{Q_s} = \begin{cases} \dfrac{Q_{sc} - Q_s}{l} & \text{if } \frac{Q_{sc} - Q_s}{l} \leq \dot{m}_t w \qquad \text{(transport-limited)} & (2a) \\[2ex] 0 & \text{if } H = H_{lim} \quad \& \quad \frac{Q_{sc} - Q_s}{l} \leq 0 & (2b) \\[2ex] \dfrac{Q_{sc} - Q_s}{l} \sigma(H) + \dot{m}_t w \left(1 - \sigma(H)\right) & \text{otherwise} \qquad \text{(supply-limited)} & (2c) \end{cases}$$

   *$Q_{sc}$ is sediment transport capacity, or the amount of sediment that could be transported under the given hydraulic conditions. $l$ is a characteristic length-scale for sediment mobilization, over which sediment mobilization adjusts to sediment transport conditions. $\sigma$ is a sigmoidal function of $H$*

In this form, Condition 2a, $\frac{Q_{sc}-Q_s}{l}$, has units $\mathrm{m^2\,s^{-1}}$ as does Condition 2b and the term $\dot{m}_t w$. As a result, $\nabla \cdot \widetilde{Q_s}$ has units $\mathrm{m^2\,s^{-1}}$. In turn, when we approximate $\nabla \cdot Q_s$ with $\frac{\nabla \cdot \widetilde{Q_s}}{w}$, the unit of $\nabla \cdot Q_s$ becomes $\mathrm{m\,s^{-1}}$. Units of $\mathrm{m\,s^{-1}}$ for $\nabla \cdot Q_s$ make Equation 5 consistent. Here $w$ is a width perpendicular to water flow.

2. **As a second point, the description of the routing scheme and solving the system is weird. From Eq. 16, I guess that you compute the hydraulic potential of each nodes by some kind of weighted mean of the potentials of the receivers and the respective hydraulic gradients. Equation 16, is however, strange because it is just $\phi_0 = \phi_0 +$ something, which cannot hold in this form. And what are the hydraulic gradients inside the sum? Are these the gradients from the node to the respective receiver or the hydraulic gradient of the receiver? The latter would be questionable if, e.g., a small channel enters a big channel.**

   We appreciate this comment, especially in light of similar comments by the other reviewer. In turn, this Section 2.3.1 has been rewritten as to address these concerns. The section in its entirety is below. In summary, $\phi_0$ is evaluated by summing the hydraulic potential up the glacier from the terminus. In one dimension, this would be evaluated by integrating the hydraulic potential gradient in the Darcy-Weisbach (Equation 1 in the manuscript) up-glacier from the terminus. In two dimensions, we use the routing information from the stack to evaluate this.

3. **Then there is a part (lines 186-195) that I do not understand well.**

   We understand why the reviewer may not understand this well. In Section 2.3.2 below, please find our updated phrasing. We believe that these matters should be clarified in the new presentation.

4. **For the following part (from line 196), I have some idea, but there still seems to be something wrong. Neither the flux of water (Eq. 18) nor the flux of sediment (Eq. 19) can be computed from a sum over the receivers. I guess it is a sum over what is sometimes called donors or donators in the literature.**

   The reviewer is correct in the comment and identified short comings and errors in our description. In evaluating the hydraulic gradient, we are moving up the glacier, so "receivers" from a downstream perspective are used. For the sediment and water discharge, we use the "donor" cells of each cell. We have updated the text to reflect this. Additionally, balances of sediment and changes in till height have been modified to reflect these changes.

**Section 2.3.1**

[revised manuscript text omitted]

We appreciate this review and the need to improve the description of the model's implementation. Additionally, we have taken to heart the reviewer's comments regarding the clarity of the text and figures. As a result, we have addressed the reviewer's comments to the best of our ability and interpretation of the comments.

Below, Reviewer 4's comments are in bold and our response is in normal font. Changes to the manuscript text are in italics.

Best regards,
Ian Delaney, on behalf of all authors

**1 General Comments**

1. **However, the paper may need to improve the way of describing the modeling framework and explaining the results. For modeling descriptions, first, this paper shows a 2D model, however, the governing equations do not show clearly show how the 2D is represented in the model descriptions. Figures 3 and 9 show the 2D geometry of the studied area are complex geometries. However, it is not clear how these complex geometries are represented by mesh, and how such a mesh is incorporated into the governing equations. These details will be necessary for us to understand how the 2D is represented and variables defined on these 2D geometries are modeled. For the river routing model, Equation 16 shows the algorithm for calculating $phi\_o$, however, the paper does not describe the governing equation for the river routing. It is not clear what governing equation and process are solved here. In addition, the model has a lot of parameters and variables that need to be solved. From the model description, it is not clear what variables are solved. Finally, the model used many equations that do not give sufficient descriptions of why you choose these equations, e.g., equations 2-4, 8, 9, 12, 16, 17, 21. I suggest the authors improve Section 2 to better describe these model equations.**

   We have tried to implement these comments as well as possible. In particular, we have reorganized Section 2.3 and re-written Section 2.3.1 to address how these equations are implemented is a 20dimensional space.

   In response to the last part of this comment, regarding the "sufficient description" of these equations, we have tried to address these. However, many of them are commonly used in the sediment transport and glaciological literature and we have provided citations. In these cases, we believe that the justification for their use is adequately described. To clarify some of the matters, we have added the following comments surrounding some of the equations mentioned above:

   - For Equations 2-4 we have added the following comments:
     *the channel's cross-sectional geometry, which impacts water pressure, is accounted for by $s$ (Hooke et al., 1990)...* and *The width of the channel floor $w_c$, needed to evaluate the surface over-which sediment transport may occur, is given by...*
     We note that Equations 3 and 4 have been moved to Section 2.2 as they are a direct control on the sediment transport relationships therein.

   - For Equation 8 (now 6), we have added the following sentences to the description
     *We evaluate the mobilization of sediment in both supply- and transport- limited conditions.*
     *With these three conditions, we can evaluate sediment transport in transport- and supply-limited regimes and pass sediment through the system when till hight is large.*

- We gave modified the text surrounding Equation 9 (now 7) to include:

  *If $H$ is greater than $3\,\Delta\sigma$, then sediment mobilization is unaffected and the system is in a transport-limited regime. When $H = \Delta\sigma$, then $\sigma(H)$ is close to $0$, sediment transport is in a supply-limited regime, and nearly no sediment mobilization takes place.*

- With regard to Equation 12 (now), we have added the sentence:

  *We assume that till armors the bed from erosion (e.g. Alley et al., 2003; Brinkerhoff et al., 2017; Delaney et al., 2019).*

- For Equations 16 and 17, the text has been rewritten substantially in this section and these equations have been modified.

- With respect to Equation 21, we describe the justification below in the specific comments. However, we note that this equation encapsulates seasonal and diurnal variations in melt. This allows us to provide a synthetic forcing to our model that mimics natural conditions.

2. **For the results interpretations, I find it hard to link the conclusions or claims to the figures. In the current version, the conclusions usually come out first and then the claim is referred to a Figure. The linkage between the results and the message the figures describe are missing and is left for the potential audience to have the best guess. For example, on line 228-229, the paper states "Simulations show that over seasonal timescales, sediment discharge increases at the onset of melt and decreases shortly thereafter, prior to the maximum amount of water discharge that occurs each melt season (Figure 4)." However, Figure 4 has two subfigures and 5 lines. It is not straightforward to align the message of Figure 4 to the claim made here. This issue occurs in most of the explanations of the results.For example, at the lines 230 (linkage to Figure 6), 243 (linkage to Figure 4c); line 247 (link to Figure 6); line 265 (link to Figure 3a); line 393 (link to Figure 10a). The paper also has a few issues with missing figure titles such as Figure 4,5 and insufficient descriptions for each data, line, and color of a lot of subfigures (Figure 4a: sloping blue line). The figures use a lot of double y-axes. It is better to clearly define what each axis mean in the caption or inside the figure, not just by using different color of lines and leaving the potential readers to identify which one is which one. I suggest the authors pay more attention to the details of the figure legends, captions, and subtitles, trying to make sure each figure tells the message on its own.**

   We are deeply grateful that the reviewer has identified these issues, and we have addressed them in the manuscript. In this current version of the manuscript, we have followed the reviewer's comments and carefully evaluated all figures to ensure their clarity. This includes reexamining the text to include references to equations and figures that support these comments. Additionally, we have restructured some aspects of the paper so that our interpretation of model outputs and their significance exists in the "Implications" section. Lastly, we note that some sentences in the text have remained from a previous experiment that was modified between the drafts. While we are uncertain of the source of this error, we are grateful that the reviewer identified these cases. I apologize for the confusion and inconsistency that have been created by my oversight.

**2 Specific Comments**

1. **Line 4: add are after sediment**

   Done.

2. **Line 41: move to to explore**

   Done.

3. **Line 69: change subglacial the to the subglacial**

   Done.

4. **Line 111: is there a reference for selecting $0.5 m^2$ as the limit?**

   There is no reference for this, explicitly. However, it is a common issue in the stability of subglacial hydrology models and has been applied previously in (Delaney et al., 2019). This paper is now cited.

5. **Line 122: is w the same as wc shown in equation 3?**

   $w$ and $w_c$ are different. $w_c$ refers to the width of a channel and is calculated based upon the hydraulic conditions and shape of the subglacial conduit. $w$ refers to a patch of the glacier bed over which sediment can be accessed. In the mesh, this would be the width of a grid cell. To clarify this matter, we have added $w$ to Figure 2 (the cartoon of a cell). The text now reads: *$w$ is the width of a patch of glacier bed perpendicular to the direction of water flow overwhich sediment can be accessed by a channel.*

6. **Equation 16: where does this equation come from? Do you have a governing equation for this equation? What is the difference between the two $phi_o$?**

   We appreciate this comment and realize how our representation of this an be confusing.

   The text now reads:

   *We assume that sediment and water moves across the glacier bed following the steepest gradient in hydraulic potential. On glaciers, we define the hydraulic potential at a cell $i$ in the grid, $\phi_i$, based upon the elevation of the glacier bed plus the ice thickness, following Shreve (1972).*

   $$\phi_i = f_f \, \rho_i \, g \, (z_{s,i} - z_{b,i}) + \rho_w \, g \, z_{b,i} \quad , \tag{1}$$

   *where $f_f$ is the flotation fraction across the glacier, $z_s$ is the glacier surface, and $z_b$ is the glacier bed.*

   *With this information, we use a multi-cell routing scheme (Quinn et al., 1991) to establish flow routing based upon the steepest hydraulic potential in Equation 16and with a single value of $f_f$ across the glacier bed. We implement this scheme in a similar way as Bovy et al. (2016), but on a regular grid in $x$ and $y$ directions, where fluxes can pass to the four surrounding cells sharing an edge. This routing scheme returns a stack ($s_t$; Table 3), which contains information about the order of cells to perform the calculations, along with the number of cells flowing in to a cell (donors; $n_d$), number of cells that a cell contributes (receivers; $n_r$), and the weight of hydraulic potential and water or sediment discharge directed from one cell to another ($w_d$ or $w_r$).*

   *For the first time step, the hydraulic potential $\phi$ is evaluated under the condition that $f_f = 1$. After the first time step, we assume that the flotation fraction, will vary in response to changing hydraulic conditions such as diurnal or seasonal water input (e.g., Iken and Bindschadler, 1986). In turn, to establish an average flotation fraction $f_f$ across the glacier bed for Equation 16, we use*

   $$f_f = \text{mean}\left( \frac{\phi_{o,i}}{\rho_i \, g \, (z_{s,i} - z_{b,i}) + \rho_w \, g \, z_{b,i}} \right) \quad , \tag{2}$$

   *where the denominator represents the hydraulic potential at overburden pressure ($f_f = 1$ in Equation 16).*

$\phi_0$ *represents the hydraulic potential evaluated from summing the hydraulic gradient $\Psi$ in Equation 1 up glacier from its outlet. $\phi_0$ at each cell $i$ is evaluated as*

$$\phi_{o,i} = \Psi_i \cdot \lambda + \sum_{j=1}^{n_r}(\phi_{0,j} \cdot w_{r,j}) \ .$$

(3)

*Here, $\Psi_i$ comes from evaluating Equation 1 from the receiver cell $j$ of $i$, $\lambda$ is edge length of a cell on a regular grid, $n_r$ is the number of receivers that the cell $i$ has, and $w_r$ is the proportion of hydraulic potential fed by the upstream cell $j$. The operation is executed on a cell by cell basis, beginning at the base of the glacier and moving up the flow paths evaluated in the routing scheme.*

7. **Line 184: how to determine the $w_{rj}$?**

   We evaluate $w_{rj}$ based on the commonly used multi-cell routing scheme from (Quinn et al., 1991). The follow paragraph has been modified from the previous manuscript that describes how $w_{rj}$ is evaluated.

   *With this information, we use a multi-cell routing scheme (Quinn et al., 1991) to establish flow routing based upon the steepest hydraulic potential in Equation 16and with a single value of $f_f$ across the glacier bed. We implement this scheme in a similar way as Bovy et al. (2016), but on a regular grid in $x$ and $y$ directions, where fluxes can pass to the four surrounding cells sharing an edge. This routing scheme returns a stack ($s_t$; Table 3), which contains information about the order of cells to perform the calculations, along with the number of cells flowing in to a cell (donors; $n_d$), number of cells that a cell contributes (receivers; $n_r$), and the weight of hydraulic potential and water or sediment discharge directed from one cell to another ($w_d$ or $w_r$).*

8. **Equation 17: What is the difference between $\phi_o$, $\phi^*$, and $\Phi_*$? How does this equation relate to Equation 6?**

   We have rewritten this section, and the text pertaining to this part of the hydraulics is discussed above.

9. **Line 191: what is mn?**

   mn refers to minutes. The text has been simplified to reflect this.

10. **Equation 18: what is the governing equation of this equation?**

    This is simply the accumulation of water as it flows up glaciers given a prescribed melt rate $\dot{m}_w$. In Section 2.1 we write *. . . we prescribe a melt rate $\dot{m}_w$ to establish $Q_w$ . . .*. Here, we simply state how water discharge at a cell is established.

11. **Equation 19: is Qs in the right-hand side the same as Qw defined in equation 18?**

    This is correct. Because $Q_s$ and $Q_w$ are calculated in different ways, we have re-written these equations as:

    *Sediment mobilization into a cell $\overline{Q_{s,i}}$ is like-wise computed by implementing Equation 6 from the top of the glacier through the stack as*

$$\overline{Q_{s,i}} = \begin{cases} \displaystyle\sum_{j=1}^{n_d} \left( \frac{Q_{sc,j} - Q_{s,j}}{l} \cdot w_{d,j} \right) & \text{if } \sum_{j=1}^{n_d} \left( \frac{Q_{sc,j} - Q_{s,j}}{l} \right) \cdot w_{d,j} \leq \dot{m}_{t,i} & \text{(4a)} \\[3mm] 0 & \text{if } H_j = H_{lim} \quad \& \quad \frac{Q_{sc,j} - Q_{s,j}}{l} \leq 0 & \text{(4b)} \\[3mm] \dot{m}_{t,i}\lambda \ (1 - \sigma(H)) + \displaystyle\sum_{j=1}^{n_d} \left( \frac{Q_{sc,j} - Q_{s,j}}{l} \right) \cdot \sigma(H)\,w_{d,j} & \text{otherwise} & \text{(4c)} \end{cases}$$

*where $Q_{sc,j}$ is the sediment transport capacity from cell $j$ flowing to $i$, $Q_{s,j}$ is sediment discharge entering from cell $j$ to cell $i$, again $l$ is a response length scale and $\lambda$ is cell length.*

*Sediment discharge $Q_{s,i}$ out of a cell $i$ is evaluated as*

$$Q_{s,i} = \overline{Q_{s,i}} \cdot \lambda + \sum_{j=1}^{n_d} Q_{s,j} \quad . \tag{5}$$

*We evaluate the change in till height at a cell by implementing Equation 5 as*

$$\frac{dH_i}{dt} = \frac{-Q_{s,i} + \sum_{j=1}^{n_d} Q_{s,j}}{\delta} + \dot{m}_{t,i} \quad , \tag{6}$$

*where again $\delta$ is cell area.*

12. **Equation 21: why do you choose Equation 21 to represent temperature?**

    Equation 21 is the temperature forcing as from the Subglacial Hydrology Model Intercomparison Project ( de Fleurian et al., 2018). We have chosen this equation to represent temperature as provides a synthetic way for seasonal and diurnal variations in temperature and thus water discharge to be considered. Additionally, the $\Delta T$ term allows us to impose a climate trend on top of the seasonal and diurnal melt patterns. To clarify this matter we have modified the text to read: *To represent hydrology that varies over seasonally and diurnally, we implement a simple spatially distributed melt model as in SHMIP. . .*

13. **Line 228/Figure 4 caption: From Figure 4, I can see that the sediment discharge is highest at years 19-26, why are you saying the highest discharge is in years 14-17?**

    This was a mistake on our part, and the caption reflected a previous experiment. We have modified the text to read:

    *Model output from alpine topography and forcing over a 30 year run with diurnal and seasonal variations in melt input. Grey box represents time period of increasing glacier melt. a) Seasonally varying water discharge ($Q_w$) increases from year 10 to 20, while till height ($H$) decreases. b) Annual sediment discharge (green) increases over with increasing melt, with highest sediment discharge occurring in year 19, when glacier melt is greatest. Once the new climate stabilizes, annual sediment discharge stabilizes at a higher level than before.*

14. **Figure 4b: the y-labels are the same for the two lines.**

    We thank the reviewer for this comment. However, the orange line represents "instantaneous" sediment discharge and has units $\mathrm{ms}^{-1}$, while the green line is an annual quantity and has units $\mathrm{ma}^{-1}$. We have adjusted some of the text in the caption to reflect this, for example: *Annual sediment discharge (green) increases over with greater melt.*

15. **Figure 4a: it seems the water discharge starts to increase at year 10, but the captions say from year 12. Could you explain?**

    We have addressed this comment above.

16. **Figure 4: subfigure title a, b are missing.**

    These components have been added to the figure.

17. **Line 230: Figure 6 includes 6 subfigures. Which subfigure should I see to support the claim you made here?**

    We have changed the text to reference Figure 6 b, d, f. Text and pointers have also been added to the figures in order to point to processes discussed in the text.

18. **Line 234-239: The descriptions in this paragraph are not well supported by the model results. These descriptions are more like conclusions but are not results. It is hard to link these claims to the results.**

We understand the reviewer's concern on this matter. In response, we have largely moved this material to Section 5.

19. **Line 242: Figure 5 shows the results are distributed values, which means these variables vary with coordinates x and y. However, I didn't see an equation in the method section describing x and y. What equations are solved to obtain these distributed values?**

We thank the reviewer for these comments and as mentioned in comments above, we have provided a more through description of our 2D implementation.

20. **Line 243: Figure 4c is missing.**

The new reference in Figure 4a.

21. **Line 247: Which subfigure in Figure 6 am I supposed to observe to understand the claim here? What is the meaning of early melt season? Do you mean the time at year 8.3? This needs accurate descriptions.**

We modified the phrasing in the text to read. Additionally, we have chosen to reference Figure 4 as it is a simpler figure, lacking the detail and complexity of Figure 6.

Additionally, we have added labels to Figure 4 that point to the features we describe.

22. **Line 250: It will be useful to draw a line in Figure 4 to show which time is spring.**

Done.

23. **Line 251-252: This claim cannot be observed in Figure 4b. I can observe that the highest sediment discharge occurs at years around 11.5 and 16.5. The sediment discharges in this time period do not show a decreasing trend. Where does this claim come from?**

These sentences have been removed from the text. Our comments come from a previous version of the paper.

24. **Line 253: Need to show where is winter time in Figure 4b.**

Done.

25. **Line 266-271: In the paragraph at line 260, Figure 6a, b has been referenced, however, the difference between different subfigures is defined in the paragraph at line 270. This makes it hard to understand the paragraph above. The order of these two paragraphs needs to switch.**

We thank the reviewer for this comment and appreciate the lack of clarity that this could cause. We have largely omitted the first paragraph and moved some of the material to the second paragraph.

The text now reads:

*For the cases described above, bedrock erosion relies only on driving stress and till thickness. Sliding and bedrock erosion did not vary seasonally with increased subglacial water discharge (Figure 4 a). This causes sediment to accumulate during the winter months, which subsequently provides ample material for transport when melt increases in the spring. To test the effects of spatially variable erosion and the role of hydrology, we present two additional cases to supplement the alpine glacier case above, ORIGINAL. One additional case, SEASON, simulates bedrock erosion by only allowing sliding, and thus erosion,*

*during the summer months (e.g., Iken and Bindschadler, 1986); the same erosion relationship is applied as the case in Section 3.1. In this case, however, erosion only occurs when the amount of water input substantially exceeds the background basal melt input rate, that is present in the winter. We choose this case to capture the seasonal variations in bedrock erosion (Ugelvig et al., 2018). In the other additional case, CONST, bedrock erosion remains constant over the entirety of the glacier at a rate of $2\,\mathrm{mm\,a^{-1}}$, independent of glacier sliding velocity.*

26. **Line 291: As you randomly select parameters, does this mean you are performing a sensitivity study? Why do you need to randomly select these parameters?**

Given the results that follow, we are performing a sensitivity study of sorts. We have chosen to randomly select parameters as this reduces the dimensionality slightly compared to using a grid search, with uniformly spaced parameters.

27. **Line 300: As the parameters affect the model results, it is necessary to show the final parameters that give the lowest absolute error.**

The text has been updated to reflect the parameters:

*The parameter search yields an optimum grain size parameter $D_m$ of $2$ cm, sliding parameter $B$ of $2.05 \times 10^{-11}$ $\mathrm{MPa\,m\,s^{-1}}$ and initial till height $H_0$ of $2.5\mathrm{mm}$. The model's ability to reproduce the validation data largely depends on the grain size parameter, $D_m$. Compared to $D_m$, the sliding parameters and initial condition parameters ($B$ and $H_0$) have a reduced influence in representing the data, given that similar values of $B$ and $H_0$ can produce largely different results in the context of $D_m$ (Figure 7).*

Additionally, the parameters are presented in Table 2.

28. **Line 304: From Figure 7, It seems the relative error between the model and observation is very large for most years except for the time between 2015-2016. Why do you only say it has trouble during 2012-2013?**

We thank the reviewer for the comment. In addressing it, we have tried to remove the subjectivity that has been addressed. The text now reads:

*The optimized model reproduces the interannual variability in sediment discharge from the Griesgletscher (Figure 7g). The absolute error between the model and the measurements is roughly $62,600$ $\mathrm{m^3}$. The error from this parameter search is slightly less than half of the $131,300$ $\mathrm{m^3}$ total sediment discharged from the Griesgletscher over this time period (Delaney et al., 2018). The model runs captures the third period from late $2014$ to late $2015$ well. However, the runs systematically overestimate the second and fourth periods and generally underestimate the high discharge period from late $2011$ until late $2013$ (Figure 7g).*

Additionally, we have updated Figure 7 to reflect the outputs of all model runs.

29. **Line 313: How is this claim supported by the results?**

We appreciate the reviewers comments here and understand how this claim may fall outside of the results supported by this section. As a result, we have omitted this comment and moved all comparisons with the one-dimensional model to Section 5.

30. **Line 314: The absolute error corresponding to B and Ht0 vary a lot. How does this support the claim of "minimal influence" of these parameters?**

The error corresponding to $B$ and $H_0$ varies substantially, as the reviewer points out. However, large and small errors can occur for very similar values of $B$ and $H_0$, and the results do not show a systematic change in model output with respect to $B$ and $H_0$ as they do with respect to $D_m$.

[Figure]

[Figure]

Figure 1: Results of the parameter search (a, b, c), the frequency of parameter values that produced a rank correlation of 1 (d, e, f) and the best fit model run amongst the parameter combinations (g). Red stars represent optimum parameter combination. Blue lines represent all model outputs, while gray line represents the optimum parameter combination.

We have slightly modified the text to make this clearer. It now reads:

*The model's ability to reproduce the validation data largely depends on the grain size parameter, $D_m$. The sliding parameters and initial condition parameters ($B$ and $H_0$) have a reduced influence, compared to $D_m$, in capturing the data, given that similar values of $B$ and $H_0$ can produce largely different results in the context of $D_m$ (Figure 7).*

31. **Line 329: In Figure 7e, the units for B is Mpa m/s. How does variable B in Figure B related to the "B" here? It seems the B has a velocity unit here, different from that in Figure 7e.**

The text we have modified the text slightly to read

*The value of $B$, from the parameter search, results in an average sliding velocity of $39\ \mathrm{m}$ $\mathrm{a}^{-1}$, and the range of values for $B$ in the parameter search result in mean sliding velocities roughly between $14\,\mathrm{ma}^{-1}$ and $70\,\mathrm{ma}^{-1}$ (Equation 14).*

We point out that sliding is defined as $u_b = B\tau_b^m$. As a result, the value of $B$ in the parameter search *results* in a sliding velocity, thus we believe that the units are correct.

32. **Line 330: How does this slide speed are calculated? From the model?**

    We appreciate this comment. We have added a reference to Equation 14 in the manuscript (see previous comment). This shows the relationship between basal shear stress and sliding velocity with respect to $B$ that we use to evaluate sliding velocity.

33. **Line 339: How do you calculate this value 2 m$^3$ $s^{-1}$ ?**

    This sentence has been omitted.

    *The best performing model run shows strong temporal variability in sediment discharge. Peaks in sediment discharge occur during the short-lived increases in water discharge (Figure 8 a). Despite the strong dependence on grain size and fluvial transport of sediment in the parameter search, sediment transport capacity $Q_{sc}$ still remains roughly an order of magnitude higher than sediment discharge $Q_s$ (Figure 8 a, b).*

34. **Line 341: I guess sediment discharge should be Qs?**

    This has been corrected.

35. **Figure captions: most of the current figure captions include certain explanations of the results. These explanations make the captions very long and not easy to understand. I suggest only including the descriptions for the line title, legends, and meaning in the captions, but leave the explanations of the results in the main text. Please try to make the captions short but can sufficiently tell the meaning of each line.**

    We thank the reviewer for the comment and understand the concerns. We have adjusted the figure captions accordingly.

---

## Author Response (AR3)

Below, Dr. Hergarten's comments are in bold and our response is in normal font. Changes to the manuscript text are in italics.

Best regards,
Ian Delaney, on behalf of all authors

**However, contrary to the explanation in your response (top of page 2), the physical dimensions in Eqs. 5 and 6 are still not correct. In detail:**

1. **"In this form, Condition ... has units $m^2 s^{-1}$." Right, but since $\nabla$ is inverse length, $\tilde{Q}_s$ would have units $m^3 s^{-1}$, so the same units as $Q_s$.**

2. **"In turn, when we approximate ... becomes $m\,s^{-1}$." Also correct, but then the unit of $\tilde{Q}_s$ becomes $m^4 s^{-1}$!**

3. **"Units of ... make Equation 5 consistent." But not in itself because Q is volume per time and $\nabla$ is $m^{-1}$!**

**I think you really have to fix this problem. As far as I can see, it would work if you define $\tilde{Q}_s$ properly as sediment flux per unit width and then write Eqs. 5 and 6 directly in terms of $\tilde{Q}_s$. Eq. 5 would be dimensionally correct then, and Eq. 6 would also be if you use $\tilde{Q}_s$ everywhere and remove $w$. Maybe you could also write Eq. 8 in terms of $\tilde{Q}_{sc}$ (since this is the genuine form).**
**However, you really have to be careful not to mess up channel width and cell size then since rates of erosion and deposition refer to the area covered by channels only (as far as I can see). This is why I am so picky at this point. I accept that there are many assumptions and simplifications which may be questioned, but I want to be sure that you transferred your ideas into your model correctly.**
**I would also ask you to perform another round of checking the text, in particular the parts that were subject to your extensive revision.**

We appreciate the comment about the units and have worked to make this clearer.
To improve this issue, we have removed $\nabla$ from Equation 6 and replaced it with simply $\widetilde{Q}_s$, sediment mobilization. In each case below, the units for $\widetilde{Q}_s$ are $m^2\,s^{-1}$. By dividing by width of patch of the glacier bed $w$, we approximate $\nabla \cdot Q_s$ as $\frac{\widetilde{Q}_s}{w}$, which has units $m\,s^{-1}$. We had considered re-writing Equation 8 for unit transport width. However, this would omit the channel width scaling that is needed to quantify sediment transport across the glacier bed with evolving channel geometry. We have also pointed out that $w$ is not channel width, but rather the width of a patch of glacier bed over which water may access sediment and remains consistent over the glacier, so $w$ and $w_c$ are different values. For instance, Equation 1 in Wickert and Schildgen (2019) uses a variable valley width in their mass conservation relationship. Because $w$ remains constant here, the term $\frac{\partial w}{\partial x}$, while in 1 dimension in Wickert and Schildgen (2019) would be set equal to 0. In the implementation of the equation $w$ would be cell width, which remains constant as we use a regular square grid here.

The text now reads:

$$\underbrace{\frac{\partial H}{\partial t}}_{\text{till evolution}} = - \underbrace{\nabla \cdot Q_s}_{\text{sediment transport}} + \underbrace{\dot{m}_t}_{\text{bedrock erosion}} , \qquad (1)$$

*$H$ is till thickness and $t$ is time (Table **??**). The first term represents fluvial sediment transport processes, where $\nabla \cdot Q_s$ represents sediment mobilization or deposition. The second term captures bedrock erosion processes, where $\dot{m}_t$ is a bedrock erosion rate.*

*We calculate sediment mobilization in both supply- and transport-limited conditions. Divergence of the sediment flux is evaluated by approximating $\nabla \cdot Q_s$ with $\frac{\widetilde{Q}_s}{w}$ using a similar mobilization scheme as in Delaney et al. (2019)*

$$
\widetilde{Q}_s =
\begin{cases}
\dfrac{Q_{sc} - Q_s}{l} & \text{if } \frac{Q_{sc}-Q_s}{l} \leq \dot{m}_t w & \text{(transport-limited)} & \text{(2a)} \\[3mm]
0 & \text{if } H = H_{lim} \quad \& \quad \frac{Q_{sc}-Q_s}{l} \leq 0 & & \text{(2b)} \\[3mm]
\dfrac{Q_{sc} - Q_s}{l}\,\sigma(H) + \dot{m}_t\, w\,(1 - \sigma(H)) & \text{otherwise} & \text{(supply-limited)} & \text{(2c)}
\end{cases}
$$

*$\widetilde{Q}_s$ is sediment mobilization across a width of the glacier bed $w$ perpendicular to the water's flow direction. Note that $w$ is not necessarily the channel width, but rather a representative width across the glacier bed over which sediment can be accessed by water flowing through the subglacial channel (Figure 2). $Q_{sc}$ is the sediment transport capacity or the maximum amount of sediment that could be transported under the given hydraulic conditions. $l$ is a characteristic length-scale for sediment mobilization, over which sediment mobilization adjusts to sediment transport conditions. $\sigma$ is a sigmoidal function of $H$*

$$
\sigma(H) = \left(1 + \exp\left(\frac{2 - \Delta\sigma H}{5}\right)\right)^{-1}, \tag{1}
$$

*which enables a smooth transition from transport- to supply- limited transport in Equation 2c. If $H$, the till thickness, is greater than $3\,\Delta\sigma$, then the impact on sediment mobilization is negligible and the system is in a transport-limited regime. When $H = \Delta\sigma$, then $\sigma(H)$ is close to $0$ and sediment transport is in a supply-limited regime; no significant sediment mobilization takes place.*

**References**

Delaney, I., Werder, M., and Farinotti, D. (2019). A Numerical Model for Fluvial Transport of Subglacial Sediment. *Journal of Geophysical Research: Earth Surface*, 124(8):2197–2223.

Wickert, A. D. and Schildgen, T. F. (2019). Long-profile evolution of transport-limited gravel-bed rivers. *Earth Surface Dynamics*, 7(1):17–43.

We thank the reviewer for their detailed review, which has greatly improved the manuscript. Generally, we have adapted the text to the reviewer's comments. A figure describing the grid and the locations of different parameters has been included in the text, additionally, we have modified the section describing the routing of the sediment and water, with the aim of improving the clarity. Additionally, we have modified or omitted comments that the reviewer points to that may not be supported in the text.

Below, Reviewer 4's comments are in bold and our response is in normal font. Changes to the manuscript text are in italics.

Best regards,
Ian Delaney, on behalf of all authors

**1   General Comments**

1. **For Section 2.1, the main goal of this section is to provide a way to "estimate" the Hydraulic diameter Dh. The main assumption is that Dh, a characteristic size of subglacial conduits, does not change too much over a certain period. With such an assumption, the Qw in Equation 1 can be estimated using a time-averaged value of discharge during the period. And the pressure gradient can be estimated from the Shreve potential gradient. With these two pieces of information, the Dh in Equation 1 can be approximately estimated and kept unchanged in other conditions. Though the paper finally achieves this goal in lines 109-111, I suggest the authors add the key motivations and assumptions of this Section at the very beginning of the Section, which will improve the readability of section 2.1.**

   We have modified the beginning of this section slightly to read:

   *SUGSET_2D requires a hydraulic model as a means to route sediment and water through the subglacial environment. The hydraulic model determines the sediment transport capacity of the subglacial water, based upon the gradient of the hydraulic potential, channel size, and water flux (Table 1, Section 2.2; e.g., Walder and Fowler, 1994; Alley et al., 1997). The hydraulic model is based on the assumption that subglacial water flows along the hydraulic potential gradient, the weight of ice pressurizes water at the bed (Shreve, 1972), and the channel size varies over a substantially longer time scale compared to water discharge. This model includes characteristics of a Röthlisberger-channel without explicitly describing properties such as creep closure and pressure melt of channel walls (Röthlisberger, 1972).*

   *The gradient of the hydraulic potential of a subglacial channel $\Psi$ (at a certain location and time) can be determined with a known hydraulic diameter $D_h$ (a function of channel size and shape) and water discharge $Q_w$...*

   We believe that this adequately and clearly describes the key components and intent of the section, in terms of 1) the need for the hydraulics model, 2) the link between the hydraulic diameter in Equation 1 and the size and shape of the glacial channel.

2. **Section 2.3.1 need improvements in two aspects: better descriptions for the grid and its sizes/area etc, donor, receiver, and the final linear combinations used for equation 18, 19, and 21; better descriptions for the routing schemes for equations 18-22. First, Section 2.3.1 is trying to obtain distributed values for hydraulic potential, water discharge, sediment discharge, and till height in a 2D domain. In a 2D domain, any grid (with grid id i) has multiple neighboring grids that can exchange potential/discharge/height to its donor grid. And how much exchanged is further affected by the grid size and the approach to calculating gradients between neighbor-**

**ing grids. In short, such a process is similar to the numerical discretization process and usually very complex. In this paper, lines 174-178, 184-185, 190-191, 192-193, 196, and 197 aim to describe these processes, however, are not successful, in my perspective, to explain how the grid size, weights, and equations (18.19,21) are organized together. To help clarify the equations in this section, I suggest the authors sketch to carefully describe mesh, donor, receiver, and variables defined at each donor grid in a 2D grid domain and then use this sketch to improve the description for equations 16-22. To draw such a sketch, the authors can check figure 3.1.1.1 or 3.1.1.2 at this link: `http://www.thevisualroom.com/finite_volume_method_3.html`. In this sketch, it is necessary to clarify the following definitions: lambda, nr, wr,j, phioi, phioj in Equation 18; delta, nd, wd,j, Qwi, Qwj in equation 19, lambda and nd, Qsi, Qsj in equation 21, and Hi, mti in equation 22. With the clarification of these terms in the sketch, section 2.3.1 will become clear.**

We have added the following figure to the text.

[Figure]

Figure 1: Routing scheme on the grid. Solid lines represent cell boundaries, blue squares are cell centers, and red squares are cell edges. $\phi$, the hydraulic potential, decreases in the direction of arrows so that water and sediment generally flow left to right and top to bottom. Edge length($\lambda$) and cell area ($\delta$) are shown. Cell numbers refer to identification in the stack ($s_t$). Select cells denote the weight of donors $w_{d,i,j}$, number of donors $n_d$, donor cells $d_n$, number of receivers $n_r$, and receiver cells $r_s$. Variables and their respective locations on the grid are shown. Some red and blue squares have been removed in some cells for clarity.

Additionally, we have modified Section 2.3.1 to improve its clarity. The specifics are addressed in the comments below.

3. **For the result analyses: in Section 3.1.2, the texts in the paper are not consistent with the Figures shown in Figure 4. In Figure 4 caption, it says "a) Seasonally varying water discharge (Qw) increases from year 10 to 20, while till height (H) decreases.**

**b) Annual sediment discharge (green) increases over with increasing melt". However, Figure 4b sediment discharge does not show an increasing trend, but shows a "decrease, increase, and then decrease" trend. A similar problem occurs for figures 5b,d,f at lines 257-258. In the author's response, the author mentioned this may be caused by using an old version of the manuscript, the authors are suggested to carefully re-exam all results in Section 3.1.2 to make sure the texts in the current version agree with the figures.**

We thank the reviewer for the comments and have carefully reviewed the text to examine these inconsistencies. With regard to the issue in Figure 4 (now Figure 5), the intended meaning was that the increase in $Q_s$ is small compared to the relatively small decreases in $Q_s$. The remainder of the text has been examined and modified to address these specific variations.

4. **For the Conclusion section, the main problem is the Conclusion is not supported by quantitative evidence. The Conclusion in the current version has 4 paragraphs: the first one introduces the 2D model; the second one discusses the limitation, the third one discusses the future work, and the last one has two sentences commenting on the results and two sentences discussing further work. From my understanding, a Conclusion should include what you did and what you have discovered, and provide quantitative evidence to support your discovery. The first paragraph did describe the 2D model, which is good. But no quantitative evidence to support your discovery makes the Conclusion very weak. Also, limitation and future work should NOT be the main texts in the Conclusion because they are not the 'Novel contribution' of the paper. Therefore, I believe the Conclusion section requires significant improvement, which needs to provide concrete QUANTITATIVE evidence to support the main Conclusions of the paper.**

We agree that the conclusion can be strengthened and have made changes to the section. While the quantitative evidence that is presented in the conclusion is quite limited, given the nature of the paper, we have provided a summary of the model, and our opinion that the model provides the basis for modeling subglacial sediment dynamics, the model's limitations, and the model's significance.

*A two-dimensional subglacial sediment transport model, SUGSET_2D, evolves a till layer in response to changing subglacial hydraulic conditions. The model represents sediment transport in supply- and transport-limited regimes, and sediment and water are routed across the bed in response to changing hydraulic conditions in two horizontal dimensions. The till layer is supplied with sediment either from bedrock erosion or by existing sediment, represented by the initial condition. Model cases utilize geometries and hydrological forcings from a synthetic case and Griesgletscher, an alpine glacier in the Swiss Alps.*

*The interdependence of a large number of parameters and their interaction with one another, for instance, sliding and erosion (Equations 12 to 15), point to the complexity of sediment transport in the subglacial system. Furthermore, the model's limited representation of the magnitude of interannual variability in the Griesgletscher simulation, from $2011$ to $2017$, points to processes not completely represented in this application of the model. This misfit could come from poorly constrained parameters and external factors, such as model inputs that may limit the model's accurate representation of sediment discharge observations. These include interannual variability of glacier velocity and, thus, bedrock erosion, changing glacier topography that routes water to different patches of the glacier bed over time, and routing of water to the glacier bed.*

*Additional insights into subglacial erosion and sediment transport processes over decadal timescales can be gained from more sophisticated parameterizations of bedrock erosion and subglacial hydrology. Even so, the foundational processes of the model presented*

*here should be considered when examining subglacial sediment transport processes at seasonal to decadal scales. These processes include: 1) fluvial transport of subglacial sediment across a glacier's bed in two dimensions in supply- and transport-limited regimes, 2) spatially-distributed bedrock erosion and sediment production, and 3) variable water routing in response to changing melt and hydraulic conditions. It is our hope that the model will be applied in the context of field observations to evaluate and isolate subglacial processes controlling sediment discharge from glaciers as they change.*

**2   Specific Comments**

1. **Line 88: change hydraulic gradient to hydraulic pressure gradient**

   We have changed the text to read: *gradient of the hydraulic potential*.

2. **Equation 1: maybe change $\Psi$ to $\Psi(x, y, t)$, which helps to clarify the pressure gradient is a time-dependent and 2D distribution variable.**

   We have chosen not to present $\Psi$ as $\Psi(x, y, t)$ as we intend for this section to establish the theory for the hydraulics model. However, we have added the following text to the paragraph preceding Equation 1:

   *The gradient of the hydraulic potential of a subglacial channel $\Psi$ (at a certain location and time) can be determined with a known hydraulic diameter $D_h$ (a function of channel size and shape) and water discharge $Q_w$.*

3. **Equation 2: can you a sketch the angle $\beta$ in Figure 1 or 2?**

   Figure 2 has been modified to the following:

[Figure]

Figure 2: llustration of model cell (a), detailing the layers of bedrock, till, water, and ice. Characteristics of the subglacial channel are noted as a polygon but shown in one dimension for clarity in (b) with Hooke angle parameterization with two different channel shapes for different values of $\beta$, Equations 2, 9 and 10.

4. **Line 96: change "establish" to "estimate" or "approximate"?**

   We have changed the text to "establish."

5. **Line 98: change "we call the source percentile" to "which is called the source percentile".**

   Done.

6. **Line 97-98: How the surface melt and resulting discharge control the subglacial conduit diameter is a complicated problem. Could you add a few citations to support this assumption, if any?**

   We appreciate this comment. Upon re-reading the text, we have moved the citations in the next sentence to this one.

7. **Line 103: I think Qw is the instantaneous discharge, while Qw\* is a representative scale of the instantaneous discharge. What is the "total instantaneous amount"?**

   The text has been clarified to read: *We sum the prescribed melt rate $\dot{m}_w$ up the glacier to define $Q_w$, not considering englacial water storage. . . .* The intent of this sentence is to make clear that the variations in water discharge occur as a result of melt, thus processes such as englacial storage that account for the transit time of water through the glacier are not accounted for.

8. **Line 96-104: some of the texts here are confusing. Qw is the water inside the subglacial conduits. Without considering ice melting inside the conduit, such water should be discharged from surface melting, therefore, I would say water in the conduit equals the surface melting into the conduit, based on mass conservation, therefore, Qw = mw. In general, the surface melting varies with time, so here you assume mw is estimated by Qw\* which could be a time-averaged value over a certain period (hours to days as mentioned). In thinking in this way, it is logical converting equation 1 to equation 3. However, lines 96, 99-100, and 103-104 cause confusion. I would suggest revising the texts between 96-104 to better reflect the logic as I suggested above.**

   We appreciate the reviewer's comment on this matter. We have modified the text to the following to address the comments.

   We sum the prescribed melt rate $\dot{m}_w$ up the glacier to define $Q_w$, not considering englacial water storage. Percentile $s_p$ over a response time period prior to the timestep $s_a$ is applied to $Q_w$ to evaluate a characteristic water discharge $Q_w^*$ that represents the size of the conduit (hours to days; c.f. Gimbert et al., 2016; de Fleurian et al., 2018; Delaney et al., 2019; Nanni et al., 2020). The timescales, $s_a$, and characteristic water discharges ($s_p$ and $Q_w^*$), responsible for changes in subglacial conduit size are poorly constrained, yet their impact can be intuited. For instance, short-lived increases in water discharge due to an hour of precipitation will not greatly impact the hydraulic diameter of the subglacial channel, whereas prolonged melt would increase the hydraulic diameter.*We assume that the hydraulic diameter $D_h$ of the channel results from a characteristic water discharge $Q_w^*$ which is evaluated by the source percentile of water discharge over a certain time period $s_p$ and a response time of the channel size $s_a$, that remains consistent throughout the model run (Table 1; Delaney et al., 2019).*

   *We sum the prescribed melt rate $\dot{m}_w$ up the glacier to define $Q_w$, not considering englacial water storage. Percentile $s_p$ over a response time period prior to the timestep $s_a$ is applied to $Q_w$ to evaluate a characteristic water discharge $Q_w^*$ that represents the size of the conduit (hours to days; c.f. Gimbert et al., 2016; de Fleurian et al., 2018; Delaney et al., 2019; Nanni et al., 2020). The timescales, $s_a$, and characteristic water discharges ($s_p$ and $Q_w^*$), responsible for changes in subglacial conduit size are poorly constrained, yet their impact can be intuited. For instance, short-lived increases in water discharge due to an*

*hour of precipitation will not greatly impact the hydraulic diameter of the subglacial channel, whereas prolonged melt would increase the hydraulic diameter.*

*While we understand the reviewer's recommendation to convert Equation 1 to Equation 3, we have chosen not to. Equation 1 represents the fundamental form of the Darcy-Weisbach, while Equation 3 represents our implementation and application of it. As a result, we believe it best to present Equation 1 first, as to clarify the basis for our application of the equation.*

9. **Line 99-102: Two variables, sa and sp are introduced. And further explanations are added. However, these two variables are not used in Equations 3 and 4. They may be not useful to help us understand why equation 3 could be derived from equation 1, but may confuse readers. I suggest removing this information from lines 96-104.**

   *We have moved the description of these variables and their influence on the hydraulic diameter down the text. The text is presented in the comment above.*

10. **Line 109: is Dh a function of x and y?**

    *In the model implementation, $D_h$ varies in $x$ and $y$ directions. However, in this section we present the theoretical underpinnings of the model, as opposed to the model implementation, so we have not represented $D_h$ as a function of $x$ and $y$. As mentioned above, we have added text to clarify that both $\Psi$ and $D_h$ vary in $x$ and $y$ directions.*

11. **Line 109: Add one sentence at the beginning of this paragraph: With the data of representative surface melting rate Qw* and the static hydraulic pressure gradient, a representative hydraulic diameter Dh can be estimated. For a given short period, such a Dh is assumed time-independent and used in Equation 1.**

    *We thank the reviewer for the input and appreciate the clarification offered. It has been implemented. The text reads: With knowledge of $D_h$, we insert the instantaneous value of $Q_w$ into Equation 1 to evaluate the instantaneous gradient of the hydraulic potential $\Psi$.*

12. **Line 124: How does this conversion happen? Do you mean Qs = Qs /w?**

    *See the response to the next question.*

13. **Equation 6: What is the necessity to use Qs rather than Qs? Trying to use variables as less as possible, if existing variables are sufficient to tell the story. Revise equation 6a to a formula of Qs but not Qs .**

    *The other reviewer pointed to concerns with this equation and it has been modified somewhat. The term $\widetilde{Q}_s$ is sediment mobilization across a width of the glacier bed perpendicular to the flow of water, thus it has units $m^2 s^{-1}$. This is different than sediment flux (units: $m^3 s^{-1}$) or $\lambda \cdot Q_s$ (units: $m s^{-1}$) in Equation 5. We scale the equation with the width of the glacier bed to implement the divergence of the flux in Equation 5. To clarify this matter we have adjusted the text to read:*

    *We calculate sediment mobilization in both supply- and transport-limited conditions. Divergence of the sediment flux is evaluated by approximating $\nabla \cdot Q_s$ with $\frac{\widetilde{Q}_s}{w}$ using a similar mobilization scheme as in Delaney et al. (2019)*

$$\widetilde{Q}_s = \begin{cases} \dfrac{Q_{sc} - Q_s}{l} & \text{if } \frac{Q_{sc}-Q_s}{l} \leq \dot{m}_t w & \text{(transport-limited)} \quad (1a) \\[2ex] 0 & \text{if } H = H_{lim} \quad \& \quad \frac{Q_{sc}-Q_s}{l} \leq 0 & (1b) \\[2ex] \dfrac{Q_{sc} - Q_s}{l}\, \sigma(H) + \dot{m}_t\, w\, (1 - \sigma(H)) & \text{otherwise} & \text{(supply-limited)} \quad (1c) \end{cases}$$

$\widetilde{Q}_s$ is sediment mobilization across a width of the glacier bed $w$ perpendicular to the water's flow direction. Note that $w$ is not necessarily the channel width, but rather a representative width across the glacier bed over which sediment can be accessed by water flowing through the subglacial channel (Figure 2). $Q_{sc}$ is the sediment transport capacity or the maximum amount of sediment that could be transported under the given hydraulic conditions. $l$ is a characteristic length-scale for sediment mobilization, over which sediment mobilization adjusts to sediment transport conditions.

14. **Equation 15: what is the difference between $\tau$ in Equation 11 and $\tau_b$ in equation 15? Do they equal to each other?**

    *We appreciate the need for clarification on this comment. $\tau$ is the shear stress between flowing water and the channel wall (i.e. bedrock, ice, or sediment). $\tau_b$ is the basal shear stress in the sliding relationship. To better clarify these differences, the text now reads: We also determine the shear stress between water flowing through the channel and the sediment below . . .*

15. **Equation 15: does $\sin\alpha = \lambda z_s$ ?**

    *No. This is simply the sine of the surface slope.*

16. **Line 172: What is the purpose of flow routing? Is it used to distribute water potential at different cells?**

    *We have added the following comments to the text: A routing scheme is implemented to (a) evaluate the hydraulic potential and thus the direction of the water flow and (b) transport sediment and water across the glacier bed, to where it is expelled or deposited.*

17. **Line 172: change "multicell routing scheme" to "2-D distributed routing scheme"?**

    *Done.*

18. **Line 174: what is a "regular grid"? square grid or rectangular grid? Also, the result in Figure 3 shows an irregular distribution, please clarify this.**

    *Excellent point and we have modified the text to read: on a regular grid with square cells, extending in $x$ and $y$ directions. In the figure, a regular grid is implemented, however, values outside of the glacier have been omitted. Thus the domain has an irregular shape, but the grid is regular.*

19. **Line 174: What do "fluxes" mean? Is water flux driven by pressure gradients?**

    *We have modified the text to read: water and sediment fluxes can pass to the four surrounding cells sharing an edge, in response to the hydraulic pressure gradient*

20. **Line 175: Not sure what is a "stack" in Table 3. It is better to draw a 2D sketch to visualize st, nd, and nr for a single cell. See major comments 1b.**

    *In the text we state: This routing algorithm returns a stack ($s_t$; Table 3), which is a vector that contains information about the order of cells to perform the calculations. . .*

    *Additionally, we are grateful for the reviewer's recommendation to make a figure about the grid (see above).*

21. **Line 175-178: These sentences describe the process of numerical discretization. The value of the cell, i.e., a stack defined here, is a linear combination of the values of certain variables of neighboring cells. There are different types of numerical discretizations and thus result in different linear combinations. As different combination means different discretization schemes have different accuracy, so it is generally required to describe what discretization schemes are used. The current**

*sentences are very general descriptions of the discretization principle. Please refer to the major comment 1b to better describe this part.*

*This as been addressed this above.*

22. **Equation 18: In lines 176-177, you defined donors and receivers, why did you only use the information of potential from receives, i.e., summation from 1 to nr in Equation 18. Here the wr,j is determined by discretization schemes. How do you calculate this?**

*We have slightly modified the text to read: the weight or the percentage of hydraulic potential and water or sediment discharge directed from one cell to another ($w_d$ or $w_r$)...*

*The summation over 1 to $n_r$ represents the cells that are directed at a cell. For instance, if $3$ cells donate to a cell, then $n_r = 3$.*

23. **Line 186-187: It is very hard to understand this sentence by reading just words. You have to provide a figure to describe what is donor, receiver, edge length, etc. This is a typical practice for papers that involve numerical discretization algorithms. Also, for 1D grid, it only has one edge length. But for 2D grid, it has two edges and thus two edge lengths. For the regular grid, if the grid is square, then we can say it has only one edge, but if it is rectangular, it has two edges. As your paper is a 2D model, please make sure if you use a square grid or not.**

*We thank the reviewer for this comment and have worked to make this section clearer. A figure with the grid has been added. Additionally, in both the figure, text, and Table 3 we have clarified that the grid is square, or put differently there is a single value of edge length $\lambda$.*

24. **Line 188: you mentioned "beginning at the based and moving up to the glacier", does this sentence mean the calculation is performed along a line from the base to somewhere? What does "flow paths evaluated in the routing scheme mean? You need to provide a sketch to carefully describe these details.**

*We have tried to address this comment by modifying the text to the following sentence: The operation is executed on a cell-by-cell basis, beginning with cells that have no receivers, such as those near the glacier terminus, and moving up the glacier using the inverted stack in $s_t$ (Figure 4).*

25. **Equation 19: So you are applying the routing scheme to calculate a distributed discharge for water? Again, what is $\delta$, $w_{d,j}$? And why you only calculate summation over donor cells? These really require a detailed figure to describe this. Combining equations 19 and 18, there have too many variables, it is impossible to accurately understand the meaning and how you calculate without a detailed sketch. By the way, as you give $\dot{m}_t$ different value for different cell, so do you mean you prescribed a distributed meltwater source $\dot{m}_{w,i}$ term along a certain line?**

*We have modified the text slightly to read: Using the routing scheme above, we evaluate the water discharge from cell $i$, $Q_{w,i}$, from melt upstream as*

$$Q_{w,i} = \sum_{j=1}^{n_d} Q_{w,j} \cdot w_{d,i,j} + \dot{m}_{w,i} \cdot \delta \ , \tag{2}$$

*where $n_d$ is the number of donor cells for cell $i$, $w_{d,i,j}$ is the percentage of water flow from cell $j$ to cell $i$, and $\dot{m}_{w,i}$ is a prescribed meltwater source term in cell $i$.*

26. **Line 192: What is $Q_{s,i}$? Where does this term come from and why you need to calculate this term? Most importantly, how are equations 20a-c derived? The term "like-wise" does not explain how this term is derived.**

*This equation is the implementation of 6 in the code. In response to this comment, we have reorganized the text slightly to read:*

*The amount of sediment leaving a cell $i$, $Q_{s,i}$, is the flux into the cell plus the sediment mobilized in the cell, which is defined as*

$$Q_{s,i} = \sum_{j=1}^{n_d} Q_{s,j} \cdot w_{d,i,j} + \widetilde{Q}_{s,i} \cdot \lambda. \tag{3}$$

*The first term is the flux of sediment into the cell $i$ from donor cells $j$. The second term is sediment mobilization, $\widetilde{Q}_{s,i}$ in cell $i$, which is computed by implementing Equation 6 as*

$$\widetilde{Q}_{s,i} = \begin{cases} \sum_{j=1}^{n_d} \left( \dfrac{Q_{sc,j} - Q_{s,j}}{l} \cdot w_{d,i,j} \right) & \text{if } \sum_{j=1}^{n_d} \left( \dfrac{Q_{sc,i,j} - Q_{s,i,j}}{l} \right) \cdot w_{d,i,j} \leq \dfrac{\dot{m}_t \lambda}{n_d} \tag{4a} \\[2em] 0 & \text{if } H_j = H_{lim} \quad \& \quad \dfrac{Q_{sc,i,j} - Q_{s,i,j}}{l} \leq 0 \tag{4b} \\[2em] \dfrac{\dot{m}_{t,i} \lambda}{n_d} \left(1 - \sigma(H)\right) + \sum_{j=1}^{n_d} \left( \dfrac{Q_{sc,j} - Q_{s,j}}{l} \right) \cdot \sigma(H) \cdot w_{d,i,j} & \text{otherwise} \tag{4c} \end{cases}$$

*where $Q_{sc,j}$ is the sediment transport capacity from cell $j$ flowing to $i$, $Q_{s,j}$ is sediment discharge entering from cell $j$ to cell $i$, $l$ is a response length scale, and $\lambda$ is edge length.*

27. **Line 195: Need a figure to describe what is $\lambda$? What is a response length scale? And where is this come from?**

    *Done. Response length scale $l$ is defined above, in Equation 6.*

28. **Line 196: how is equation 21 derived? Why Qsi =. . . ?**

    *See the response to Line 192, above.*

29. **Line 198: Need to define cell area?**

    *Done.*

30. **Line 215: Need a figure to show where the edge cells are.**

    *Done.*

31. **Section 3.1 title: You only have one synthetic case. Use a singular form**

    *Done.*

32. **Line 235: It is better to reproduce and visualize the synthetic glacier geometry in your paper.**

    *We have referenced the figure with appropriate geometry.*

33. **Line 238: what is "laterally"? You haven't defined any coordinates here, so no way to understand "laterally".**

    *The text now reads: variable ice thickness mean that variable hydrologic gradients will occur perpendicular to the flow, thus water and sediment are routed across multiple cells.*

34. **Equation 23: What is the unit of T? For 0, is the unit K or C? Line 249: o is not the same as Celsius (oC).**

    *The units have been switched to C.*

35. **Line 255: Figure 4 appears earlier than Figure 3.**

    *This has been resolved.*

36. ***Line 256-257: In Figures 5b,d,f, each figure has two lines. One is purple and the other is yellow. So which line is the "Daily-averaged sediment discharge"? However, both curves in 5b,d,f shows complex behaviors. For purple lines b and f, they show decreasing, increasing, and decreasing trends. For d, it shows an increasing, constant, and decreasing trend. The yellow lines show more complex behaviors. In short, figures 5b,d,f show different behaviors compared to what you say at line 257 (decreases until....). Please carefully analyze the figures and make your text descriptions consistent with your figure.***

    *Maximum and average quantities of daily sediment discharge decrease until the very end of the melt season, when sediment discharge increases very slightly again*

    *The following text has been added to the figure caption Data are plotted at a $6\,\mathrm{hr}$ interval so that daily maximums and minimums are visible.*

37. ***Line 264: How do you define mean till height?***

    *The text now reads: the mean till height across the glacier*

38. ***Line 295: Can you elaborate on the 4 time periods?***

    *We modified the text very slightly. Subglacial sediment discharge from the glacier is determined over four different time periods ($2011\text{-}2013$, $2013\text{-}2014$, $2014\text{-}2015$, $2015\text{-}2016$) by differencing the bathymetry maps collected through this period and considering proglacial erosion quantities (Delaney et al., 2018, 2019).*

    *However, the citation for Delaney et al. (2018) is given, and we believe that at present the information is adequate for the model application here.*

39. ***Line 314: add (see red stars in Figure 7a-c) after "... till height H0 of 2.5 cm".***

    *Done.*

40. ***Line 319: Why is the data a constant within each year? For example, the data is constant between late 2011 - late 2013?***

    *We have debated about how best to present the data and thought that the uneven time periods and long time spans made it so that the sum over a time period would be most logical. We also considered presenting the data as a table, however, in this case, it would be far more difficult to present the ensemble of model runs together. We have altered the plot to represent an average flux of sediment over the time period $\mathrm{m^3\,a^{-1}}$. This figure has been modified to:*

[Figure]

[Figure]

Figure 3: Results of the parameter search (a, b, c), the frequency of parameter values that produced a rank correlation of 1 (d, e, f) and average sediment flux from model run amongst the parameter combinations over the time periods (g) in the synthetic alpine glacier. Red stars represent the optimum parameter combination with an absolute error of roughly $62,600 \mathrm{~m}^3$. Blue lines represent all model outputs, while the gray line represents the optimum parameter combination.

41. **_Line 319: From Figure 7a,b,c, What is the absolute error for the optimal run (red star case)? Is the value 62,600? I can not see it clearly from the y-axis in Figure 6a._**

    _The text has been modified to read:_

    _Red stars represent the optimum parameter combination with an absolute error of roughly_ $62,600 \mathrm{~m}^3$.

42. **_Line 325: change "short-lived' to "short-lived period"._**

    _The text now reads: short periodic increases in water discharge ._

43. **_Line 325: This claim needs supporting evidence. From Figure 8a, you can identify the peak values for the sediment discharge. Meanwhile, you can obtain the corresponding water discharge. You can calculate a correlation between the two values. If the correlation coefficient is high, then this claim is supported._**

    _We have modified the text slightly to the following. Additionally, we have referenced Figure 10 in the text. While this figure does not have information on a correlation coefficient, it is evident that the highest water discharge values do not necessarily result in the greatest sediment discharge values._

    _Some of the peaks in sediment discharge occur during the short periodic increases in water discharge. Yet the greatest sediment discharge values do not necessarily occur at the highest water discharge values (Figure 9 a and Figure 10 a)._

44. **_Line 328: what is "high on the glacier"?_**

*high on the glacier (lower left of panels in Figure 10)*

**References**

Alley, R. B., Cuffey, K. M., Evenson, E. B., Strasser, J. C., Lawson, D. E., and Larson, G. J.: How glaciers entrain and transport basal sediment: physical constraints, Quaternary Science Reviews, 16, 1017–1038, doi:10.1016/S0277-3791(97)00034-6, 1997.

de Fleurian, B., Werder, M. A., Beyer, S., Brinkerhoff, D., Delaney, I., Dow, C., Downs, J., Hoffman, M., Hooke, R., Seguinot, J., and Sommers, A.: SHMIP The Subglacial Hydrology Model Intercomparison Project, Journal of Glaciology, 64, 897–916, doi:10.1017/jog.2018.78, 2018.

Delaney, I., Bauder, A., Huss, M., and Weidmann, Y.: Proglacial erosion rates and processes in a glacierized catchment in the Swiss Alps, Earth Surface Processes and Landfroms, 43, 765–778, doi:10.1002/esp.4239, 2018.

Delaney, I., Werder, M., and Farinotti, D.: A Numerical Model for Fluvial Transport of Subglacial Sediment, Journal of Geophysical Research: Earth Surface, 124, 2197–2223, doi:10.1029/2019JF005004, 2019.

Gimbert, F., Tsai, V. C., Amundson, J. M., Bartholomaus, T. C., and Walter, J. I.: Subseasonal changes observed in subglacial channel pressure, size, and sediment transport, Geophysical Research Letters, 43, 3786–3794, 2016.

Nanni, U., Gimbert, F., Vincent, C., Gräff, D., Walter, F., Piard, L., and Moreau, L.: Quantification of seasonal and diurnal dynamics of subglacial channels using seismic observations on an Alpine glacier, The Cryosphere, 14, 1475–1496, doi:10.5194/tc-14-1475-2020, 2020.

Röthlisberger, H.: Water pressure in intra– and subglacial channels, Journal of Glaciology, 11, 177–203, 1972.

Shreve, R. L.: Movement of water in glaciers, Journal of Glaciology, 11, 205–214, 1972.

Walder, J. S. and Fowler, A.: Channelized subglacial drainage over a deformable bed, Journal of Glaciology, 40, 3–15, doi:10.3189/S0022143000003750, 1994.

---

## Author Response (AR4)

Below, Reviewer 4's comments are in bold and our response is in normal font. Changes to the manuscript text are in italics.

Best regards,
Ian Delaney, on behalf of all authors

- **Line 10: change "Experiments" to "Numerical experiment".**

  Done.

- **Line 12: add "also" between "we" and "apply.**

  Done.

- **Equation 1: It seems wc was not explained. Also, in the previous version of the paper, Equation 1 does not have a wc term, but you have one in this version, could you explain the difference?**

  This was a mistake and the term as been removed.

- **Equation 3: Equation 1 has the variable wc, while Equation 3 does not. Please check this inconsistency.**

  Please see the comment above.

- **Line 124: Change "the first term" to "the first term on the right-hand side".**

  Done.

- **Line 128: what is the difference between "a width of the glacier bed w" at line 128 and "channel width wc" defined at Equation 9? Are they the same or not? If they are the same, please explicitly describe this. If not, then please explain the difference and clarify how to calculate glacier bed width w.**

  The text now reads:

  *Note that $w$ is not necessarily the channel width, but rather a representative width across the glacier bed over which sediment can be accessed by water flowing through the sub-glacial channel (Figure 2). The channel width $w_c$ is used to calculate the width over which to apply the sediment discharge capacity and is discussed below in Equation 9, that converts the hydraulic diameter $D_h$ to channel width.*

- **Equation 8: In the previous version, the right-hand term is multiplied by wc, but in this new version, the wc is omitted. Is there a reason to do this?**

  This has been added.

- **Line 135: In Equation 7, is the term, (2 - Delta sigma H)/5, a dimensionless value? I suspect the format Delta sigma\*H should be Delta sigma/H? From the texts at Line 135, Delta sigma has the same unit of H. The unit will be the squared unit of H if you multiply Delta sigma with H. Please add clarification for this. If I am correct that (2 - Delta sigma H)/5 should be (2 - Delta sigma/H)/5, then H = Delta sigma, means sigma(H) = (1+exp(1/5))$^{-1}$ = 0.45, which is not 0 as you mentioned at line 135. Please check on this.**

  We thank the reviewer for this comment, and note that the other reviewer brought similar concerns to light. The proper equation, and the one used in the code, is

$$\sigma(H) = \left(1 + \exp\left(10 - 5\frac{H}{\Delta\sigma}\right)\right)^{-1}. \tag{1}$$

The units of $\Delta\sigma$ are now $\mathrm{m}$ (Table 3), so the $\sigma(H)$ term is dimensionless. The text has been rewritten as *As $H$ approaches $\Delta\sigma$....*

- **Line 254: add parenthesis to separate T and C?**

  Done.

- **Line 339: Here you mentioned that the grain size is the most influential factor controlling the model's predictive capability. In my understanding, the grain size also has impacts on the Darcy- Weisbach friction factor. In this paper, the friction factor is assumed as a constant. Could you add a few comments on how the combined impacts of grain size on friction factor (Equations 1 and 11) and transport capacity (Equation 8) can likely affect the mode performance?**

  We appreciate this comment and believe that this is topic of important research. However, we are hesitant to comment on this given the assumptions of our hydraulics model and poorly constrained hydraulic factors, such as channel shape, sinuosity, and the variations between bedrock and sediment that could cause the factor to in response to till height.

  However, we have brought the matter to the reader's attention by including this comment in the "Model limitations" section:

  *Lastly, we have chosen single friction factor $f_r$ for the entirety of the run. This factor can vary in time (Pohle et al., 2022) and can be impacted by other factors such as sediment grain size or bedrock along the channel bed.*

- **Line 359: add a space between "13" and "are".**

  Done.

**References**

Pohle, A., Werder, M. A., Gräff, D., and Farinotti, D. (2022). Characterising englacial R-channels using artificial moulins. *Journal of Glaciology*, page 1–12.

Below, Dr. Hergarten's comments are in bold and our response is in normal font. Changes to the manuscript text are in italics.

Best regards,
Ian Delaney, on behalf of all authors

- **Eq. 1 has an additional factor $w_c$ compared to your 2019 paper, which makes it dimensionally incorrect.**

  This factor has been removed, and its addition was an unintentional.

- **The sigmoidal function (Eq. 7) is the same as in your 2019 paper, but the interpretation in the text is wrong. The property $\Delta\sigma$ must be inverse length, so that something like $\Delta\sigma = H$ makes no sense.**

  We appreciate this comment and am glad that Dr. Hergarten brought it to our attention. We have changed the text to read *As $H$ approaches $\Delta\sigma$*. Additionally, while working on another project it was pointed out by a co-author that the equation for $\sigma$ should be written as:

$$\sigma(H) = \left(1 + \exp\left(10 - 5\,\frac{H}{\Delta\sigma}\right)\right)^{-1}, \tag{1}$$

  The equation in the manuscript in has been updated. Note that the sediment connectivity factor (or mobilization height) is treated differently in **?**, such that here $H$ is divided by $\Delta\sigma$, where as they are multiplied in **?**. However, the units here have been changed such that $\Delta\sigma$ is in $\mathrm{m}$.

- **$Q_{sc}$ in Eq. 8 is per unit width. So the channel width is missing (compare to your 2019 paper).**

  The term $w_c$ has been added to the equation.